# Infinite Horizon Markov Economies

**Denizalp Goktas**[*]
Cornell Tech, Computer Science

**Sadie Zhao**[*] **& Yiling Chen**
Harvard University, Computer Science

**Amy Greenwald**
Brown University, Computer Science

## Abstract

We study Markov pseudo-games, a framework that unifies Markov games and pseudo-games by incorporating both dynamic uncertainty and action-dependent feasibility. We establish the existence of equilibria and present a first-order solution method with polynomial-time guarantees. As an application, we show that recursive Radner equilibria in infinite horizon Markov exchange economies can be formulated as equilibria of concave Markov pseudo-games, thereby establishing their existence and enabling algorithmic approximation. Finally, we demonstrate the practical effectiveness of our method by implementing a generative adversarial policy network and computing equilibria in several infinite horizon economies.

## 1 Introduction

A central goal of economics is to understand how rational agents interact in dynamic, uncertain environments and how such interactions give rise to equilibria. From Walras' model of markets as systems of supply and demand, to Arrow & Debreu's competitive economies as pseudo-games, *general equilibrium theory* has provided a rigorous mathematical framework for modeling economies. Yet, classical formulations are inherently *static*: they capture only a single period of trade and, even when commodities are made contingent on future states, they rely on the unrealistic assumption of a *complete* market, meaning a full set of state-contingent assets. As a result, they fail to capture the ongoing uncertainty and dynamic decision-making that characterize real-world economies, especially those involving sequential trade of financial assets, intertemporal borrowing and lending, and evolving shocks to productivity or preferences.

Radner (1972) introduced *stochastic exchange economies*, finite-horizon models in which agents trade commodities and assets under uncertainty, leading to the canonical notion of *Radner equilibrium*. These models provide a dynamic extension of Arrow–Debreu economies by linking a sequence of spot markets through asset trade, but they retain a finite time horizon. Beyond the finite, infinite horizon stochastic economies are particularly attractive for macroeconomic applications, as they are better suited to modeling long-run phenomena such as asset bubbles (Huang & Werner, 2000), economic growth, and persistent aggregate shocks. However, infinite horizons introduce substantial theoretical difficulties. For example, they permit the possibility of Ponzi schemes in asset markets, in which agents indefinitely roll over debt, thereby complicating equilibrium existence in incomplete markets. Magill & Quinzii (1994) extended Radner's framework to the infinite-horizon setting, albeit with financial assets rather than arbitrary assets, and provided conditions under which sequential competitive equilibria exist. Despite these advances, computational progress has been limited, with most solution methods still largely confined to finite-horizon environments (Sargent & Ljungqvist, 2000; Taylor & Woodford, 1999; Fernández-Villaverde, 2023). These difficulties highlight the need for new computational methods and theoretical frameworks in which to analyze infinite horizon models.

We address this challenge in the context of *Markov pseudo-games*, a model we develop that combines the dynamic uncertainty of Markov games with the action-dependent feasibility of pseudo-games. This framework not only extends Arrow & Debreu's pseudo-games to a dynamic setting, but it is also

---

[*]Equal contribution.

sufficiently expressive to model *infinite horizon Markov exchange economies with potentially incomplete markets*. In this way, our framework unifies the game-theoretic perspective and the economic perspective: it serves as a Markov game model with computable equilibria, while simultaneously capturing the structure of general infinite horizon economies.

**Contributions** In Section 2, we introduce Markov pseudo-games (MPGs), and we establish the existence of (pure) *generalized Markov perfect equilibria (GMPE)* in concave Markov pseudo-games (Theorem 2.1), extending Arrow–Debreu's classical existence result for pseudo-games to dynamic settings. This result implies the existence of pure (or deterministic) Markov perfect equilibria in a large class of continuous-action Markov games for which existence, to the best of our knowledge, was heretofore known only in mixed (or randomized) policies (Fink, 1964; Takahashi, 1964). Although the computation of GMPE is PPAD-hard in general, by taking advantage of the recent progress on solving generative adversarial learning problems (e.g., (Lin et al., 2020; Daskalakis et al., 2020)), we show that an approximate stationary point of the exploitability (i.e., the players' cumulative maximum regret) can be computed in polynomial time under mild assumptions (Theorem 2.2). This result implies that a policy profile that satisfies necessary first-order stationarity conditions for a GMPE in Markov pseudo-games with a bounded best-response mismatch coefficient (Lemma 5) can be computed via policy gradient in polynomial time, a result which is analogous to known computational results for zero-sum Markov games (Daskalakis et al., 2020).

In Section 3, we introduce an extension of Magill & Quinzii (1994)'s infinite horizon exchange economy, which we call an *infinite horizon Markov exchange economy*.[1] The Markov restriction allows us to establish the existence of a *recursive Radner equilibrium (RRE)* (Theorem 3.1). Our proof reformulates the set of RRE of any infinite horizon Markov exchange economy as the set of GMPE of an associated Markov pseudo-game (Theorem 3.1). To our knowledge, ours is the first result of its kind for such a general setting, as previous recursive competitive equilibrium existence proofs were restricted to economies with one consumer (also called the representative agent), one commodity, or one asset (Mehra & Prescott, 1977; Prescott & Mehra, 1980). The aforementioned computability results allow us to conclude that an approximate stationary point of the exploitability of the Markov pseudo-game associated with any infinite horizon Markov exchange economy can be computed in polynomial time (Theorem 3.2).

Finally, in Section 4 we implement our policy gradient method in the form of a generative adversarial policy network (GAPNet), and use it to search for RRE in three infinite horizon Markov exchange economies with three different types of consumer utility functions. Experimentally, we observe that GAPNet finds approximate equilibrium policies that are closer to GMPE than those produced by a standard baseline for solving stochastic economies.

## 2 MARKOV PSEUDO-GAMES

We begin by developing our formal game model.[2] The games we study are stochastic, in the sense of Shapley (1953), Fink (1964), and Takahashi (1964). Further, they are pseudo-games, in the sense of Arrow & Debreu (1954a), as the players' feasible action sets are determined by other players' choices. We model stochastic pseudo-games, and dub them *Markov pseudo-games* (MPGs);[3] the games are Markov in that the stochastic transitions depend only on the most recent state and player actions.

**Model** An *(infinite horizon discounted) Markov pseudo-game* $\mathcal{M} \doteq (n, m, \mathcal{S}, \mathcal{A}, \mathcal{X}, \boldsymbol{r}, p, \gamma, \mu)$ is an $n$-player dynamic game played over an infinite discrete time horizon. The game starts at time $t = 0$ in some initial state $S^{(0)} \sim \mu \in \Delta(\mathcal{S})$ drawn randomly from a set of states $\mathcal{S} \subseteq \mathbb{R}^l$. At this and each subsequent time period $t = 1, 2, \ldots$, the players encounter a state $\boldsymbol{s}^{(t)} \in \mathcal{S}$, in which each player $i \in [n]$ simultaneously takes an *action* $\boldsymbol{a}_i^{(t)} \in \mathcal{X}_i(\boldsymbol{s}^{(t)}, \boldsymbol{a}_{-i}^{(t)})$ from a *set of feasible actions* $\mathcal{X}_i(\boldsymbol{s}^{(t)}, \boldsymbol{a}_{-i}^{(t)}) \subseteq \mathcal{A}_i \subseteq \mathbb{R}^m$, determined by a *feasible action correspondence* $\mathcal{X}_i : \mathcal{S} \times \mathcal{A}_{-i} \rightrightarrows \mathcal{A}_i$, which takes as input the current state $\boldsymbol{s}^{(t)}$ and the other players' actions $\boldsymbol{a}_{-i}^{(t)} \in \mathcal{A}_{-i}$, and outputs a subset of player $i$'s action space $\mathcal{A}_i$. We define $\mathcal{X}(\boldsymbol{s}, \boldsymbol{a}) \doteq \bigtimes_{i \in [n]} \mathcal{X}_i(\boldsymbol{s}, \boldsymbol{a}_{-i})$.

---

[1]On the one hand, our model generalizes Magill & Quinzii's to a setting with arbitrary not just financial assets; on the other hand, we restrict the transition model to be Markov.

[2]Explanations of notation and definitions of mathematical terminology can be found in Appendix A.

[3]following Littman (1994).

Once the players have taken their actions $\boldsymbol{a}^{(t)} \doteq (\boldsymbol{a}_1^{(t)}, \ldots, \boldsymbol{a}_n^{(t)})$, each player $i \in [n]$ receives a *reward* $r_i(\boldsymbol{s}^{(t)}, \boldsymbol{a}^{(t)})$ given by a *reward function* $\boldsymbol{r} : \mathcal{S} \times \mathcal{A} \to \mathbb{R}^n$, after which the game either ends with probability $1 - \gamma$, where $\gamma \in (0, 1)$ is called the *discount factor*, or continues on to time period $t + 1$, transitioning to a new state $S^{(t+1)} \sim p(\cdot \mid \boldsymbol{s}^{(t)}, \boldsymbol{a}^{(t)})$, according to a *transition* probability function $p : \mathcal{S} \times \mathcal{S} \times \mathcal{A} \to \mathbb{R}_+$, where $p(\boldsymbol{s}^{(t+1)} \mid \boldsymbol{s}^{(t)}, \boldsymbol{a}^{(t)}) \in [0, 1]$ denotes the probability of transitioning to state $\boldsymbol{s}^{(t+1)} \in \mathcal{S}$ from state $\boldsymbol{s}^{(t)} \in \mathcal{S}$ when action profile $\boldsymbol{a}^{(t)} \in \mathcal{A}$ is played.

Our focus is on continuous-state and continuous-action MPGs, where the state and action spaces are non-empty and compact, and the reward functions are continuous and bounded in each of $\boldsymbol{s}$ and $\boldsymbol{a}$.

A *history* $\boldsymbol{h} = ((\boldsymbol{s}^{(t)}, \boldsymbol{a}^{(t)})_{t=0}^{\tau-1}, \boldsymbol{s}^{(\tau)}) \in \mathcal{H}^\tau \doteq (\mathcal{S} \times \mathcal{A})^\tau \times \mathcal{S}$ of length $\tau \in \mathbb{N} \cup \{\infty\}$ is a sequence of states and action profiles. Overloading notation, we define the *history space* $\mathcal{H} \doteq \bigcup_{\tau=0}^\infty \mathcal{H}^\tau$. For any player $i \in [n]$, a *policy* $\boldsymbol{\pi}_i : \mathcal{H} \to \mathcal{A}_i$ is a mapping from histories of any length to $i$'s space of (pure) actions. We define the space of all (deterministic) policies as $\mathcal{P}_i \doteq \{\boldsymbol{\pi}_i : \mathcal{H} \to \mathcal{A}_i\}$. A *Markov policy* (Maskin & Tirole, 2001) $\boldsymbol{\pi}_i$ is a policy s.t. $\boldsymbol{\pi}_i(\boldsymbol{s}^{(\tau)}) = \boldsymbol{\pi}_i(\boldsymbol{h})$, for all histories $\boldsymbol{h} \in \mathcal{H}^\tau$ of length $\tau \in \mathbb{N}_+$, where $\boldsymbol{s}^{(\tau)}$ denotes the final state of history $\boldsymbol{h}$. As Markov policies are only state-contingent, we can compactly represent the space of all Markov policies for player $i \in [n]$ as $\mathcal{P}_i^{\mathrm{markov}} \doteq \{\boldsymbol{\pi}_i : \mathcal{S} \to \mathcal{A}_i\}$.

Fixing player $i \in [n]$ and $\boldsymbol{\pi}_{-i} \in \mathcal{P}_{-i}$, we define the *feasible policy correspondence* $\mathcal{F}_i(\boldsymbol{\pi}_{-i}) \doteq \{\boldsymbol{\pi}_i \in \mathcal{P}_i \mid \forall \boldsymbol{h} \in \mathcal{H}, \boldsymbol{\pi}_i(\boldsymbol{h}) \in \mathcal{X}_i(\boldsymbol{s}^{(\tau)}, \boldsymbol{\pi}_{-i}(\boldsymbol{h}))\}$, given history $\boldsymbol{h} \in \mathcal{H}^\tau$, and the *feasible subclass policy correspondence* $\mathcal{F}_i^{\mathrm{sub}}(\boldsymbol{\pi}_{-i}) \doteq \{\boldsymbol{\pi}_i \in \mathcal{P}_i^{\mathrm{sub}} \mid \forall \boldsymbol{s} \in \mathcal{S}, \boldsymbol{\pi}_i(\boldsymbol{s}) \in \mathcal{X}_i(\boldsymbol{s}, \boldsymbol{\pi}_{-i}(\boldsymbol{s}))\}$, for any $\mathcal{P}^{\mathrm{sub}} \subseteq \mathcal{P}^{\mathrm{markov}}$. Of particular interest is $\mathcal{F}_i^{\mathrm{markov}}(\boldsymbol{\pi}_{-i})$ itself, obtained when $\mathcal{P}^{\mathrm{sub}} = \mathcal{P}^{\mathrm{markov}}$. We also define the feasible policy profile correspondence $\mathcal{F}(\boldsymbol{\pi}) \doteq \bigtimes_{i \in [n]} \mathcal{F}_i(\boldsymbol{\pi}_{-i})$ and the feasible subclass policy profile correspondence $\mathcal{F}^{\mathrm{sub}}(\boldsymbol{\pi}) \doteq \bigtimes_{i \in [n]} \mathcal{F}_i^{\mathrm{sub}}(\boldsymbol{\pi}_{-i})$, for any $\mathcal{P}^{\mathrm{sub}} \subseteq \mathcal{P}^{\mathrm{markov}}$.

Given a policy profile $\boldsymbol{\pi} \in \mathcal{P}$ and a history $\boldsymbol{h} \in \mathcal{H}^\tau$, we denote the *discounted history distributions* $\nu_{\boldsymbol{s}}^{\boldsymbol{\pi},\tau}$ and $\nu_\mu^{\boldsymbol{\pi},\tau}$ that originates at state $\boldsymbol{s}$ and given initial state distribution $\mu$, respectively. Next, we define the set of all realizable trajectories of length $\tau$ under policy $\boldsymbol{\pi}$ as $\mathcal{H}_\mu^{\boldsymbol{\pi},\tau} \doteq \mathrm{supp}(\nu_\mu^{\boldsymbol{\pi},\tau})$. Moreover, given a policy profile $\boldsymbol{\pi} \in \mathcal{P}$, we denote the *state-value function* $\boldsymbol{v}^{\boldsymbol{\pi}} : \mathcal{S} \to \mathbb{R}^n$ as $\boldsymbol{v}^{\boldsymbol{\pi}}(\boldsymbol{s}) \doteq \mathbb{E}_{H \sim \nu_{\boldsymbol{s}}^{\boldsymbol{\pi}}} \left[\sum_{t=0}^\infty \boldsymbol{r}(S^{(t)}, A^{(t)})\right]$ and the *action-value function* $\boldsymbol{q}^{\boldsymbol{\pi}} : \mathcal{S} \times \mathcal{A} \to \mathbb{R}^n$ as $\boldsymbol{q}^{\boldsymbol{\pi}}(\boldsymbol{s}, \boldsymbol{a}) \doteq \boldsymbol{r}(\boldsymbol{s}, \boldsymbol{a}) + \mathbb{E}_{S' \sim p(S' \mid \boldsymbol{s}, \boldsymbol{a})} [\boldsymbol{v}^{\boldsymbol{\pi}}(S')]$. Finally, we define the *discounted state-visitation distribution* $\delta_\mu^{\boldsymbol{\pi}}(\boldsymbol{s}) \doteq \sum_{\tau=0}^\infty \int_{\boldsymbol{h} \in \mathcal{H}_\mu^{\boldsymbol{\pi},\tau}:\boldsymbol{s}^{(\tau)}=\boldsymbol{s}} \nu_\mu^{\boldsymbol{\pi},\tau}(\boldsymbol{h})$ and the *(expected) payoff* of policy profile $\boldsymbol{\pi}$ as $\boldsymbol{u}(\boldsymbol{\pi}) \doteq \mathbb{E}_{S \sim \mu} [\boldsymbol{v}^{\boldsymbol{\pi}}(S)] = \mathbb{E}_{S \sim \delta_\mu^{\boldsymbol{\pi}}} [\boldsymbol{r}(S, \boldsymbol{\pi}(S))]$.

**Solution Concepts and Existence** Having defined our game model, we now define two natural solution concepts, and establish their existence. The first applies the usual notion of Nash equilibrium (1950b) to MPGs. The second is based on the notion of subgame-perfect equilibrium in extensive-form games, a strengthening of Nash equilibrium with the additional requirement that an equilibrium be Nash not just at the start of the game, but at all states encountered during play.

An *$\varepsilon$-generalized Markov perfect equilibrium ($\varepsilon$-GMPE)* $\boldsymbol{\pi}^* \in \mathcal{F}^{\mathrm{markov}}(\boldsymbol{\pi}^*)$ is a Markov policy profile s.t. for all states $\boldsymbol{s} \in \mathcal{S}$ and players $i \in [n]$, $v_i^{\boldsymbol{\pi}^*}(\boldsymbol{s}) \geq \max_{\boldsymbol{\pi}_i \in \mathcal{F}_i(\boldsymbol{\pi}_{-i}^*)} v_i^{(\boldsymbol{\pi}_i, \boldsymbol{\pi}_{-i}^*)}(\boldsymbol{s}) - \varepsilon$. An *$\varepsilon$-generalized Nash equilibrium ($\varepsilon$-GNE)* $\boldsymbol{\pi}^* \in \mathcal{F}(\boldsymbol{\pi}^*)$ is a policy profile s.t. for all states $\boldsymbol{s} \in \mathcal{S}$ and players $i \in [n]$, $u_i(\boldsymbol{\pi}^*) \geq \max_{\boldsymbol{\pi}_i \in \mathcal{F}_i(\boldsymbol{\pi}_{-i}^*)} u_i(\boldsymbol{\pi}_i, \boldsymbol{\pi}_{-i}^*) - \varepsilon$. We call a 0-GMPE (0-GNE) simply a GMPE (GNE). As GMPE is a stronger notion than GNE, every $\varepsilon$-GMPE is an $\varepsilon$-GNE.

To establish existence of GMPE, we introduce two assumptions: the first is the standard convexity and continuity assumption (see Assumption 2 in Appendix D.1); the second, introduced as Condition 1 in Bhandari & Russo (2019), ensures that the policy class under consideration (e.g., $\mathcal{P}^{\mathrm{sub}} \subseteq \mathcal{P}^{\mathrm{markov}}$) is expressive enough to include best responses (see Assumption 3 in Appendix D.1).

**Theorem 2.1.** *If $\mathcal{M}$ is an MPG for which Assumption 2 holds, and $\mathcal{P}^{\mathrm{sub}} \subseteq \mathcal{P}^{\mathrm{markov}}$ is a subspace of Markov policy profiles that satisfies Assumption 3, then there exists a GMPE $\boldsymbol{\pi}^* \in \mathcal{P}^{\mathrm{sub}}$ of $\mathcal{M}$.*

**Equilibrium Computation** Our approach to computing a GMPE in a MPG $\mathcal{M}$ is to minimize a *merit function* associated with $\mathcal{M}$, i.e., a function whose minima coincide with the pseudo-game's GMPE. Our choice of merit function, a common one in game theory, is *exploitability* i.e., the sum of the players' maximal unilateral payoff deviations, denoted $\varphi(\boldsymbol{\pi})$ for policy $\boldsymbol{\pi} \in \mathcal{P}$. Exploitability,

however, is a merit function for GNE, *not* GMPE; *state exploitability* at all states $s \in \mathcal{S}$ is a merit function for GMPE. Nevertheless, as we show in the sequel, for a large class of MPGs, namely those with a bounded best-response mismatch coefficient, the set of Markov policies that minimize exploitability alone equals the set of GMPE, making our approach a sensible one.

We are not out of the woods yet, however, as exploitability is non-convex in general, even in one-shot finite games (Nash, 1950a), GNE computation is PPAD-hard, as NE computation is PPAD-hard (Chen et al., 2009; Daskalakis et al., 2009) and MPGs generalize one-shot games. Accordingly, we instead set our sights on computing a *stationary point* of the exploitability, i.e., a policy profile $\boldsymbol{\pi}^* \in \mathcal{F}^{\mathrm{markov}}(\boldsymbol{\pi}^*)$ s.t. for any other policy $\boldsymbol{\pi} \in \mathcal{F}^{\mathrm{markov}}(\boldsymbol{\pi}^*)$, it holds that $\min_{\boldsymbol{h} \in \mathcal{D}\varphi(\boldsymbol{\pi}^*)} \langle \boldsymbol{h}, \boldsymbol{\pi}^* - \boldsymbol{\pi} \rangle \leq 0$, where $\mathcal{D}\varphi(\boldsymbol{\pi})$ denotes the subdifferential of $\varphi$ at $\boldsymbol{\pi}$. Under suitable assumptions, such a point satisfies the necessary conditions of a GMPE.

In this paper, we study MPGs with possibly continuous state and action spaces. As such, we can only hope to compute an *approximate* stationary point of exploitability in finite time. Defining a notion of approximate stationarity for exploitability is, however, a challenge.

Given an approximation parameter $\varepsilon \geq 0$, a natural definition of an $\varepsilon$-stationary point might be a policy profile $\boldsymbol{\pi}^* \in \mathcal{F}^{\mathrm{markov}}(\boldsymbol{\pi}^*)$ s.t. for any other policy $\boldsymbol{\pi} \in \mathcal{F}^{\mathrm{markov}}(\boldsymbol{\pi}^*)$, it holds that $\min_{\boldsymbol{h} \in \mathcal{D}\varphi(\boldsymbol{\pi}^*)} \langle \boldsymbol{h}, \boldsymbol{\pi}^* - \boldsymbol{\pi} \rangle \leq \varepsilon$. Exploitability is not necessarily Lipschitz-smooth, however, so in general it may not be possible to compute an $\varepsilon$-stationary point in $\mathrm{poly}(1/\varepsilon)$ evaluations of the (sub)gradient of the exploitability.

To address this challenge, a common approach in the optimization literature (see, for instance Definition 19 of Liu et al. (2021)) is to consider an alternative definition known as $(\varepsilon, \delta)$-stationarity. Given $\varepsilon, \delta \geq 0$, an $(\varepsilon, \delta)$-stationary point of exploitability is a policy profile $\boldsymbol{\pi}^* \in \mathcal{F}^{\mathrm{markov}}(\boldsymbol{\pi}^*)$ for which there exists a $\delta$-close policy $\boldsymbol{\pi}^\dagger \in \mathcal{P}$ with $\left\| \boldsymbol{\pi}^\dagger - \boldsymbol{\pi}^* \right\| \leq \delta$ s.t. for any other policy $\boldsymbol{\pi} \in \mathcal{F}^{\mathrm{markov}}(\boldsymbol{\pi}^\dagger)$, it holds that $\min_{\boldsymbol{h} \in \mathcal{D}\varphi(\boldsymbol{\pi}^\dagger)} \langle \boldsymbol{h}, \boldsymbol{\pi}^\dagger - \boldsymbol{\pi} \rangle \leq \varepsilon$. The exploitability minimization method we introduce can indeed compute such an approximate stationary point in polynomial time: i.e., a point in the neighborhood of an approximate stationary point of exploitability. Furthermore, asymptotically, our method is guaranteed to converge to an exact stationary point of exploitability.

**Exploitability Minimization** Given an MPG $\mathcal{M}$ and two policy profiles $\boldsymbol{\pi}, \boldsymbol{\pi}' \in \mathcal{P}$, we define the *state cumulative regret* at state $s \in \mathcal{S}$ as $\psi(s, \boldsymbol{\pi}, \boldsymbol{\pi}') = \sum_{i \in [n]} \left[ v_i^{(\boldsymbol{\pi}_i', \boldsymbol{\pi}_{-i})}(s) - v_i^{\boldsymbol{\pi}}(s) \right]$; the *expected cumulative regret* for an arbitrary state distribution $\upsilon \in \Delta(\mathcal{S})$ as $\psi(\upsilon, \boldsymbol{\pi}, \boldsymbol{\pi}') = \mathbb{E}_{S \sim \upsilon}[\psi(S, \boldsymbol{\pi}, \boldsymbol{\pi}')]$; and the *cumulative regret* as $\Psi(\boldsymbol{\pi}, \boldsymbol{\pi}') = \psi(\mu, \boldsymbol{\pi}, \boldsymbol{\pi}')$. Additionally, for a policy profile $\boldsymbol{\pi}$, we define the *state exploitability* at state $s \in \mathcal{S}$ as $\phi(s, \boldsymbol{\pi}) = \sum_{i \in [n]} \max_{\boldsymbol{\pi}_i' \in \mathcal{F}_i^{\mathrm{markov}}(\boldsymbol{\pi}_{-i})} v_i^{(\boldsymbol{\pi}_i', \boldsymbol{\pi}_{-i})}(s) - v_i^{\boldsymbol{\pi}}(s)$; the *expected state exploitability* for an arbitrary state distribution $\upsilon \in \Delta(\mathcal{S})$ as $\phi(\upsilon, \boldsymbol{\pi}) = \mathbb{E}_{S \sim \upsilon}[\phi(S, \boldsymbol{\pi})]$; and the (global) *exploitability* as $\varphi(\boldsymbol{\pi}) = \sum_{i \in [n]} \max_{\boldsymbol{\pi}_i' \in \mathcal{F}_i^{\mathrm{markov}}(\boldsymbol{\pi}_{-i})} u_i(\boldsymbol{\pi}_i', \boldsymbol{\pi}_{-i})$.

With these definitions in hand, we can reformulate the problem of computing a GMPE as the quasi-minimization problem of minimizing state exploitability, i.e., $\min_{\boldsymbol{\pi} \in \mathcal{F}^{\mathrm{markov}}(\boldsymbol{\pi})} \phi(s, \boldsymbol{\pi})$, at all states $s \in \mathcal{S}$ simultaneously. We can also compute a GNE and exploitability, analogously.

**Lemma 1.** *Given an MPG $\mathcal{M}$, a Markov policy profile $\boldsymbol{\pi}^* \in \mathcal{F}^{\mathrm{markov}}(\boldsymbol{\pi}^*)$ is a GMPE iff $\phi(s, \boldsymbol{\pi}^*) = 0$, for all states $s \in \mathcal{S}$. Similarly, a policy profile $\boldsymbol{\pi}^* \in \mathcal{F}(\boldsymbol{\pi}^*)$ is an GNE iff $\varphi(\boldsymbol{\pi}^*) = 0$.*

This straightforward reformulation of GMPE (resp. GNE) in terms of state exploitability (resp. exploitability) does not immediately lend itself to computation, as exploitability minimization is non-trivial, because exploitability is neither convex nor differentiable in general. Following (Goktas & Greenwald, 2022), we can reformulate these problems yet again, this time as coupled quasi-min-max optimization problems (Wald, 1945). We proceed to do so now; however, we restrict our attention to exploitability, and hence GNE, knowing that we will later show that minimizing exploitability suffices to minimize state exploitability, and thereby find GMPE.

**Observation 1.** *Given an MPG $\mathcal{M}$, $\min_{\boldsymbol{\pi} \in \mathcal{F}(\boldsymbol{\pi})} \varphi(\boldsymbol{\pi}) = \min_{\boldsymbol{\pi} \in \mathcal{F}(\boldsymbol{\pi})} \max_{\boldsymbol{\pi}' \in \mathcal{F}^{\mathrm{markov}}(\boldsymbol{\pi})} \Psi(\boldsymbol{\pi}, \boldsymbol{\pi}')$.*

While the above observation makes progress towards our goal of reformulating exploitability minimization in a tractable manner, the problem remains challenging to solve for two reasons: first, a fixed point computation is required to solve the outer player's minimization problem; second, the

inner player's policy space depends on the choice of outer policy. We overcome these difficulties by choosing suitable policy parameterizations.

**Policy Parameterization**   In a coupled min-max optimization problem, any solution to the inner player's maximization problem is implicitly parameterized by the outer player's decision. We restructure the jointly feasible Markov policy class to represent this dependence explicitly.

Define the class of *dependent policies* $\mathcal{R} \doteq \{\boldsymbol{\rho} : \mathcal{S} \times \mathcal{A} \to \mathcal{A} \mid \forall (\boldsymbol{s}, \boldsymbol{a}) \in \mathcal{S} \times \mathcal{A}, \ \boldsymbol{\rho}(\boldsymbol{s}, \boldsymbol{a}) \in \mathcal{X}(\boldsymbol{s}, \boldsymbol{a})\} = \bigtimes_{i \in [n]} \{\boldsymbol{\rho}_i : \mathcal{S} \times \mathcal{A}_i \to \mathcal{A}_{-i} \mid \forall (\boldsymbol{s}, \boldsymbol{a}_{-i}) \in \mathcal{S} \times \mathcal{A}_{-i}, \ \boldsymbol{\rho}_i(\boldsymbol{s}, \boldsymbol{a}_{-i}) \in \mathcal{X}_i(\boldsymbol{s}, \boldsymbol{a}_{-i})\}$. With this definition in hand we arrive at an *uncoupled* quasi-min-max optimization problem:

**Lemma 2.** *Given an MPG $\mathcal{M}$,*
$\min_{\boldsymbol{\pi} \in \mathcal{F}(\boldsymbol{\pi})} \max_{\boldsymbol{\pi}' \in \mathcal{F}^{\mathrm{markov}}(\boldsymbol{\pi})} \Psi(\boldsymbol{\pi}, \boldsymbol{\pi}') = \min_{\boldsymbol{\pi} \in \mathcal{F}(\boldsymbol{\pi})} \max_{\boldsymbol{\rho} \in \mathcal{R}} \Psi(\boldsymbol{\pi}, \boldsymbol{\rho}(\cdot, \boldsymbol{\pi}(\cdot)))$.

It can be expensive to represent the aforementioned dependence in policies explicitly. This situation can be naturally rectified, however, by a suitable policy parameterization. A suitable policy parameterization can also allow us to represent the set of fixed points $\boldsymbol{\pi} \in \mathcal{P}$ s.t. $\boldsymbol{\pi} \in \mathcal{F}^{\mathrm{markov}}(\boldsymbol{\pi})$ more efficiently in practice (Goktas et al., 2023a). Define a *parameterization scheme* $(\boldsymbol{\pi}, \boldsymbol{\rho}, \mathbb{R}^\Omega, \mathbb{R}^\Sigma)$ as comprising an unconstrained parameter space $\mathbb{R}^\Omega$ and parametric policy profile function $\boldsymbol{\pi} : \mathcal{S} \times \mathbb{R}^\Omega \to \mathcal{A}$ for the outer player, and an unconstrained parameter space $\mathbb{R}^\Sigma$ and parametric policy profile function $\boldsymbol{\rho} : \mathcal{S} \times \mathcal{A} \times \mathbb{R}^\Sigma \to \mathcal{A}$ for the inner player. Given such a scheme, we restrict the players' policies to be parameterized: i.e., the outer player's space of policies $\mathcal{P}^{\mathbb{R}^\Omega} = \{\boldsymbol{\pi} : \mathcal{S} \times \mathbb{R}^\Omega \to \mathcal{A} \mid \boldsymbol{\omega} \in \mathbb{R}^\Omega\} \subseteq \mathcal{P}^{\mathrm{markov}}$, while the inner player's space of policies $\mathcal{R}^{\mathbb{R}^\Sigma} = \{\boldsymbol{\rho} : \mathcal{S} \times \mathcal{A} \times \mathbb{R}^\Sigma \to \mathcal{A} \mid \boldsymbol{\sigma} \in \mathbb{R}^\Sigma\}$.[4]

We further impose following assumptions on the parameterization scheme so that it specifies the two structural properties: (1) it enforces the fixed-point feasibility condition for the outer player's policy, and (2) it explicitly encodes the dependence of the inner player's policy on the outer player's through the dependent-policy architecture (i.e., $\boldsymbol{\rho}$ takes $\boldsymbol{\pi}(\boldsymbol{s})$ as input).[5]

**Assumption 1** (Parameterization for Min-Max Optimization). *Given an MPG $\mathcal{M}$ together with a parameterization scheme $(\boldsymbol{\pi}, \boldsymbol{\rho}, \mathbb{R}^\Omega, \mathbb{R}^\Sigma)$, assume 1. fixing $\boldsymbol{\omega} \in \mathbb{R}^\Omega$, $\boldsymbol{\pi}(\boldsymbol{s}; \boldsymbol{\omega}) \in \mathcal{X}(\boldsymbol{s}, \boldsymbol{\pi}(\boldsymbol{s}; \boldsymbol{\omega}))$, for all $\boldsymbol{s} \in \mathcal{S}$; and 2. fixing $\boldsymbol{\sigma} \in \mathbb{R}^\Sigma$, $\boldsymbol{\rho}(\boldsymbol{s}, \boldsymbol{a}; \boldsymbol{\sigma}) \in \mathcal{X}(\boldsymbol{s}, \boldsymbol{a})$, for all $(\boldsymbol{s}, \boldsymbol{a}) \in \mathcal{S} \times \mathcal{A}$.*

Under such a parameterization, we can express state exploitability minimization as the following min–max optimization problem: $\min_{\boldsymbol{\omega} \in \mathbb{R}^\Omega} \max_{\boldsymbol{\sigma} \in \mathbb{R}^\Sigma} \Psi(\boldsymbol{\omega}, \boldsymbol{\sigma}) \doteq \Psi(\boldsymbol{\pi}(\cdot; \boldsymbol{\omega}), \boldsymbol{\rho}(\cdot, \boldsymbol{\pi}(\cdot; \boldsymbol{\omega}); \boldsymbol{\sigma}))$ .

In summary, by choosing a suitable parameterization scheme, we resolve the two challenges highlighted in Observation 1. First, the unconstrained parameter space facilitates an efficient representation of the outer player's policy space, i.e., the set of fixed points $\{\boldsymbol{\pi} \in \mathcal{P}^{\mathrm{markov}} \mid \boldsymbol{\pi} \in \mathcal{F}^{\mathrm{markov}}(\boldsymbol{\pi})\}$. Second, the parameterization provides an explicit representation of the class of dependent policies, thereby eliminating the dependence of the inner player's policy space on the outer player's policy.

Now, given an unconstrained parameter space, we are able to simplify our definition of $(\varepsilon, \delta)$-*stationary point of exploitability*, namely, a policy parameter $\boldsymbol{\omega}^* \in \mathbb{R}^\Omega$ for which there exists a $\delta$-close policy parameter $\boldsymbol{\omega}^\dagger \in \mathbb{R}^\Omega$ with $\|\boldsymbol{\omega}^* - \boldsymbol{\omega}^\dagger\| \leq \delta$ s.t. $\min_{\boldsymbol{h} \in \mathcal{D}\varphi(\boldsymbol{\omega}^\dagger)} \|\boldsymbol{h}\| \leq \varepsilon$.

**State Exploitability Minimization**   Returning to our objective, namely *state* exploitability minimization, we now turn our attention to obtaining a tractable characterization of this goal. Specifically, we prove two lemmas, which together establish that it suffices to minimize exploitability, rather than state exploitability, as any policy profile that is a stationary point of exploitability is also a stationary point of state exploitability across all states simultaneously, under suitable assumptions.

Our first lemma (Lemma 4, Appendix D.1) states that a stationary point of exploitability is almost surely also a stationary point of state exploitability at all states. Our second lemma (Lemma 5, Appendix D.1) upper bounds gradient of state exploitability in terms of gradient of exploitability,

---

[4]Using these parameterizations, we redefine $\boldsymbol{v}^{\boldsymbol{\omega}} \doteq \boldsymbol{v}^{\boldsymbol{\pi}(\cdot; \boldsymbol{\omega})}$, $\boldsymbol{q}^{\boldsymbol{\omega}} \doteq \boldsymbol{q}^{\boldsymbol{\pi}(\cdot; \boldsymbol{\omega})}$, $\boldsymbol{u}(\boldsymbol{\omega}) = \boldsymbol{u}(\boldsymbol{\pi}(\cdot; \boldsymbol{\omega}))$, and $\nu_\mu^{\boldsymbol{\omega}} = \nu_\mu^{\boldsymbol{\pi}(\cdot; \boldsymbol{\omega})}$; and $\boldsymbol{v}^{\boldsymbol{\sigma}(\boldsymbol{\omega})} \doteq \boldsymbol{v}^{\boldsymbol{\rho}(\cdot, \boldsymbol{\pi}(\cdot; \boldsymbol{\omega}); \boldsymbol{\sigma})}$; $\boldsymbol{q}^{\boldsymbol{\sigma}(\boldsymbol{\omega})} \doteq \boldsymbol{q}^{\boldsymbol{\rho}(\cdot, \boldsymbol{\pi}(\cdot; \boldsymbol{\omega}); \boldsymbol{\sigma})}$; $\boldsymbol{u}(\boldsymbol{\sigma}(\boldsymbol{\omega})) = \boldsymbol{u}(\boldsymbol{\rho}(\cdot, \boldsymbol{\pi}(\cdot; \boldsymbol{\omega}); \boldsymbol{\sigma}))$; $\nu_\mu^{\boldsymbol{\sigma}(\boldsymbol{\omega})} = \nu_\mu^{\boldsymbol{\rho}(\cdot, \boldsymbol{\pi}(\cdot; \boldsymbol{\omega}); \boldsymbol{\sigma})}$; and so on.

[5]In Lemma 3 (Appendix D.1), we showed that such parameterization is guaranteed to exist for a large class of Markov pseudo-games, i.e., any Markov pseudo games with a DAG dependency structure, and this class includes any Exchange Economy Markov Pseudo-Game.

when the best-response mismatch coefficient is bounded. Given $\mathcal{M}$ with initial state distribution $\mu$ and alternative state distribution $\upsilon \in \Delta(\mathcal{S})$, and letting $\Phi_i(\boldsymbol{\pi}_{-i}) \doteq \arg\max_{\boldsymbol{\pi}'_i \in \mathcal{F}_i^{\text{markov}}(\boldsymbol{\pi}_{-i})} u_i(\boldsymbol{\pi}'_i, \boldsymbol{\pi}_{-i})$ denote the set of best response policies for player $i$ when the other players play policy profile $\boldsymbol{\pi}_{-i}$, we define the *best-response mismatch coefficient* for policy profile $\boldsymbol{\pi}$ as $C_{br}(\boldsymbol{\pi}, \mu, \upsilon) \doteq \max_{i \in [n]} \max_{\boldsymbol{\pi}'_i \in \Phi_i(\boldsymbol{\pi}_{-i})} (1/1-\gamma)^2 \|\delta_\upsilon^{(\boldsymbol{\pi}'_i, \boldsymbol{\pi}_{-i})}/\mu\|_\infty \|\delta_\upsilon^{\boldsymbol{\pi}}/\mu\|_\infty$.

**Algorithm and Convergence** Next, we present our algorithm for computing an approximate stationary point of exploitability, and thus state exploitability. The algorithm we use is two time-scale stochastic simultaneous gradient descent-ascent (TTSSGDA)(Appendix D.1, Algorithm 1), first analyzed by (Lin et al., 2020; Daskalakis et al., 2020), which, for polynomial-time convergence, typically requires that the objective be Lipschitz smooth in both decision variables and gradient dominated in the inner one. We thus impose regularity assumptions to ensure these conditions.

In particular, $\Psi(\boldsymbol{\omega}, \boldsymbol{\sigma})$ being Lipschitz smooth in $(\boldsymbol{\omega}, \boldsymbol{\sigma})$ is ensured by the assumption that policy parameterization, reward function, and transition function are all twice continuously differentiable (Assumption 4); while $\Psi(\boldsymbol{\omega}, \boldsymbol{\sigma})$ being gradient dominated in $\boldsymbol{\sigma}$ is ensured by assumption that parametrized policy classes are closed under policy improvement, and each player's action value function is concave in $\boldsymbol{\sigma}$ (Assumption 5).

As the gradient of cumulative regret involves an expectation over histories, we assume that we can simulate trajectories of play $\boldsymbol{h} \sim \nu_\mu^{\boldsymbol{\pi}}$ according to the history distribution $\nu_\mu^{\boldsymbol{\pi}}$, for any policy profile $\boldsymbol{\pi}$, and that doing so provides both value and gradient information for the rewards and transition probabilities along simulated trajectories. That is, we rely on a differentiable game simulator (Suh et al., 2022), meaning a stochastic first-order oracle that returns the gradients of the rewards and transition probabilities, which we query to estimate deviation payoffs and cumulative regrets.

Under this assumption, we estimate these values using realized trajectories from the history distribution $\boldsymbol{h}^{\boldsymbol{\omega}} \sim \nu_\mu^{\boldsymbol{\omega}}$ induced by the outer player's policy, and the deviation history distribution $\boldsymbol{h}^{\boldsymbol{\sigma}} \sim \bigtimes_{i \in [n]} \nu_\mu^{(\sigma_i(\boldsymbol{\omega}_{-i}), \boldsymbol{\omega}_{-i})}$ induced by the inner player's policy.[6]

Our main theorem requires one final definition: the *equilibrium distribution mismatch coefficient* $\|\partial\delta_\mu^{\boldsymbol{\pi}^*}/\partial\mu\|_\infty$, defined as the Radon-Nikodym derivative of the state-visitation distribution of the GNE $\boldsymbol{\pi}^*$ w.r.t. the initial state distribution $\mu$. This coefficient, which measures the inherent difficulty of visiting states under the equilibrium policy $\boldsymbol{\pi}^*$ *without knowing* $\boldsymbol{\pi}^*$ is closely related to other distribution mismatch coefficients used to analyze policy gradient methods (Agarwal et al., 2020).

We now state our main theorem, namely that, under the assumptions outlined above[7], Algorithm 1 computes values for the policy parameters that nearly satisfy the necessary conditions for a GMPE in polynomially many gradient steps, or equivalently, calls to the differentiable simulator.

**Theorem 2.2.** *Given an MPG $\mathcal{M}$ and a parameterization scheme $(\boldsymbol{\pi}, \boldsymbol{\rho}, \mathbb{R}^\Omega, \mathbb{R}^\Sigma)$ satisfying Assumption 1, assume Assumptions 2, 4, and 5 hold. For any $\delta > 0$, set $\varepsilon = \delta\|C_{br}(\cdot, \mu, \cdot)\|_\infty^{-1}$. If Algorithm 1 is run with inputs that satisfy $\eta_{\boldsymbol{\omega}}, \eta_{\boldsymbol{\sigma}} \asymp \text{poly}(\varepsilon, \|\partial\delta_\mu^{\boldsymbol{\pi}^*}/\partial\mu\|_\infty, \frac{1}{1-\gamma}, \ell_{\nabla\Psi}^{-1}, \ell_\Psi^{-1})$, then there exists $T \in \text{poly}\left(\varepsilon^{-1}, (1-\gamma)^{-1}, \|\partial\delta_\mu^{\boldsymbol{\pi}^*}/\partial\mu\|_\infty, \ell_{\nabla\Psi}, \ell_\Psi, \text{diam}(\mathbb{R}^\Omega \times \mathbb{R}^\Sigma), \eta_{\boldsymbol{\omega}}^{-1}\right)$ and $k \leq T$ s.t. $\boldsymbol{\omega}_{\text{best}}^{(T)} = \boldsymbol{\omega}^{(k)}$ is an $(\varepsilon, \varepsilon/2\ell_\Psi)$-stationary point of exploitability, i.e., there exists $\boldsymbol{\omega}^* \in \mathbb{R}^\Omega$ s.t. $\left\|\boldsymbol{\omega}_{\text{best}}^{(T)} - \boldsymbol{\omega}^*\right\| \leq \varepsilon/2\ell_\Psi$ and $\min_{\boldsymbol{h} \in \mathcal{D}_\varphi(\boldsymbol{\omega}^*)} \|\boldsymbol{h}\| \leq \varepsilon$. And, for any distribution $\upsilon \in \Delta(\mathcal{S})$, if $\phi(\upsilon, \cdot)$*

---

[6]More specifically, for all policies $\boldsymbol{\pi} \in \mathcal{P}^{\text{markov}}$ and histories $\boldsymbol{h} \in \mathcal{H}^\tau$, the *payoff estimator* for player $i \in [n]$ is given by $\widehat{u}_i(\boldsymbol{\pi}; \boldsymbol{h}) \doteq \sum_{t=0}^{\tau-1} \mu(\boldsymbol{s}^{(0)}) r_i(\boldsymbol{s}^{(t)}, \boldsymbol{\pi}'(\boldsymbol{s}^{(t)})) \prod_{k=0}^{t-1} \gamma^k p(\boldsymbol{s}^{(k+1)} \mid \boldsymbol{s}^{(k)}, (\boldsymbol{s}^{(k)}))$. Moreover, for all $\boldsymbol{\omega} \in \mathbb{R}^\Omega$, $\boldsymbol{\sigma} \in \mathbb{R}^\Sigma$, $\boldsymbol{h}^{\boldsymbol{\omega}} \sim \nu_\mu^{\boldsymbol{\omega}}$, and $\boldsymbol{h}^{\boldsymbol{\sigma}} = (\boldsymbol{h}_1^{\boldsymbol{\sigma}}, \cdots, \boldsymbol{h}_n^{\boldsymbol{\sigma}}) \sim \bigtimes_{i \in [n]} \nu_\mu^{(\sigma_i(\boldsymbol{\omega}_{-i}), \boldsymbol{\omega}_i)}$, the *cumulative regret estimator* is given by $\widehat{\Psi}(\boldsymbol{\omega}, \boldsymbol{\sigma}; \boldsymbol{h}^{\boldsymbol{\omega}}, \boldsymbol{h}^{\boldsymbol{\sigma}}) \doteq \sum_{i \in [n]} \widehat{u}_i(\boldsymbol{\rho}_i(\cdot, \boldsymbol{\pi}_{-i}(\cdot; \boldsymbol{\omega}); \boldsymbol{\sigma}), \boldsymbol{\pi}_{-i}(\cdot, \boldsymbol{\omega}); \boldsymbol{h}_i^{\boldsymbol{\sigma}}) - \widehat{u}_i(\boldsymbol{\pi}(\cdot; \boldsymbol{\omega}); \boldsymbol{h}^{\boldsymbol{\omega}})$, while the *cumulative regret gradient estimator* is given by $\widehat{G}(\boldsymbol{\omega}, \boldsymbol{\sigma}; \boldsymbol{h}^{\boldsymbol{\omega}}, \boldsymbol{h}^{\boldsymbol{\sigma}}) \doteq (\nabla_{\boldsymbol{\omega}} \widehat{\Psi}(\boldsymbol{\omega}, \boldsymbol{\sigma}; \boldsymbol{h}^{\boldsymbol{\omega}}, \boldsymbol{h}^{\boldsymbol{\sigma}}), \nabla_{\boldsymbol{\sigma}} \widehat{\Psi}(\boldsymbol{\omega}, \boldsymbol{\sigma}; \boldsymbol{h}^{\boldsymbol{\omega}}, \boldsymbol{h}^{\boldsymbol{\sigma}}))$.

[7]The Lipschitz smoothness and bounded mismatch assumptions are imposed to obtain polynomial-time convergence guarantees and are standard in modern analyses of policy-gradient methods. They should be viewed as learning-theoretic stability conditions rather than economic restrictions; empirically, our method performs well even in settings with non-smooth primitives (e.g., Leontief utilities), where neural parameterizations provide effective smooth approximations and practical exploration reduces state-distribution mismatch.

*is differentiable at $\boldsymbol{\omega}^*$, then* $\left\|\nabla_{\boldsymbol{\omega}}\varphi(\upsilon,\boldsymbol{\omega}^*)\right\| \leq \delta$, *i.e.,* $\boldsymbol{\omega}_{\text{best}}^{(T)}$ *is an* $(\varepsilon,\delta)$-*stationary point of expected state exploitability* $\phi(\upsilon,\cdot)$.

In other words, by running Algorithm 1 on $\mathcal{M}$, we compute a policy profile $\boldsymbol{\omega}_{\text{best}}^{(T)}$ in the neighborhood of $\boldsymbol{\omega}^*$, an approximate stationary point of exploitability. By Lemma 5, $\boldsymbol{\omega}^*$ is also an approximate stationary point of state exploitability at all states in $\mathcal{M}$ simultaneously, and therefore approximately satisfies the necessary conditions of a GMPE. Therefore, Algorithm 1 converges to a point $\boldsymbol{\omega}_{\text{best}}^{(T)}$ in the neighborhood of a point $\boldsymbol{\omega}^*$ that approximately satisfies the necessary conditions of an GMPE. While arguably a relatively weak theoretical conclusion, our experiments demonstrate that in practice our method succeeds at approximating GMPEs in exchange economy MPGs. Moreover, in the limit, Algorithm 1 converges to a point that exactly satisfies the necessary conditions of an GMPE.

## 3 INCOMPLETE MARKOV ECONOMIES

Having developed a mathematical formalism for MPGs, along with a proof of existence of GMPE as well as an algorithm that computes them, we now move on to our main agenda, namely modeling incomplete stochastic economies in this formalism. We establish the first proof, to our knowledge, of the existence of recursive competitive equilibria in standard incomplete stochastic economies, and we provide a polynomial-time algorithm for approximating them.

**Infinite Horizon Markov Exchange Economies**   Classic Arrow–Debreu economies provide the static foundation for our framework; full details appear in Appendix C.2. Building on these static models, we introduce our infinite-horizon model.

An *infinite horizon Markov exchange economy (MEE)* $\mathcal{I} \doteq (n, m, l, d, \mathcal{S}, \mathcal{X}, \mathcal{Y}, \mathcal{E}, \mathcal{T}, \boldsymbol{r}, \gamma, p, \mathcal{R}, \mu)$, comprises $n \in \mathbb{N}$ consumers who, over an infinite discrete time horizon $t = 0, 1, 2, \ldots$, repeatedly encounter the opportunity to buy a consumption of $m \in \mathbb{N}$ commodities from a space $\mathcal{X}_i \subset \mathbb{R}^m$ of consumptions and a portfolio of $l \in \mathbb{N}$ assets from a space $\mathcal{Y}_i \subset \mathbb{R}^l$ of asset portfolios (or investments), where $y_{ik} \geq 0$ denotes units of asset $k$ bought (long) by consumer $i$, while $y_{ik} < 0$ denotes units that are sold (short). Assets are assumed to be *short-lived* (Magill & Quinzii, 1994), meaning that any asset purchased at time $t$ pays its dividends in the subsequent time period $t+1$, and then expires. The consumers' collective decisions lead them through the *state space* $\mathcal{S} \doteq \mathcal{O} \times (\mathcal{E} \times \mathcal{T})$, which comprises a *world state space* $\mathcal{O}$ and a *spot market space* $\mathcal{E} \times \mathcal{T}$, the product of the consumers' endowment and type spaces. The spot market space is a collection of *spot markets*, each one a static exchange market $(\boldsymbol{E}, \boldsymbol{\Theta}) \in \mathcal{E} \times \mathcal{T} \subseteq \mathbb{R}^m \times \mathbb{R}^d$.

Each *asset* $k \in [l]$ is a *generalized Arrow security*, i.e., a divisible contract that transfers to its owner a quantity of the $j$th commodity at any world state $o \in \mathcal{O}$ determined by a matrix of asset returns $\boldsymbol{R}_o \doteq \left(\boldsymbol{r}_{o1}, \ldots, \boldsymbol{r}_{ol}\right)^T \in \mathbb{R}^{l \times m}$ s.t. $r_{okj} \in \mathbb{R}$ denotes the quantity of commodity $j$ transferred at world state $o$ for one unit of asset $k$. The collection of asset returns across all world states is given by $\mathcal{R} \doteq \{\boldsymbol{R}_o\}_{o \in \mathcal{O}}$. Assets allow consumers to insure themselves against future realizations of the spot market (i.e., endowments and types), by allowing them to transfer wealth across world states.

The economy starts at time period $t = 0$ in an *initial state* $S^{(0)} \sim \mu$ determined by an initial state distribution $\mu \in \Delta(\mathcal{S})$. At each time step $t = 0, 1, 2, \ldots$, the state of the economy is $\boldsymbol{s}^{(t)} \doteq (o^{(t)}, \boldsymbol{E}^{(t)}, \boldsymbol{\Theta}^{(t)}) \in \mathcal{S}$. Each consumer $i \in [n]$, observes the world state $o^{(t)} \in \mathcal{O}$, and participates in a spot market $(\boldsymbol{E}^{(t)}, \boldsymbol{\Theta}^{(t)})$, where it purchases *a consumption* $\boldsymbol{x}_i^{(t)} \in \mathcal{X}_i$ at *commodity prices* $\boldsymbol{p}^{(t)} \in \Delta_m$, and an *asset market* $\mathcal{Y} = \bigtimes_{i \in [n]} \mathcal{Y}_i$ by which it invests in an *asset portfolio* $\boldsymbol{y}_i^{(t)} \in \mathcal{Y}_i$ at *assets prices* $\boldsymbol{q}^{(t)} \in \mathbb{R}^l$. Every consumer is constrained to buy a consumption $\boldsymbol{x}_i^{(t)} \in \mathcal{X}_i$ and invest in an asset portfolio $\boldsymbol{y}_i^{(t)} \in \mathcal{Y}_i$ with a total cost weakly less than the value of its current endowment $\boldsymbol{e}_i^{(t)} \in \mathcal{E}_i$. Formally, the set of consumptions and investment portfolios that a consumer $i$ can afford with its current endowment $\boldsymbol{e}_i^{(t)} \in \mathcal{E}_i$ at current commodity prices $\boldsymbol{p}^{(t)} \in \Delta_m$ and current asset prices $\boldsymbol{q}^{(t)} \in \mathbb{R}^l$, i.e., its *budget set* $\mathcal{B}_i(\boldsymbol{e}_i^{(t)}, \boldsymbol{p}^{(t)}, \boldsymbol{q}^{(t)})$, is determined by its *budget correspondence* $\mathcal{B}_i(\boldsymbol{e}_i, \boldsymbol{p}, \boldsymbol{q}) \doteq \{(\boldsymbol{x}_i, \boldsymbol{y}_i) \in \mathcal{X}_i \times \mathcal{Y}_i \mid \boldsymbol{x}_i \cdot \boldsymbol{p} + \boldsymbol{y}_i \cdot \boldsymbol{q} \leq \boldsymbol{e}_i \cdot \boldsymbol{p}\}$.

After the consumers make their consumption and investment decisions, they each receive *reward* $r_i(\boldsymbol{x}_i^{(t)}; \boldsymbol{\theta}_i^{(t)})$ as a function of their consumption and type, and then the economy either collapses with probability $1 - \gamma$, or survives to see another day with probability $\gamma$, where $\gamma \in (0, 1)$ is called

the *discount rate*. If the economy survives, then a new state is realized, namely $(O', E', \Theta') \sim p(\cdot \mid \boldsymbol{s}^{(t)}, \boldsymbol{Y}^{(t)})$, according to a *transition probability function* $p : \mathcal{S} \times \mathcal{S} \times \mathcal{Y} \to [0, 1]$ that depends on the consumers' investment portfolio profile $\boldsymbol{Y}^{(t)} \doteq (\boldsymbol{y}_1^{(t)}, \dots, \boldsymbol{y}_n^{(t)})^T \in \mathcal{Y}$, after which the economy transitions to a new state $S^{(t+1)} \doteq (O', E' + \boldsymbol{Y}^{(t)} \boldsymbol{R}_{O'}, \Theta')$, where the consumers' new endowments depends on their returns $\boldsymbol{Y}^{(t)} \boldsymbol{R}_{O'} \in \mathbb{R}^{n \times m}$ on their investments.

A *history* $\boldsymbol{h} = ((\boldsymbol{s}^{(t)}, \boldsymbol{X}^{(t)}, \boldsymbol{Y}^{(t)}, \boldsymbol{p}^{(t)}, \boldsymbol{q}^{(t)})_{t=0}^{\tau-1}, \boldsymbol{s}^{(\tau)}) \in \mathcal{H}^\tau$ is a sequence of tuples comprising states, consumption profiles, investment profiles, commodity price, and asset prices. Overloading notation, we define the *history space* $\mathcal{H} \doteq \bigcup_{\tau=0}^\infty \mathcal{H}^\tau$, and then *consumption*, *investment*, *commodity price* and *asset price policies* as mappings $\boldsymbol{x}_i : \mathcal{H} \to \mathcal{X}_i$, $\boldsymbol{y}_i : \mathcal{H} \to \mathcal{Y}_i$, $\boldsymbol{p} : \mathcal{H} \to \Delta_m$, and $\boldsymbol{q} : \mathcal{H} \to \mathbb{R}^l$ from histories to consumptions, investments, commodity prices, and asset prices, respectively, s.t. $(\boldsymbol{x}_i, \boldsymbol{y}_i)(\boldsymbol{h})$ is the consumption-investment decision of consumer $i \in [n]$, and $(\boldsymbol{p}, \boldsymbol{q})(\boldsymbol{h})$ are commodity and asset prices, both at history $\boldsymbol{h} \in \mathcal{H}$. A *consumption policy profile* (resp. *investment policy profile*) $\boldsymbol{X}(\boldsymbol{h}) \doteq (\boldsymbol{x}_1, \dots, \boldsymbol{x}_n)(\boldsymbol{h})^T$ (resp. $\boldsymbol{Y}(\boldsymbol{h}) \doteq (\boldsymbol{y}_1, \dots, \boldsymbol{y}_n)(\boldsymbol{h})^T$) is a collection of consumption (resp. investment) policies for all consumers. A consumption policy $\boldsymbol{x}_i : \mathcal{S} \to \mathcal{X}_i$ is *Markov* if it depends only on the last state of the history, i.e., $\boldsymbol{x}_i(\boldsymbol{h}) = \boldsymbol{x}_i(\boldsymbol{s}^{(\tau)})$, for all histories $\boldsymbol{h} \in \mathcal{H}^\tau$ of all lengths $\tau \in \mathbb{N}$. An analogous definition extends to investment, commodity price, and asset price policies.

Given $\boldsymbol{\pi} \doteq (\boldsymbol{X}, \boldsymbol{Y}, \boldsymbol{p}, \boldsymbol{q})$ and a history $\boldsymbol{h} \in \mathcal{H}^\tau$, we denote the *discounted history distribution* assuming initial state distribution $\mu$ by $\nu_\mu^{\boldsymbol{\pi}, \tau}$. Overloading notation, we define the set of all realizable trajectories $\mathcal{H}^{\boldsymbol{\pi}, \tau}$ of length $\tau$ under policy profile $\boldsymbol{\pi}$ as $\mathcal{H}^{\boldsymbol{\pi}, \tau} \doteq \operatorname{supp}(\nu_\mu^{\boldsymbol{\pi}, \tau})$.

**Solution Concepts and Existence** An *outcome* $(\boldsymbol{X}, \boldsymbol{Y}, \boldsymbol{p}, \boldsymbol{q}) : \mathcal{H} \to \mathcal{X} \times \mathcal{Y} \times \Delta_m \times \mathbb{R}^l$ of an infinite horizon MEE is a tuple consisting of a commodity prices policy, an asset prices policy, a consumption policy profile, and an investment policy profile. An outcome is *Markov* if all its constituent policies are Markov.

While Radner equilibrium is the canonical solution concept for finite-horizon stochastic economies, its history dependence becomes infinite-dimensional in infinite-horizon settings, rendering equilibria analytically and computationally intractable. Following standard practice in macroeconomics (e.g., Stokey et al. (1989); Prescott & Mehra (1980)), we restrict our attention to Markov outcomes.

Given a Markov consumption and investment profile $(\boldsymbol{X}, \boldsymbol{Y})$, the *consumption state-value function* $v_i^{(\boldsymbol{X}, \boldsymbol{Y}, \boldsymbol{p}, \boldsymbol{q})} : \mathcal{S} \to \mathbb{R}$ for consumer $i$ is defined as $v_i^{(\boldsymbol{X}, \boldsymbol{Y}, \boldsymbol{p}, \boldsymbol{q})}(\boldsymbol{s}) \doteq \mathbb{E}_{H \sim \nu_{\boldsymbol{s}}^{(\boldsymbol{X}, \boldsymbol{Y}, \boldsymbol{p}, \boldsymbol{q})}} \left[ \sum_{t=0}^\infty \gamma^t r_i \left( \boldsymbol{x}_i(\boldsymbol{s}^{(t)}); \Theta^{(t)} \right) \right]$. A Markov outcome $(\boldsymbol{X}^*, \boldsymbol{Y}^*, \boldsymbol{p}^*, \boldsymbol{q}^*)$ is *perfect* for $i$ if $i$ maximizes its consumption state-value function over all affordable consumption and investment policies, i.e., $(\boldsymbol{x}_i^*, \boldsymbol{y}_i^*) \in \arg\max_{\substack{(\boldsymbol{x}_i, \boldsymbol{y}_i): \mathcal{S} \to \mathcal{X}_i \times \mathcal{Y}_i : \forall \boldsymbol{s} \in \mathcal{S}, \\ (\boldsymbol{x}_i, \boldsymbol{y}_i)(\boldsymbol{s}) \in \mathcal{B}_i(\boldsymbol{e}_i, \boldsymbol{p}^*(\boldsymbol{s}), \boldsymbol{q}^*(\boldsymbol{s}))}} \left\{ v_i^{(\boldsymbol{x}_i, \boldsymbol{x}_{-i}^*, \boldsymbol{y}_i, \boldsymbol{y}_{-i}^*, \boldsymbol{p}^*, \boldsymbol{q}^*)}(\boldsymbol{s}) \right\} \forall \boldsymbol{s} \in \mathcal{S}$.

A Markov consumption policy $\boldsymbol{X}$ is said to be *feasible* iff for all time horizons $\tau \in \mathbb{N}$ and state $\boldsymbol{s}^{(\tau)}$, $\sum_{i \in [n]} \boldsymbol{x}_i(\boldsymbol{s}^{(\tau)}) - \sum_{i \in [n]} \boldsymbol{e}_i^{(\tau)} \leq \boldsymbol{0}_m$, where $\boldsymbol{e}_i^{(\tau)} \in \mathcal{E}_i$ is consumer $i$'s endowment at state $\boldsymbol{s}^{(\tau)}$. Similarly, an investment policy is *feasible* if $\sum_{i \in [n]} \boldsymbol{y}_i(\boldsymbol{s}^{(\tau)}) \leq \boldsymbol{0}_l$. If all the consumption and investment policies associated with an outcome are feasible, we say the outcome is *feasible* as well.

A Markov outcome $(\boldsymbol{X}, \boldsymbol{Y}, \boldsymbol{p}, \boldsymbol{q})$ is said to satisfy *Walras' law* iff for all time horizons $\tau \in \mathbb{N}$ and state $\boldsymbol{s}^{(\tau)} \in \mathcal{S}$, $\boldsymbol{p}(\boldsymbol{s}^{(t)}) \cdot \left( \sum_{i \in [n]} \boldsymbol{x}_i(\boldsymbol{s}^{(t)}) - \sum_{i \in [n]} \boldsymbol{e}_i^{(\tau)} \right) + \boldsymbol{q}(\boldsymbol{s}^{(\tau)}) \cdot \left( \sum_{i \in [n]} \boldsymbol{y}_i(\boldsymbol{s}^{(\tau)}) \right) = 0$, where, as above, $\boldsymbol{e}_i^{(\tau)} \in \mathcal{E}_i$ is consumer $i$'s endowment at state $\boldsymbol{s}^{(\tau)}$.

A refinement of Radner equilibrium in the infinite horizon setting is recursive Radner equilibrium.

**Definition 1** (Recursive Radner Equilibrium). *A recursive Radner (or Walrasian or competitive) equilibrium (RRE) (Mehra & Prescott, 1977; Prescott & Mehra, 1980) of an infinite horizon MEE $\mathcal{I}$ is a Markov outcome $(\boldsymbol{X}^*, \boldsymbol{Y}^*, \boldsymbol{p}^*, \boldsymbol{q}^*)$ that is 1. Markov perfect, for all consumers $i \in [n]$; 2. feasible; and 3. satisfies Walras' law.*

We establish the existence of RRE in infinite horizon MEEs under standard economic assumptions: convexity, boundedness, no-satiation (see, for example, Geanakoplos (1990)). To do so, we first associate an *exchange economy MPG* $\mathcal{M}$ with a given infinite horizon MEE $\mathcal{I}$.

**Definition 2** (Exchange Economy Markov Pseudo-Game). *Let $\mathcal{I}$ be an infinite horizon MEE. The corresponding exchange economy MPG $\mathcal{M} = (n + 1, m + l, \mathcal{S}, \bigtimes_{i \in [n]} (\mathcal{X}_i \times \mathcal{Y}_i) \times (\mathcal{P} \times \mathcal{Q}), \mathcal{B}', r', p', \gamma', \mu')$ is defined as*

- *The $n + 1$ players comprise $n$ consumers, players $1, \ldots, n$, and one auctioneer, player $n + 1$.*
- *The set of states $\mathcal{S} = \mathcal{O} \times \mathcal{E} \times \mathcal{T}$. At each state $\boldsymbol{s} = (o, \boldsymbol{E}, \boldsymbol{\Theta}) \in \mathcal{S}$,*
  - *each consumer $i \in [n]$ chooses an action $\boldsymbol{a}_i = (\boldsymbol{x}_i, \boldsymbol{y}_i) \in \mathcal{B}'_i(\boldsymbol{s}, \boldsymbol{a}_{-i}) \subseteq \mathcal{X}_i \times \mathcal{Y}_i$ from a set of feasible actions $\mathcal{B}'_i(\boldsymbol{s}, \boldsymbol{a}_{-i}) = \mathcal{B}_i(\boldsymbol{e}_i, \boldsymbol{a}_{n+1}) \cap \{(\boldsymbol{x}_i, \boldsymbol{y}_i) \mid \sum_{i \in [n]} \boldsymbol{x}_i \leq \sum_{i \in [n]} \boldsymbol{e}_i, \sum_{i \in [n]} \boldsymbol{y}_i \leq \boldsymbol{0}_m, (\boldsymbol{X}, \boldsymbol{Y}) \in \mathcal{X} \times \mathcal{Y}\}$ and receives reward $r'_i(\boldsymbol{s}, \boldsymbol{a}) \doteq r_i(\boldsymbol{x}_i; \boldsymbol{\theta}_i)$;*
  - *the auctioneer $n + 1$ chooses an action $\boldsymbol{a}_{n+1} = (\boldsymbol{p}, \boldsymbol{q}) \in \mathcal{B}'_{n+1}(\boldsymbol{s}, \boldsymbol{a}_{-(n+1)}) \doteq \mathcal{P} \times \mathcal{Q}$ where $\mathcal{P} \doteq \Delta_m$ and $\mathcal{Q} = [0, \max_{\boldsymbol{E} \in \mathcal{E}} \sum_{i \in [n]} \sum_{j \in [m]} e_{ij}]^l$, and receives reward $r'_{n+1}(\boldsymbol{s}, \boldsymbol{a}) \doteq \boldsymbol{p} \cdot \left(\sum_{i \in [n]} \boldsymbol{x}_i - \sum_{i \in [n]} \boldsymbol{e}_i\right) + \boldsymbol{q} \cdot \left(\sum_{i \in [n]} \boldsymbol{y}_i\right)$.*
- *The transition probability function is defined as $p'(\boldsymbol{s}' \mid \boldsymbol{s}, \boldsymbol{a}) \doteq p(\boldsymbol{s}' \mid \boldsymbol{s}, \boldsymbol{Y})$.*
- *The discount rate $\gamma' = \gamma$ and the initial state distribution $\mu' = \mu$.*

Our existence proof reformulates the set of RRE of any infinite horizon MEE as the set of GMPE of the exchange economy MPG.

**Theorem 3.1.** *Consider an infinite horizon MEE $\mathcal{I}$. Under Assumption 6, the set of RRE of $\mathcal{I}$ is equal to the set of GMPE of the associated exchange economy MPG $\mathcal{M}$.*

We can define the exploitability (resp. expected state exploitability) of any infinite horizon MEE $\mathcal{I}$ satisfying Assumption 6 as the exploitability (resp. expected state exploitability) of its associated exchange economy MPG $\mathcal{M}$. By Theorem 3.1, an outcome that minimizes the exploitability of the economy $\mathcal{I}$ is a Radner equilibrium (RE), while an outcome that minimizes state exploitability at all states simultaneously is an RRE. The following corollary now follows from Theorem 2.1.

**Corollary 1.** *Under Assumption 6, the set of RRE of an infinite horizon MEE is non-empty.*

**Equilibrium Computation**    Since an RRE is infinite-dimensional when the state space is continuous, its computation is FNP-hard (Murty & Kabadi, 1987). As such, it is generally believed that the best we can hope to find in polynomial time is a solution that approximately satisfies the necessary conditions of an RRE. Since the set of RRE of any infinite horizon MEE $\mathcal{I}$ is equal to the set of GMPE of the associated exchange economy MPG $\mathcal{M}$ (Theorem 3.1), we can apply Theorem 2.2 to compute a policy profile with the following computational complexity guarantees for Algorithm 1, when run on the exchange economy MPG associated with an infinite horizon MEE.

**Theorem 3.2.** *Given an infinite horizon MEE $\mathcal{I}$ for which Assumption 6 holds, and the associated exchange economy MPG $\mathcal{M}$. If $(\boldsymbol{\pi}, \boldsymbol{\rho}, \mathbb{R}^{\Omega}, \mathbb{R}^{\Sigma})$ is a parametrization scheme for $\mathcal{M}$ such that Assumptions 4 and 5 hold, then the convergence results in Theorem 2.2 hold, meaning Algorithm 1 converges to a point in the neighborhood of a point that approximately satisfies the necessary conditions of a GMPE in $\mathcal{M}$, which is likewise a point that approximately satisfies the necessary conditions of an RRE of $\mathcal{I}$. Moreover, beyond its finite-time guarantees, in the limit, Algorithm 1 converges to a point that satisfies these conditions exactly.*

## 4 EXPERIMENTS

Given an infinite horizon MEE $\mathcal{I}$, we associate with it an exchange economy MPG $\mathcal{M}$, and we then construct a deep learning network to solve $\mathcal{M}$. To do so, we assume a parametrization scheme $(\boldsymbol{\pi}, \boldsymbol{\rho}, \mathbb{R}^{\Omega}, \mathbb{R}^{\Sigma})$, where the parametric policy profiles $(\boldsymbol{\pi}, \boldsymbol{\rho})$ are represented by neural networks with $(\mathbb{R}^{\Omega}, \mathbb{R}^{\Sigma})$ as the corresponding network parameter spaces. Computing an RRE via Algorithm 1 can then be seen as the result of training a generative adversarial neural network (Goodfellow et al., 2014; Goktas et al., 2023a), where $\boldsymbol{\pi}$ (resp. $\boldsymbol{\rho}$) is the output of the generator (resp. adversarial) network. We call such a neural representation a *generative adversarial policy network (GAPNet)*.

**Convergence in Base Economies**    Following our proposed approach, we built a GAPNet to approximate an RRE in two types of infinite-horizon MEEs: the first with deterministic transitions, and the second with stochastic. In each, we experimented with three randomly sampled economies

with 10 consumers, 10 commodities, 1 asset, 5 world states. The consumers were characterized by three distinct classes of reward functions, linear, Cobb-Douglas, and Leontief, each of which imparts different smoothness properties on their state-value functions.[8]

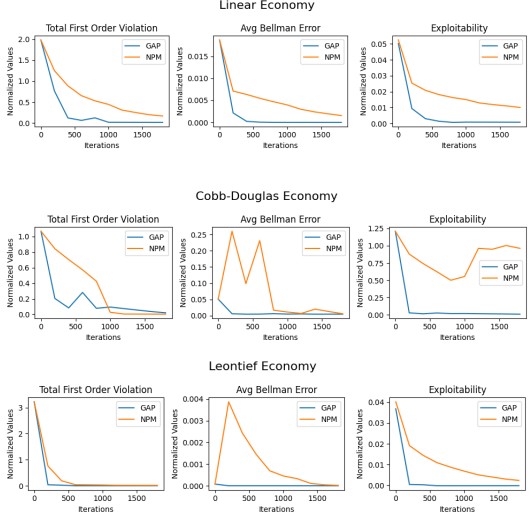

Figure 1: Economies with Stochastic Transitions.

We compare our GAPNet results with a classic neural projection method (NPM), also known as deep equilibrium nets (Azinovic et al., 2022), which macroeconomists and others use to solve stochastic economies. This method seeks a policy profile that minimizes the norm of the system of first-order necessary and sufficient conditions that characterize RRE (e.g., Fernández-Villaverde (2023)). We use the same network architecture for both methods, and select hyperparameters for each through grid search. In all experiments, we evaluate the performance of the ensuing policy profiles using three metrics: total first-order violations, average Bellman errors,[9] and exploitability.

Experiments in economies with deterministic transitions are presented in Section E. While NPM performs well under metrics it is designed to minimize, GAPNet performs well under all three metrics, highlighting its ability to approximate the necessary conditions of an RRE. Moreover, stochasticity further hinders NPM's performance (see Figure 1), while GAPNet successfully minimizes all three metrics across all economies.

**Economic Interpretation of Learned Equilibria** Beyond measuring convergence, it is also necessary to verify that the learned policies exhibit economically meaningful behavior. We thus analyze equilibrium consumption and investment patterns across different preference specifications and discount factors. The results illustrate that GAPNet recovers classical economic predictions across all settings. We include the results of these experiments in Appendix E.

*Utility specifications* Under linear preferences, agents exhibit almost no asset demand and allocate roughly 97–98% of wealth to current consumption. Under Cobb-Douglas preferences, consumption shares fall to 88–90% and agents take positive asset positions, as Cobb-Douglas utilities are strictly quasi-concave and exhibit diminishing marginal utility in our experiments, motivating positive intertemporal smoothing. Under Leontief preferences, spending rises to nearly 99% of wealth and asset demand approaches zero. This is because Leontief utility is limited by the scarcest good, any reduction in present consumption is extremely costly. Agents therefore avoid intertemporal substitution and concentrate spending on bottleneck goods, leading to low investment. These patterns are consistent with well-known predictions from standard consumer theory and dynamic consumption-savings models under different curvature assumptions(Mas-Colell et al., 1995; Deaton, 1991; Stokey et al., 1989).

*Discount-rate effects.* We also vary the discount factor $\gamma$ in the Cobb-Douglas economy. As expected, impatient agents (small discount rate $\gamma$) devote nearly all wealth to current consumption, while more patient agents invest more heavily and smooth consumption intertemporally.

**Scalability to Larger Economies** To assess the scalability of our approach, we extend our experiments to substantially larger exchange economies: 20 consumers, 20 commodities, 5 assets, and 10 world states. This setting increases both the dimensionality of the joint action space and the complexity of the endogenous state transitions, making equilibrium computation significantly more challenging. Due to the higher variance introduced by larger economies, we observe slightly increased instability and greater sensitivity to learning-rate choices. Nevertheless, our extended experiments consistently show clear convergence toward near-zero exploitability (Figure 5, Appendix E), demonstrating that our approach remains effective even as the scale of the economy grows.

---

[8]Our code can be found here.

[9]The definitions of these two metrics can be found in Section E.1.

