ACKNOWLEDGMENTS

This research was supported by the Office of Navel Research under grant N00014-24-1-2657 and the National Science Foundation under grant IIS2147187. Denizalp Goktas was supported by a JP Morgan Chase Fellowship (Award No. 008493). The authors also wish to thank anonymous reviewers for their helpful feedback.

REFERENCES

Daron Acemoglu. *Introduction to modern economic growth*. Princeton university press, 2008. 20

Yves Achdou, Jiequn Han, Jean-Michel Lasry, Pierre-Louis Lions, and Benjamin Moll. Income and wealth distribution in macroeconomics: A continuous-time approach. *The review of economic studies*, 89(1):45–86, 2022. 20

Alekh Agarwal, Sham M Kakade, Jason D Lee, and Gaurav Mahajan. Optimality and approximation with policy gradient methods in markov decision processes. In *Conference on Learning Theory*, pp. 64–66. PMLR, 2020. 6

S Rao Aiyagari. Uninsured idiosyncratic risk and aggregate saving. *The Quarterly Journal of Economics*, 109(3):659–684, 1994. 20

Kenneth Arrow and Gerard Debreu. Existence of an equilibrium for a competitive economy. *Econometrica: Journal of the Econometric Society*, pp. 265–290, 1954a. 2, 22

Kenneth J Arrow. An extension of the basic theorems of classical welfare economics. In *Proceedings of the second Berkeley symposium on mathematical statistics and probability*, volume 2, pp. 507–533. University of California Press, 1951. 22

Kenneth J Arrow. Le role des valeurs boursieres pour la repartition la meilleure des risques, econometrie, 41-47, english translation as the role of securities in the optimal allocation of risk-bearing. *Review of Economic Studies*, 31:91–96, 1964. 22, 23

Kenneth J. Arrow and Gerard Debreu. Existence of an Equilibrium for a Competitive Economy. *Econometrica*, 22(3):265–290, 1954b. ISSN 0012-9682. doi: 10.2307/1907353. URL https://www.jstor.org/stable/1907353. Publisher: [Wiley, Econometric Society]. 1

Alp E Atakan. Stochastic convexity in dynamic programming. *Economic Theory*, 22:447–455, 2003a. 35

Alp E. Atakan. Stochastic convexity in dynamic programming. *Economic Theory*, 22(2):447–455, 2003b. ISSN 09382259, 14320479. URL http://www.jstor.org/stable/25055693. 18

Adrien Auclert, Bence Bardóczy, Matthew Rognlie, and Ludwig Straub. Using the sequence-space jacobian to solve and estimate heterogeneous-agent models. *Econometrica*, 89(5):2375–2408, 2021. 19, 21

Marlon Azinovic, Luca Gaegauf, and Simon Scheidegger. Deep equilibrium nets. *International Economic Review*, 63(4):1471–1525, 2022. 10, 36

Xiaohui Bei, Jugal Garg, and Martin Hoefer. Tatonnement for linear and gross substitutes markets. *CoRR abs/1507.04925*, 2015. 21

Truman Bewley. A difficulty with the optimum quantity of money. *Econometrica: Journal of the Econometric Society*, pp. 1485–1504, 1983. 20

Jalaj Bhandari and Daniel Russo. Global optimality guarantees for policy gradient methods. *arXiv preprint arXiv:1906.01786*, 2019. 3, 32

Fischer Black and Myron Scholes. The pricing of options and corporate liabilities. *Journal of political economy*, 81(3):637–654, 1973. 23

Olivier Jean Blanchard and Charles M Kahn. The solution of linear difference models under rational expectations. *Econometrica: Journal of the Econometric Society*, pp. 1305–1311, 1980. 21

Mathieu Blondel, Quentin Berthet, Marco Cuturi, Roy Frostig, Stephan Hoyer, Felipe Llinares-López, Fabian Pedregosa, and Jean-Philippe Vert. Efficient and modular implicit differentiation. *arXiv preprint arXiv:2105.15183*, 2021. 40

James Bradbury, Roy Frostig, Peter Hawkins, Matthew James Johnson, Chris Leary, Dougal Maclaurin, George Necula, Adam Paszke, Jake VanderPlas, Skye Wanderman-Milne, and Qiao Zhang. JAX: composable transformations of Python+NumPy programs, 2018. URL http://github.com/google/jax. 40

Simina Brânzei, Nikhil Devanur, and Yuval Rabani. Proportional dynamics in exchange economies. In *Proceedings of the 22nd ACM Conference on Economics and Computation*, pp. 180–201, 2021. 21

David Cass. *Competitive equilibrium with incomplete financial markets*. University of Pennsylvania, Center for Analytic Research in Economics and . . . , 1984. 19

David Cass. *On the" number" of equilibrium allocations with incomplete financial markets*. University of Pennsylvania, Center for Analytic Research in Economics and . . . , 1985. 19

Xi Chen and Xiaotie Deng. Settling the complexity of two-player nash equilibrium. In *2006 47th Annual IEEE Symposium on Foundations of Computer Science (FOCS'06)*, pp. 261–272. IEEE, 2006. 20

Xi Chen, Xiaotie Deng, and Shang-Hua Teng. Settling the complexity of computing two-player nash equilibria. *Journal of the ACM (JACM)*, 56(3):1–57, 2009. 4

David Childers, Jesús Fernández-Villaverde, Jesse Perla, Christopher Rackauckas, and Peifan Wu. Differentiable state-space models and hamiltonian monte carlo estimation. Technical report, National Bureau of Economic Research, 2022. 21

Lawrence J Christiano, Martin S Eichenbaum, and Mathias Trabandt. On dsge models. *Journal of Economic Perspectives*, 32(3):113–140, 2018. 20

Richard Clarida, Jordi Gali, and Mark Gertler. Monetary policy rules and macroeconomic stability: evidence and some theory. *The Quarterly journal of economics*, 115(1):147–180, 2000. 20

James Cloyne, Clodomiro Ferreira, and Paolo Surico. Monetary policy when households have debt: new evidence on the transmission mechanism. *The Review of Economic Studies*, 87(1):102–129, 2020. 20

Bruno Codenotti, Benton McCune, and Kasturi Varadarajan. Market equilibrium via the excess demand function. In *Proceedings of the thirty-seventh annual ACM symposium on Theory of computing*, pp. 74–83, 2005. 21

Bruno Codenotti, Amin Saberi, Kasturi Varadarajan, and Yinyu Ye. Leontief economies encode nonzero sum two-player games. In *SODA*, volume 6, pp. 659–667, 2006. 20

John C Cox and Stephen A Ross. The valuation of options for alternative stochastic processes. *Journal of financial economics*, 3(1-2):145–166, 1976. 19

John C Cox, Stephen A Ross, and Mark Rubinstein. Option pricing: A simplified approach. *Journal of financial Economics*, 7(3):229–263, 1979. 19

John C Cox, Jonathan E Ingersoll Jr, and Stephen A Ross. An intertemporal general equilibrium model of asset prices. *Econometrica: Journal of the Econometric Society*, pp. 363–384, 1985. 19

Michael Curry, Alexander Trott, Soham Phade, Yu Bai, and Stephan Zheng. Learning solutions in large economic networks using deep multi-agent reinforcement learning. 21

Constantinos Daskalakis, Paul W Goldberg, and Christos H Papadimitriou. The complexity of computing a nash equilibrium. *SIAM Journal on Computing*, 39(1):195–259, 2009. 4, 20

Constantinos Daskalakis, Dylan J Foster, and Noah Golowich. Independent policy gradient methods for competitive reinforcement learning. *Advances in neural information processing systems*, 33: 5527–5540, 2020. 2, 6, 32

Constantinos Daskalakis, Dylan J. Foster, and Noah Golowich. Independent Policy Gradient Methods for Competitive Reinforcement Learning, January 2021. URL http://arxiv.org/abs/2101.04233. arXiv:2101.04233 [cs]. 25

Damek Davis, Dmitriy Drusvyatskiy, Kellie J MacPhee, and Courtney Paquette. Subgradient methods for sharp weakly convex functions. *Journal of Optimization Theory and Applications*, 179:962–982, 2018. 32

Angus Deaton. Saving and liquidity constraints. *Econometrica*, 59(5):1221–1248, 1991. doi: 10.2307/2938366. 10

Gerard Debreu. The coefficient of resource utilization. *Econometrica*, 19(3):273–292, 1951. ISSN 00129682, 14680262. URL http://www.jstor.org/stable/1906814. 22

Gerard Debreu et al. Representation of a preference ordering by a numerical function. *Decision processes*, 3:159–165, 1954. 24

Xiaotie Deng and Ye Du. The computation of approximate competitive equilibrium is ppad-hard. *Information Processing Letters*, 108(6):369–373, 2008. 20

N. R. Devanur, C. H. Papadimitriou, A. Saberi, and V. V. Vazirani. Market equilibrium via a primal-dual-type algorithm. In *The 43rd Annual IEEE Symposium on Foundations of Computer Science, 2002. Proceedings.*, pp. 389–395, 2002. doi: 10.1109/SFCS.2002.1181963. 20

Peter Diamond. Income taxation with fixed hours of work. *Journal of Public Economics*, 13(1): 101–110, 1980. 19

Peter A. Diamond. The role of a stock market in a general equilibrium model with technological uncertainty. *The American Economic Review*, 57(4):759–776, 1967. ISSN 00028282. URL http://www.jstor.org/stable/1815367. 19, 23

Steven Diamond and Stephen Boyd. CVXPY: A Python-embedded modeling language for convex optimization. *Journal of Machine Learning Research*, 17(83):1–5, 2016. 40

Jacques H Dreze. Investment under private ownership: optimality, equilibrium and stability. In *Allocation under Uncertainty: Equilibrium and Optimality: Proceedings from a Workshop sponsored by the International Economic Association*, pp. 129–166. Springer, 1974. 23

Darrell Duffie. Stochastic equilibria with incomplete financial markets. *Journal of Economic Theory*, 41(2):405–416, 1987. 19

Darrell Duffie and Wayne Shafer. Equilibrium in incomplete markets: I: A basic model of generic existence. *Journal of Mathematical Economics*, 14(3):285–300, 1985. 19

Darrell Duffie and Wayne Shafer. Equilibrium in incomplete markets: Ii: Generic existence in stochastic economies. *Journal of Mathematical Economics*, 15(3):199–216, 1986. 19

James Durbin and Siem Jan Koopman. *Time series analysis by state space methods*, volume 38. OUP Oxford, 2012. 21

Maria Eskelinen. Monetary policy, agent heterogeneity and inequality: Insights from a three-agent new keynesian model. 2021. 20

Ky Fan. Fixed-point and minimax theorems in locally convex topological linear spaces. *Proceedings of the National Academy of Sciences of the United States of America*, 38(2):121–126, 1952. ISSN 0027-8424. 25, 26

J. Fernández-Villaverde, J.F. Rubio-Ramírez, and F. Schorfheide. Chapter 9 - solution and estimation methods for dsge models. volume 2 of *Handbook of Macroeconomics*, pp. 527–724. Elsevier, 2016. doi: https://doi.org/10.1016/bs.hesmac.2016.03.006. URL https://www.sciencedirect.com/science/article/pii/S1574004816000070. 19, 21

Jesús Fernández-Villaverde. *Computational Methods for Macroeconomics*. University of Pennsylvania, 2023. URL https://www.sas.upenn.edu/~jesusfv/teaching.html. Lecture notes for one-year course on computational methods for economists. 1, 10, 24

Anthony V Fiacco and Jerzy Kyparisis. Convexity and concavity properties of the optimal value function in parametric nonlinear programming. *Journal of optimization theory and applications*, 48(1):95–126, 1986. 26

Arlington M Fink. Equilibrium in a stochastic $n$-person game. *Journal of science of the hiroshima university, series ai (mathematics)*, 28(1):89–93, 1964. 2

Yuan Gao and Christian Kroer. First-order methods for large-scale market equilibrium computation. In Hugo Larochelle, Marc'Aurelio Ranzato, Raia Hadsell, Maria-Florina Balcan, and Hsuan-Tien Lin (eds.), *Advances in Neural Information Processing Systems 33: Annual Conference on Neural Information Processing Systems 2020, NeurIPS 2020, December 6-12, 2020, virtual*, 2020. URL https://proceedings.neurips.cc/paper/2020/hash/f75526659f31040afeb61cb7133e4e6d-Abstract.html. 20

Rahul Garg and Sanjiv Kapoor. Auction algorithms for market equilibrium. In *Proceedings of the thirty-sixth annual ACM symposium on Theory of computing*, pp. 511–518, 2004. 21

John Geanakoplos. An introduction to general equilibrium with incomplete asset markets. *Journal of mathematical economics*, 19(1-2):1–38, 1990. 8, 19, 22, 23

John D Geanakoplos and Herakles M Polemarchakis. Walrasian indeterminacy and keynesian macroeconomics. *The Review of Economic Studies*, 53(5):755–779, 1986. 19

Denizalp Goktas and Amy Greenwald. Exploitability minimization in games and beyond. In *Advances in Neural Information Processing Systems*, 2022. 4

Denizalp Goktas, David C Parkes, Ian Gemp, Luke Marris, Georgios Piliouras, Romuald Elie, Guy Lever, and Andrea Tacchetti. Generative adversarial equilibrium solvers. *arXiv preprint arXiv:2302.06607*, 2023a. 5, 9

Denizalp Goktas, Jiayi Zhao, and Amy Greenwald. Tâtonnement in homothetic Fisher markets. *arXiv preprint arXiv:2306.04890*, 2023b. 20, 21

Ian Goodfellow, Jean Pouget-Abadie, Mehdi Mirza, Bing Xu, David Warde-Farley, Sherjil Ozair, Aaron Courville, and Yoshua Bengio. Generative adversarial nets. In Z. Ghahramani, M. Welling, C. Cortes, N. Lawrence, and K.Q. Weinberger (eds.), *Advances in Neural Information Processing Systems*, volume 27. Curran Associates, Inc., 2014. URL https://proceedings.neurips.cc/paper/2014/file/5ca3e9b122f61f8f06494c97b1afccf3-Paper.pdf. 9

Bruce C Greenwald and Joseph E Stiglitz. Externalities in economies with imperfect information and incomplete markets. *The quarterly journal of economics*, 101(2):229–264, 1986. 19

Jiequn Han, Yucheng Yang, et al. Deepham: A global solution method for heterogeneous agent models with aggregate shocks. *arXiv preprint arXiv:2112.14377*, 2021. 20, 21

Charles R. Harris, K. Jarrod Millman, Stéfan J van der Walt, Ralf Gommers, Pauli Virtanen, David Cournapeau, Eric Wieser, Julian Taylor, Sebastian Berg, Nathaniel J. Smith, Robert Kern, Matti Picus, Stephan Hoyer, Marten H. van Kerkwijk, Matthew Brett, Allan Haldane, Jaime Fernandez del Rio, Mark Wiebe, Pearu Peterson, Pierre Gérard-Marchant, Kevin Sheppard, Tyler Reddy, Warren Weckesser, Hameer Abbasi, Christoph Gohlke, and Travis E. Oliphant. Array programming with NumPy. *Nature*, 585:357–362, 2020. doi: 10.1038/s41586-020-2649-2. 40

Oliver D Hart. On the optimality of equilibrium when the market structure is incomplete. *Journal of economic theory*, 11(3):418–443, 1975. 23

Tom Hennigan, Trevor Cai, Tamara Norman, Lena Martens, and Igor Babuschkin. Haiku: Sonnet for JAX, 2020. URL http://github.com/deepmind/dm-haiku. 40

Kevin XD Huang and Jan Werner. Asset price bubbles in arrow-debreu and sequential equilibrium. *Economic Theory*, 15:253–278, 2000. 1, 24

Gur Huberman. A simple approach to arbitrage pricing theory. *Journal of Economic Theory*, 28(1): 183–191, 1982. 19

Mark Huggett. The risk-free rate in heterogeneous-agent incomplete-insurance economies. *Journal of economic Dynamics and Control*, 17(5-6):953–969, 1993. 20

J. D. Hunter. Matplotlib: A 2d graphics environment. *Computing in Science and Engineering*, 9(3): 90–95, 2007. doi: 10.1109/MCSE.2007.55. 40

Kamal Jain, Vijay V Vazirani, and Yinyu Ye. Market equilibria for homothetic, quasi-concave utilities and economies of scale in production. In *SODA*, volume 5, pp. 63–71, 2005. 20, 21

Kenneth L Judd. Projection methods for solving aggregate growth models. *Journal of Economic Theory*, 58(2):410–452, December 1992. ISSN 0022-0531. doi: 10. 1016/0022-0531(92)90061-L. URL https://www.sciencedirect.com/science/article/pii/002205319290061L. 36

Jinill Kim and Sunghyun Henry Kim. Spurious welfare reversals in international business cycle models. *journal of International Economics*, 60(2):471–500, 2003. 21

Robert G King, Charles I Plosser, and Sergio T Rebelo. Production, growth and business cycles: Technical appendix. *Computational Economics*, 20:87–116, 2002. 21

Finn E Kydland and Edward C Prescott. Time to build and aggregate fluctuations. *Econometrica: Journal of the Econometric Society*, pp. 1345–1370, 1982. 19, 20, 21

Tianyi Lin, Chi Jin, and Michael Jordan. On gradient descent ascent for nonconvex-concave minimax problems. In *International Conference on Machine Learning*, pp. 6083–6093. PMLR, 2020. 2, 6, 33

John Lintner. The valuation of risk assets and the selection of risky investments in stock portfolios and capital budgets. In *Stochastic optimization models in finance*, pp. 131–155. Elsevier, 1975. 23

Michael L. Littman. Markov games as a framework for multi-agent reinforcement learning. In *Proceedings of the Eleventh International Conference on International Conference on Machine Learning*, ICML'94, pp. 157–163, San Francisco, CA, USA, 1994. Morgan Kaufmann Publishers Inc. ISBN 1558603352. 2

Mingrui Liu, Hassan Rafique, Qihang Lin, and Tianbao Yang. First-order Convergence Theory for Weakly-Convex-Weakly-Concave Min-max Problems, July 2021. URL http://arxiv.org/abs/1810.10207. arXiv:1810.10207. 4

John B Long Jr and Charles I Plosser. Real business cycles. *Journal of political Economy*, 91(1): 39–69, 1983. 20

Lee Hsien Loong and Richard Zeckhauser. Pecuniary externalities do matter when contingent claims markets are incomplete. *The Quarterly Journal of Economics*, 97(1):171–179, 1982. 19

Robert E Lucas Jr. Asset prices in an exchange economy. *Econometrica: journal of the Econometric Society*, pp. 1429–1445, 1978. 19

Robert E Lucas Jr and Edward C Prescott. Investment under uncertainty. *Econometrica: Journal of the Econometric Society*, pp. 659–681, 1971. 19

Michael Magill and Martine Quinzii. Infinite horizon incomplete markets. *Econometrica: Journal of the Econometric Society*, pp. 853–880, 1994. 1, 2, 7, 24

Michael Magill and Martine Quinzii. *Theory of incomplete markets*, volume 1. Mit press, 2002. 19, 23

Michael Magill and Wayne Shafer. Incomplete markets. *Handbook of mathematical economics*, 4: 1523–1614, 1991. 23

Lilia Maliar, Serguei Maliar, John Taylor, and Inna Tsener. A tractable framework for analyzing a class of nonstationary markov models. Technical report, National Bureau of Economic Research, 2015. 21

Lilia Maliar, Serguei Maliar, and Pablo Winant. Deep learning for solving dynamic economic models. *Journal of Monetary Economics*, 122:76–101, September 2021. ISSN 0304-3932. doi: 10.1016/j.jmoneco.2021.07.004. URL https://www.sciencedirect.com/science/article/pii/S0304393221000799. 36

Andreu Mas-Colell, Michael D. Whinston, and Jerry R. Green. *Microeconomic Theory*. Number 9780195102680 in OUP Catalogue. Oxford University Press, 1995. ISBN ARRAY(0x4cf9c5c0). URL https://ideas.repec.org/b/oxp/obooks/9780195102680.html. 10, 23

Eric Maskin and Jean Tirole. Markov perfect equilibrium: I. observable actions. *Journal of Economic Theory*, 100(2):191–219, 2001. 3, 18

R Mehra and EC Prescott. Recursive competitive equilibria and capital asset pricing. *Essays in Financial Economics*, 1977. 2, 8

Franco Modigliani and Merton H Miller. The cost of capital, corporation finance and the theory of investment. *The American economic review*, 48(3):261–297, 1958. 23

Jan Mossin. Equilibrium in a capital asset market. *Econometrica: Journal of the econometric society*, pp. 768–783, 1966. 23

Katta G Murty and Santosh N Kabadi. Some np-complete problems in quadratic and nonlinear programming. *Mathematical Programming*, 39:117–129, 1987. 9

John F. Nash. The bargaining problem. *Econometrica*, 18(2):155–162, 1950a. ISSN 00129682, 14680262. URL http://www.jstor.org/stable/1907266. 4

John F. Nash. Equilibrium points in n-person games. *Proceedings of the National Academy of Sciences*, 36(1):48–49, 1950b. doi: 10.1073/pnas.36.1.48. URL https://www.pnas.org/doi/abs/10.1073/pnas.36.1.48. 3

David MG Newbery and Joseph E Stiglitz. *The theory of commodity price stabilization. A study in the economics of risk.* 1982. 19

Juan Carlos Parra-Alvarez. *Solution Methods and Inference in Continuous-time Dynamic Equilibrium Economies:(with Applications in Asset Pricing and Income Fluctuation Models): a PhD Thesis Submitted to School of Business and Social Sciences, Aarhus University, in Partial Fulfilment of the Requirements of the PhD Degree in Economics and Business*. Department of Economics and Business, Aarhus University, 2015. 21

Edward C. Prescott and Rajnish Mehra. Recursive competitive equilibrium: The case of homogeneous households. *Econometrica*, 48(6):1365–1379, 1980. ISSN 00129682, 14680262. URL http://www.jstor.org/stable/1912812. 2, 8

Roy Radner. Competitive equilibrium under uncertainty. *Econometrica: Journal of the Econometric Society*, pp. 31–58, 1968. 23

Roy Radner. Existence of equilibrium of plans, prices, and price expectations in a sequence of markets. *Econometrica: Journal of the Econometric Society*, pp. 289–303, 1972. 1, 23

Roy Radner. Rational expectations equilibrium: Generic existence and the information revealed by prices. *Econometrica: Journal of the Econometric Society*, pp. 655–678, 1979. 23

Stephen A Ross. Options and efficiency. *The Quarterly Journal of Economics*, 90(1):75–89, 1976. 19

Thomas J Sargent and Lars Ljungqvist. Recursive macroeconomic theory. *Massachusetss Institute of Technology*, 2000. 1, 21, 24

Maxime Sauzet. Projection Methods via Neural Networks for Continuous-Time Models, December 2021. URL https://papers.ssrn.com/abstract=3981838. 36

M. J. P. Selby. *Economica*, 57(227):413–415, 1990. ISSN 00130427, 14680335. URL http://www.jstor.org/stable/2554945. 19

Lloyd S Shapley. Stochastic games. *Proceedings of the national academy of sciences*, 39(10): 1095–1100, 1953. 2

David Silver, Guy Lever, Nicolas Heess, Thomas Degris, Daan Wierstra, and Martin Riedmiller. Deterministic policy gradient algorithms. In *International conference on machine learning*, pp. 387–395. Pmlr, 2014. 31

Joseph E Stiglitz. Self-selection and pareto efficient taxation. *Journal of public economics*, 17(2): 213–240, 1982. 19

Nancy L Stokey. *Recursive methods in economic dynamics*. Harvard University Press, 1989. 21

Nancy L. Stokey, Robert E. Lucas, and Edward C. Prescott. *Recursive Methods in Economic Dynamics*. Harvard University Press, 1989. ISBN 978-0-674-75096-8. doi: 10.2307/j.ctvjnrt76. URL https://www.jstor.org/stable/j.ctvjnrt76. 8, 10

Hyung Ju Suh, Max Simchowitz, Kaiqing Zhang, and Russ Tedrake. Do differentiable simulators give better policy gradients? In *International Conference on Machine Learning*, pp. 20668–20696. PMLR, 2022. 6

Masayuki Takahashi. Equilibrium points of stochastic non-cooperative $n$-person games. *Journal of Science of the Hiroshima University, Series AI (Mathematics)*, 28(1):95–99, 1964. 2

John B Taylor and Michael Woodford. *Handbook of macroeconomics*, volume 1. Elsevier, 1999. 1, 24

Guido Van Rossum and Fred L Drake Jr. *Python tutorial*. Centrum voor Wiskunde en Informatica Amsterdam, The Netherlands, 1995. 40

Abraham Wald. Statistical decision functions which minimize the maximum risk. *Annals of Mathematics*, pp. 265–280, 1945. 4

Leon Walras. *Elements de l'economie politique pure, ou, Theorie de la richesse sociale*. F. Rouge, 1896. 1, 21

Jan Werner. Equilibrium in economies with incomplete financial markets. *Journal of Economic Theory*, 36(1):110–119, 1985. 19

## LLM USAGE DISCLOSURE

In accordance with ICLR guidelines on the responsible use of large language models (LLMs), we note that LLMs were used exclusively for refining language and improving formatting. They were not used to generate research ideas, mathematical content, theoretical results, or experimental findings. The authors are solely responsible for the accuracy and integrity of all scientific contributions in this work.

## A PRELIMINARIES AND FULL DEFINITIONS

### A.1 PRELIMINARIES

**Notation.** We use caligraphic uppercase letters to denote sets (e.g., $\mathcal{X}$), bold uppercase letters to denote matrices (e.g., $\boldsymbol{X}$), bold lowercase letters to denote vectors (e.g., $\boldsymbol{p}$), lowercase letters to denote scalar quantities (e.g., $x$), and uppercase letters to denote random variables (e.g., $X$). We denote the $i$th row vector of a matrix (e.g., $\boldsymbol{X}$) by the corresponding bold lowercase letter with subscript $i$ (e.g., $\boldsymbol{x}_i$). Similarly, we denote the $j$th entry of a vector (e.g., $\boldsymbol{p}$ or $\boldsymbol{x}_i$) by the corresponding lowercase letter with subscript $j$ (e.g., $p_j$ or $x_{ij}$). We denote functions by a letter determined by the value of the function, e.g., $f$ if the mapping is scalar valued, $\boldsymbol{f}$ if the mapping is vector valued, and $\mathcal{F}$ if the mapping is set valued.

We denote the set $\{1, \ldots, n\}$ by $[n]$, the set $\{0, 1, \ldots, n\}$ by $[n^*]$, the set of natural numbers by $\mathbb{N}$, and the set of real numbers by $\mathbb{R}$. We denote the positive and strictly positive elements of a set using a + or ++ subscript, respectively, e.g., $\mathbb{R}_+$ and $\mathbb{R}_{++}$.

For any $n \in \mathbb{N}$, we denote the $n$-dimensional vector of zeros and ones by $\mathbf{0}_n$ and $\mathbf{1}_n$, respectively. We let $\Delta_n = \{\boldsymbol{x} \in \mathbb{R}_+^n \mid \sum_{i=1}^n x_i = 1\}$ denote the unit simplex in $\mathbb{R}^n$, and $\Delta(A)$ denote the set of all probability distributions over a given set $A$. We also define the support of a probability density function $f \in \Delta(\mathcal{X})$ as $\mathrm{supp}(f) \doteq \{\boldsymbol{x} \in \mathcal{X} : f(\boldsymbol{x}) > 0\}$. Finally, we denote the orthogonal projection operator onto a set $C$ by $\Pi_C$, i.e., $\Pi_C(\boldsymbol{x}) \doteq \arg\min_{\boldsymbol{y} \in C} \|\boldsymbol{x} - \boldsymbol{y}\|^2$.

We define the subdifferential of a function $f : \mathcal{X} \times \mathcal{Y} \to \mathbb{R}$ w.r.t. variable $\boldsymbol{x}$ at a point $(\boldsymbol{a}, \boldsymbol{b}) \in \mathcal{X} \times \mathcal{Y}$ by $\mathcal{D}_{\boldsymbol{x}} f(\boldsymbol{a}, \boldsymbol{b}) \doteq \{\boldsymbol{h} \mid f(\boldsymbol{x}, \boldsymbol{b}) \geq f(\boldsymbol{a}, \boldsymbol{b}) + \boldsymbol{h}^T(\boldsymbol{x} - \boldsymbol{a})\}$, and we denote the derivative operator (resp. partial derivative operator w.r.t. $\boldsymbol{x}$) of a function $\boldsymbol{g} : \mathcal{X} \times \mathcal{Y} \to \mathcal{Z}$ by $\partial \boldsymbol{g}$ (resp. $\partial_{\boldsymbol{x}} \boldsymbol{g}$).

**Terminology.** Fix any norm $\|\cdot\|$. Given $\mathcal{A} \subset \mathbb{R}^d$, the function $f : \mathcal{A} \to \mathbb{R}$ is said to be $\ell_f$-*Lipschitz-continuous* iff $\forall \boldsymbol{x}_1, \boldsymbol{x}_2 \in \mathcal{X}, \|f(\boldsymbol{x}_1) - f(\boldsymbol{x}_2)\| \leq \ell_f \|\boldsymbol{x}_1 - \boldsymbol{x}_2\|$. If the gradient of $f$ is $\ell_{\nabla f}$-Lipschitz-continuous, $f$ is called $\ell_{\nabla f}$-*Lipschitz-smooth*.

We require notions of stochastic convexity related to stochastic dominance of probability measures (Atakan, 2003b). Given non-empty and convex parameter and outcome spaces $\mathcal{W}$ and $\mathcal{O}$ respectively, a conditional probability distribution $\boldsymbol{w} \mapsto p(\cdot \mid \boldsymbol{w}) \in \Delta(\mathcal{O})$ is said to be *stochastically convex* (resp. *stochastically concave*) in $\boldsymbol{w} \in \mathcal{W}$ if for all continuous, bounded, and convex (resp. concave) functions $v : \mathcal{O} \to \mathbb{R}$, $\lambda \in (0, 1)$, and $\boldsymbol{w}', \boldsymbol{w}^\dagger \in \mathcal{W}$ s.t. $\bar{\boldsymbol{w}} = \lambda \boldsymbol{w}' + (1 - \lambda) \boldsymbol{w}^\dagger$, it holds that $\mathbb{E}_{O \sim p(\cdot | \bar{\boldsymbol{w}})}[v(O)] \leq$ (resp. $\geq$) $\lambda \mathbb{E}_{O \sim p(\cdot | \boldsymbol{w}')}[v(O)] + (1 - \lambda) \mathbb{E}_{O \sim p(\cdot | \boldsymbol{w}^\dagger)}[v(O)]$.

For $\mathcal{X} \subseteq \mathbb{R}^d$, we say $f$ is $(c, \mu)$–*gradient dominated* over $\mathcal{X}$ if there exist constants $c > 0$ and $\mu \geq 0$ such that $\min_{x' \in \mathcal{X}} f(x') \geq f(x) + \min_{x' \in \mathcal{X}} \left[ c\langle \nabla f(x), x' - x \rangle + \mu/2 \|x - x'\|_2^2 \right], \forall x \in \mathcal{X}$. The function is said to be gradient dominated with degree one if $\mu = 0$ and gradient dominated with degree two if $\mu > 0$.

## A.2 Omitted Formal Definitions from Section 2

A *history* $\boldsymbol{h} \in \mathcal{H}^\tau \doteq (\mathcal{S} \times \mathcal{A})^\tau \times \mathcal{S}$ of length $\tau \in \mathbb{N}$ is a sequence of states and action profiles $\boldsymbol{h} = ((\boldsymbol{s}^{(t)}, \boldsymbol{a}^{(t)})_{t=0}^{\tau-1}, \boldsymbol{s}^{(\tau)})$ s.t. a history of length 0 corresponds only to the initial state of the game. For any history $\boldsymbol{h} = ((\boldsymbol{s}^{(t)}, \boldsymbol{a}^{(t)})_{t=0}^{\tau-1}, \boldsymbol{s}^{(\tau)})$ of length $\tau \in \mathbb{N}$, we denote by $\boldsymbol{h}_{:\tau'}$ the first $\tau' \in [\tau^*]$ steps of $\boldsymbol{h}$, i.e., $\boldsymbol{h}_{:\tau'} = ((\boldsymbol{s}^{(t)}, \boldsymbol{a}^{(t)})_{t=0}^{\tau'-1}, \boldsymbol{s}^{(\tau')})$. Overloading notation, we define the *history space* $\mathcal{H} \doteq \bigcup_{\tau=0}^\infty \mathcal{H}^\tau$. For any player $i \in [n]$, a *policy* $\boldsymbol{\pi}_i : \mathcal{H} \to \mathcal{A}_i$ is a mapping from histories of any length to $i$'s space of (pure) actions. We define the space of all (deterministic) policies as $\mathcal{P}_i \doteq \{\boldsymbol{\pi}_i : \mathcal{H} \to \mathcal{A}_i\}$. A *Markov policy* (Maskin & Tirole, 2001) $\boldsymbol{\pi}_i$ is a policy s.t. $\boldsymbol{\pi}_i(\boldsymbol{s}^{(\tau)}) = \boldsymbol{\pi}_i(\boldsymbol{h}_{:\tau})$, for all histories $\boldsymbol{h} \in \mathcal{H}^\tau$ of length $\tau \in \mathbb{N}_+$, where $\boldsymbol{s}^{(\tau)}$ denotes the final state of history $\boldsymbol{h}$. As Markov policies are only state-contingent, we can compactly represent the space of all Markov policies for player $i \in [n]$ as $\mathcal{P}_i^{\mathrm{markov}} \doteq \{\boldsymbol{\pi}_i : \mathcal{S} \to \mathcal{A}_i\}$.

Given a policy profile $\boldsymbol{\pi} \in \mathcal{P}$ and a history $\boldsymbol{h} \in \mathcal{H}^\tau$, the *discounted history distribution that originates at state* $\boldsymbol{s}$ is defined as $\nu_{\boldsymbol{s}}^{\boldsymbol{\pi}, \tau}(\boldsymbol{h}) = \mathbb{1}_{\boldsymbol{s}}(\boldsymbol{s}^{(0)}) \prod_{t=0}^{\tau-1} \gamma^t p(\boldsymbol{s}^{(t+1)} \mid \boldsymbol{s}^{(t)}, \boldsymbol{a}^{(t)}) \mathbb{1}_{\{\boldsymbol{\pi}(\boldsymbol{h}_{:t})\}}(\boldsymbol{a}^{(t)})$. Furthermore, the *discounted history distribution given initial state distribution* $\mu$ is defined as $\nu_\mu^{\boldsymbol{\pi}, \tau}(\boldsymbol{h}) = \mathbb{E}_{S \sim \mu}\left[\nu_S^{\boldsymbol{\pi}, \tau}(\boldsymbol{h})\right]$. Next, we define the set of all realizable trajectories of length $\tau$ under policy $\boldsymbol{\pi}$ as $\mathcal{H}_\mu^{\boldsymbol{\pi}, \tau} \doteq \mathrm{supp}(\nu_\mu^{\boldsymbol{\pi}, \tau})$, i.e., the set of all histories that occur with non-zero probability given initial state distribution $\mu$, and we let $\mathcal{H}_\mu^{\boldsymbol{\pi}} \doteq \mathcal{H}_\mu^{\boldsymbol{\pi}, \infty}$ and $\nu_\mu^{\boldsymbol{\pi}} \doteq \nu_\mu^{\boldsymbol{\pi}, \infty}$. We can now write $H = \left(S^{(0)}, (A^{(t)}, S^{(t+1)})_{t=0}^{\tau-1}\right)$ to denote a history $\boldsymbol{h} \in \mathcal{H}_\mu^{\boldsymbol{\pi}, \tau}$ sampled from $\nu_\mu^{\boldsymbol{\pi}, \tau}$.

Now, given a policy profile $\boldsymbol{\pi} \in \mathcal{P}$, we define the *state-value function* $\boldsymbol{v}^{\boldsymbol{\pi}} : \mathcal{S} \to \mathbb{R}^n$ and the *action-value function* $\boldsymbol{q}^{\boldsymbol{\pi}} : \mathcal{S} \times \mathcal{A} \to \mathbb{R}^n$, respectively, as

$$\boldsymbol{v}^{\boldsymbol{\pi}}(\boldsymbol{s}) \doteq \mathbb{E}_{H \sim \nu_s^{\boldsymbol{\pi}}} \left[ \sum_{t=0}^{\infty} \boldsymbol{r}(S^{(t)}, A^{(t)}) \right] \tag{1}$$

$$= \mathbb{E}_{H \sim \nu_\mu^{\boldsymbol{\pi}}} \left[ \sum_{t=0}^{\infty} \boldsymbol{r}(S^{(t)}, A^{(t)}) \mid S^{(0)} = \boldsymbol{s} \right] \tag{2}$$

$$= \int_{H \in \mathcal{H}_\mu^{\boldsymbol{\pi}} : S^{(0)} = \boldsymbol{s}} \sum_{t=0}^{\infty} \boldsymbol{r}(S^{(t)}, A^{(t)}) \, \nu_\mu^{\boldsymbol{\pi},\tau}(H) \, dH \tag{3}$$

$$\boldsymbol{q}^{\boldsymbol{\pi}}(\boldsymbol{s}, \boldsymbol{a}) \doteq \boldsymbol{r}(\boldsymbol{s}, \boldsymbol{a}) + \mathbb{E}_{S' \sim p(S' | \boldsymbol{s}, \boldsymbol{a})} \left[ \boldsymbol{v}^{\boldsymbol{\pi}}(S') \right] \tag{4}$$

Finally, we define the *discounted state-visitation distribution* $\delta_\mu^{\boldsymbol{\pi}}(\boldsymbol{s}) \doteq \sum_{\tau=0}^{\infty} \int_{\boldsymbol{h} \in \mathcal{H}_\mu^{\boldsymbol{\pi},\tau} : \boldsymbol{s}^{(\tau)} = \boldsymbol{s}} \nu_\mu^{\boldsymbol{\pi},\tau}(\boldsymbol{h})$ and the *(expected) payoff* of policy profile $\boldsymbol{\pi}$ as $\boldsymbol{u}(\boldsymbol{\pi}) \doteq \mathbb{E}_{S \sim \mu} [\boldsymbol{v}^{\boldsymbol{\pi}}(S)] = \mathbb{E}_{S \sim \delta_\mu^{\boldsymbol{\pi}}} [\boldsymbol{r}(S, \boldsymbol{\pi}(S))]$.

## B  RELATED WORK

Beyond the works mentioned earlier, our paper is close to two literature on stochastic economies, one in financial economics which theoretical and focuses on understanding mathematical properties of general equilibrium competitive equilibrium in incomplete markets Duffie (1987); Selby (1990); Duffie & Shafer (1985; 1986), and another one in macroeconomics which focuses on the computation of sequential or recursive competitive equilibrium in incomplete stochastic economies to simulate various macroeconomic issues; see, for instance, Kydland & Prescott (1982) and Lucas Jr & Prescott (1971).[10]

**Financial economics**  Regarding the literature in financial economics, we refer the reader to the survey work of Magill & Quinzii (2002), and mention here only a few of some the influential models for the development of stochastic economies. Following the initial interest of the early 1970, the literature on stochastic economies in financial economics mostly focused on stochastic economies with two time-periods up to the end of the 1980s. In the early 1980s, there was an explosion of option pricing studies and arbitrage pricing in the early 1980s (See, for example, Cox & Ross (1976); Ross (1976) Cox et al. (1979), Cox et al. (1985), and Huberman (1982).] By the mid-1980s, the theory of stochastic economies made great strides, with two influential papers, Cass (1984; 1985) showing that existence of a general equilibrium could be guaranteed if all the assets promise delivery in fiat money, and he showed that with such financial assets there could be a multiplicity of equilibrium. In contrast, our existence result does not assume the existence of fiat money. Almost simultaneously Werner (1985) also gave a proof of existence of equilibrium with financial assets, and Geanakoplos & Polemarchakis (1986) showed the same for economies with real assets that promise delivery in the same consumption good. Duffie (1987) then extended the existence results for purely financial assets to arbitrary finite horizon stochastic economies. As stochastic economies with incomplete asset markets have been shown to not satisfy a first welfare theorem of economics, following preliminary insight from Diamond (1967) the literature turned its attention definine notions of constrained efficiency. Successive refinements of the definition were given by Diamond (1980), Loong & Zeckhauser (1982), Newbery & Stiglitz (1982), Stiglitz (1982), and Greenwald & Stiglitz (1986) with a mostly accepted definition of constrained efficiency becoming clear by the late 1980, with Geanakoplos (1990) eventually proving that sequential competitive equilibrium are is constrained efficient inefficient.

**Macroeconomics**  The literature on stochastic economies in macroeconomics is known under the name of *dynamic stochastic general equilibrium (DSGE) models*. Stochastic economies have received interest in macroeconomics after Lucas Jr's (1978) seminal work, in which he derived a

---

[10]Since the 90s, a sizable body of work in financial economics (see for instance Fernández-Villaverde et al. (2016); Auclert et al. (2021) has considered computational approaches to solving general equilibrium models of financial markets; however, much of this work can be seen as extension of the macroeconomics literature.

recursive competitive equilibrium in closed form in a stochastic economy with one commodity and and one consumer allowing him to analyze asset prices in his model. Unfortunately, beyond Lucas' simpler model, it became apparent that analyzing the solutions of stochastic economies required the use computation. One of the earliest popular stochastic economy models in economics which was solved via computational methods is the *Real Business Cycle (RBC)* model Kydland & Prescott (1982); Long Jr & Plosser (1983). The RBC model is a parameterized stochastic economy whose parameters are calibrated to accurately model the US economy. RBC models are characterized by demand generated by a representative infinitely-lived agent, with supply generated exogenously by a standard (or Solow) growth model Acemoglu (2008), or by a representative firm. These models have fallen out of favor, because some of their assumptions were invalidated by data (see, for example, Section 2 of Christiano et al. (2018)). They were replaced by a class of DSGE models known as *Representative Agent New Keynesian (RANK)* models (see, for instance, Clarida et al. (2000)). As RBC and RANK models derive their modelling assumptions from two different schools of macroeconomic thought (i.e., the New Keynesian and New Classical schools, respectively),from a mathematical and computational perspective they can be seen as the same, as both are characterized by a representative consumer and an exogenous growth model, or a representative firm.

Following the financial crisis of 2008, these representative-agent models, too, fell out of favor, and the literature turned to modeling heterogeneity, because of its importance in understanding inequality, in particular across consumers. *Heterogeneous agent new Keynesian models (HANK)* are stochastic economies which are built on top *Bewley-Huggett-Aiyagari* models Bewley (1983); Huggett (1993); Aiyagari (1994), and are characterized by demand and supply generated by an infinite population of agents with differing characteristics. These models are mathematically and computationally much more different than the RBC and RANK models and have been shown to be possible to model as single population mean-field games Achdou et al. (2022), i.e., games with an infinite population of players. More recently, a new class of stochastic economies called *Many Agent New Keynesian (MANK)* has emerged. This class of models bridges the gap between the infinite population regime of heterogeneous agent models and the single agent regime of representative agent models. These models are characterized by a demand and a supply generated by multiple consumers and firms, but are arguably more interpretable Eskelinen (2021) (see, for instance, Cloyne et al. (2020); Eskelinen (2021)) and have shown to approximate the solutions of heterogeneous agent models effectively when the number of agents in the economy is large enough Han et al. (2021). That is MANK models are sufficiently expressive to capture a range of models, corresponding to RANK at one extreme, and to HANK at the other. Ignoring the stylized details of the aforementioned stochastic economies, all of them feature static markets, linked over time, often although not always, by incomplete financial asset markets, and differ in the number and heterogeneity of the agents, firms and good in the economy, as well as the types of transitions they employ, whether it be transition functions which model *aggregate shocks* (i.e., transitions functions which change the state of each consumer and firm in the economy in the same way) or *idiosyncratic shock* (i.e. transition function which model transition the state of each consumer in the economy in distinct way). *The infinite horizon Markov exchange economy that we develop in this paper corresponds to a many agent stochastic economy model, and can be coupled with either the New Keynesian or New Classical paradigm to capture most of the models proposed in the literature.*

**Computation of competitive equilibrium**   The study of the computational complexity of competitive equilibria was initiated by Devanur et al. (2002), who provided a polynomial-time method for computing competitive equilibrium in a special case of the Arrow-Debreu (exchange) market model, namely Fisher markets, when buyers utilities are linear. Jain et al. (2005) subsequently showed that a large class of Fisher markets with homogeneous utility functions could be solved in polynomial-time using interior point methods. Gao & Kroer (2020) studied an alternative family of first-order methods for solving Fisher markets, assuming linear, quasilinear, and Leontief utilities, as such methods can be more efficient when markets are large. More recently, Goktas et al. (2023b) showed that tâtonnement converges to competitive equilibrium in homothetic Fisher markets, assuming bounded elasticity of Hicksian demand.

Devising algorithms for the computation of competitive equilibrium in general Arrow-Debreu markets is still an active area of research. While the computation of competitive equilibrium is PPAD-hard in general Chen & Deng (2006); Daskalakis et al. (2009), the computation of competitive equilibrium in Arrow-Debreu markets with Leontief buyers is equivalent to the computation of Nash equilibrium in bimatrix games Codenotti et al. (2006); Deng & Du (2008), and hence PPAD-hard as well, there exist

polynomial-time algorithms to compute competitive equilibrium in special cases of Arrow-Debreu markets, including markets whose excess demand satisfies the weak gross substitutes condition Codenotti et al. (2005); Bei et al. (2015) and Arrow-Debreu markets with buyers whose utilities are linear Garg & Kapoor (2004); Brânzei et al. (2021) or satisfy constant elasticity of substitution, which gives rise to weak gross substitute demands Brânzei et al. (2021).

**Solution methods in macroeconomics.** As stochastic economies can be analytically intractable to solve without restrictive assumptions, such as homogeneous consumers (e.g., representative agent new Keynesian models models, for a survey, see Sargent & Ljungqvist (2000)), researchers have attempted to solve them via dynamic programming. These methods often discretize the continuous state and action spaces, and then apply variants of value and policy iteration Stokey (1989); Sargent & Ljungqvist (2000); Auclert et al. (2021). Unfortunately, this approach is unwieldy when applied to incomplete markets with multiple commodities and/or heterogeneous consumers Fernández-Villaverde et al. (2016). As a result, many of these methods lack optimality guarantees, and thus might not produce correct solutions, which may lead to drastically different policy recommendations, as inaccurate solutions to stochastic economies have been known to cause spurious welfare reversal Kim & Kim (2003). Perhaps even more importantly, while static markets afford efficient, i.e., polynomial-time, algorithms for computing competitive equilibrium under suitable assumption (see, for instance Jain et al. (2005) or Goktas et al. (2023b) for a more recent survey), to the best of our knowledge, there is no known class of stochastic economies (excluding the special case of static economies) for which the computation of a sequential or recursive competitive equilibrium is polynomial-time. Yet the macroeconomics literature speaks to the need for efficient methods to solve these models, or at least better understand the trade-offs between the speed and accuracy of proposed solution techniques Fernández-Villaverde et al. (2016).

We describe only a few of the most influential computational approaches to solving stochastic economies in macroeconomics, and refer the reader to Fernández-Villaverde et al. (2016) for a detailed survey. Durbin & Koopman 2012 developed an extended path algorithm. The idea was to solve, for a terminal date sufficiently far into the future, the path of endogenous variables using a shooting algorithm. Recently, Maliar et al. (2015) extended this idea, developing the extended function path (EFP) algorithm, applicable to models that do not admit stationary Markov equilibria. Kydland & Prescott (1982) exploit the fact that their model admits a Pareto-optimal recursive equilibrium, and thus they solve the social planner's problem, instead of solving for an equilibrium. To do so, they rely on a linear quadratic approximation, and exploit the fast algorithms known to solve that class of optimization problems. King et al. (2002) (in the widely disseminated technical appendix, not published until 2002), building on Blanchard & Kahn (1980)'s approach, linearized the equilibrium conditions of their model (optimality conditions, market clearing conditions, etc.), and solved the resulting system of stochastic linear difference equations. More recently, a growing literature has been applying deep learning methods in attempt to stochastic economies (see, for instance, Curry et al.; Han et al. (2021); Childers et al. (2022)). There also exists a large literature in macroeconomics on solution methods in continuous rather than discrete time, which is out of the scope of this paper. We refer the interested reader to Parra-Alvarez (2015).

## C  HISTORICAL BACKGROUND AND EARLY MODELS

### C.1  DEVELOPMENT OF GENERAL EQUILIBRIUM MODELS

In 1896, Léon Walras formulated a mathematical model of markets as a system for resource allocation comprising supply and demand functions that map values for resources, called *prices*, to quantities of resources—*ceteris paribus*, i.e., all else being equal. Walras argued that any market would eventually settle into a steady state, which he called *competitive* (nowadays, also called *Walrasian*) *equilibrium*, as a collection of prices and associated supply and demand such that the demand is *feasible*, i.e., the demand for each resource is less than or equal to its supply, and *Walras' law* holds, i.e., the value of the supply is equal to the value of the demand. Unlike in Walras' model, real-world markets do not exist in isolation but are part of an *economy*. Indeed, the supply and demand of resources in one market depend not only on prices in that market, but also on the supply and demand of resources in other markets. If every market in an economy is simultaneously at a competitive equilibrium, Walras' law holds for the economy as a whole; this steady state, now a property of the economy, is called a *general equilibrium*.

Beyond Walras' early forays into competitive equilibrium analysis, foremost to the development of the theory of general equilibrium was the introduction of a broad mathematical framework for modeling economies, which is known today as the *Arrow-Debreu competitive economy* Arrow & Debreu (1954a). In this same paper, Arrow & Debreu developed their seminal game-theoretic model, namely (quasi)concave pseudo-games, and proved the existence of generalized Nash equilibrium in this model. Since this game-theoretic model is sufficiently rich to capture Arrow-Debreu economies, they obtained as a corollary the existence of general equilibrium in these economies.

In their model, Arrow & Debreu posit a set of resources, modeled as commodities, each of which is assigned a price; a set of consumers, each choosing a quantity of each commodity to consume in exchange for their endowment; and a set of firms, each choosing a quantity of each commodity to produce, with prices determining *(aggregate) demand*, i.e., the sum of the consumptions across all consumers, and *(aggregate) supply*, i.e., the sum of endowments and productions across all consumers and firms, respectively. This model is *static*, as it comprises only a single period model, but it is nonetheless rich, as commodities can be state and time contingent, with each one representing a good or service which can be bought or sold in a single time period, but that encodes delivery opportunities at a finite number of distinct points in time. Following Arrow & Debreu's seminal existence result, the literature slowly turned away from static economies, such as Arrow-Debreu competitive economies, which do not *explicitly* involve time and uncertainty.

Arrow & Debreu's model fails to provide a comprehensive account of the economic activity observed in the real world, especially that which is designed to account for *time* and *uncertainty*. Chief among these activities are *asset markets*, which allow consumers and firms to insure themselves against uncertainty about future states of the world. Indeed, while static economies with state- and time-contingent commodities can *implicitly* incorporate time and uncertainty, the assumption that a complete set of state- and time-contingent commodities are available at the time of trade is highly unrealistic. Arrow (1964) thus proposed to enhance the Arrow-Debreu competitive economy with *assets* (or *securities* or *stocks*),[11] i.e., contracts between two consumers, which promise the delivery of commodities by its seller to its buyer at a future date. In particular, Arrow introduced an asset type nowadays known as the *numéraire Arrow security*, which transfers one unit of a designated commodity used as a unit of account—*the numéraire*—when a particular state of the world is observed, and nothing otherwise. As the numéraire is often interpreted as money, assets which deliver only some amount of the numéraire, are called *financial assets*.

Formally, Arrow considered a *two-step stochastic exchange economy*. In the initial state, consumers can buy or sell *numéraire* Arrow securities in a *financial asset market*. Following these trades, the economy stochastically transitions to one of finitely many other states in which consumers receive returns on their initial investment and participate in a *spot market*, i.e., a market for the immediate delivery of commodities, modeled as a static exchange economy—which, for our purposes, is better called an *exchange market*.[12] A general equilibrium of this economy is then simply prices for financial assets *and* commodities, which lead to a feasible allocation of all resources (i.e., financial assets and spot market commodities) that satisfies Walras' law.

Arrow (1964) demonstrated that the general equilibrium consumptions of an exchange economy with state- and time-contingent commodities can be implemented by the general equilibrium spot market consumptions of a two-step stochastic economy with a considerably smaller, yet *complete* set of *numéraire* Arrow securities, i.e., a set of securities available for purchase in the first period that allow consumers to transfer wealth to *all* possible states of the world that can be realized in the second period. In conjunction with the welfare theorems Debreu (1951); Arrow (1951), this result implies that economies with *complete financial asset markets*, i.e., economies with such a complete set of securities, achieve a Pareto-optimal allocation of commodities by ensuring optimal risk-bearing via financial asset markets; and conversely, any Pareto-optimal allocation of commodities in economies

---

[11]Some authors (e.g., Geanakoplos (1990)) distinguish between assets, stocks, and securities, instead defining securities (resp. stocks) as those assets which are defined exogenously (resp. endogenously), e.g., government bonds (resp. company stocks). As this distinction makes no mathematical difference to our results, and is only relevant to stylized models, we make no such distinction.

[12]An (Arrow-Debreu) exchange economy is simply an (Arrow-Debreu) competitive economy without firms. Historically, for simplicity, it has become standard practice *not* to model firms, as most, if not all, results extend directly to settings with firms. In line with this practice, we do not model firms, but note that our results and methods also extend directly to settings that include firms.

with time and uncertainty can be realized as a competitive equilibrium of a complete financial asset market.

Arrow's contributions led to the development of a new class of general equilibrium models, namely *stochastic economies* (or *dynamic stochastic general equilibrium—DSGE—models*) Geanakoplos (1990).[13] At a high-level, these models comprise a sequence of world states and spot markets, which are linked across time by asset markets, with each next state of the world (resp. spot market) determined by a stochastic process that is independent of market interactions (resp. dependent only on their asset purchase) in the current state. Mathematically, the key difference between a static and a stochastic economy is that consumers in a stochastic economy face a collection of budget constraints, one per time-step, rather than only one. Indeed, Arrow (1964)'s proof that general equilibrium consumptions in stochastic complete economies are equivalent to general equilibrium consumptions in static state- and time-contingent commodity economies relies on proving that the many budget constraints in a complete stochastic economy can be reduced to a single one.

Stochastic economies were introduced to model arbitrary finite time horizons Radner (1968) and a variety of risky asset classes (e.g., stocks Diamond (1967), risky assets Lintner (1975), derivatives Black & Scholes (1973), capital assets Mossin (1966), debts Modigliani & Miller (1958) etc.), eventually leading to the emergence of *stochastic economies with incomplete asset markets* Magill & Shafer (1991); Magill & Quinzii (2002); Geanakoplos (1990), or colloquially, *(incomplete) stochastic economies*.[14] Unlike in Arrow's stochastic economy, the asset market is not complete in such economies, so consumers cannot necessarily insure themselves against all future world states.

The archetypal stochastic economy is the *Radner stochastic exchange economy*, deriving its name from Radner's proof of existence of a general equilibrium in his model Radner (1972). Radner's economy is a finite-horizon stochastic economy comprising a sequence of spot markets, linked across time by asset markets. At each time period, a finite set of consumers observe a world state and trade in an asset market and a spot market, modeled as an exchange market. Each *asset market* comprises *assets*, modelled as time-contingent *generalized Arrow securities*, which specify quantities of the commodities the seller is obliged to transfer to its buyer, should the relevant state of the economy be realized at some specified future time.[15] Consumers can buy and sell assets, thereby transferring their wealth across time, all the while insuring themselves against uncertainty about the future. The canonical solution concept for stochastic economies, *Radner equilibrium* (also called *sequential competitive equilibrium*[16] Mas-Colell et al. (1995), *rational expectations equilibrium* Radner (1979), and *general equilibrium with incomplete markets* Geanakoplos (1990)), is a collection of history-dependent prices for commodities and assets, as well as history-dependent consumptions of commodities and portfolios of assets, such that, for all histories, the aggregate demand for commodities and the *aggregate demand for assets* (i.e., the total number of assets bought) are feasible and satisfy Walras' law.

In spite of substantial interest in stochastic economies among microeconomists throughout the 1970s, the literature eventually trailed off, perhaps due to the difficulty of proving existence of a general equilibrium in simple economies with incomplete asset markets that allow assets to be sold short Geanakoplos (1990), or to the lack of a second welfare theorem Dreze (1974); Hart (1975). Financial and macroeconomists stepped up, however, with financial economists seeking to further develop the theoretical aspects of stochastic economies (see, for instance, Magill & Quinzii (2002)), and macroeconomists seeking practical methods by which to solve stochastic economies in order

---

[13]As these models incorporate both time and uncertainty, they are often referred to as dynamic stochastic general equilibrium models. Nonetheless, we opt for the stochastic economy nomenclature, because, as we demonstrate in this paper, these economies can be seen as instances of (generalized) stochastic games.

[14]While many authors have called these models incomplete economies Geanakoplos (1990); Magill & Quinzii (2002); Magill & Shafer (1991), these models capture both incomplete and complete asset markets. In contrast, we refer to stochastic economies with incomplete or complete asset markets as *stochastic economies*, adding the (in)complete epithet only when necessary to indicate that the asset market is (in)complete.

[15]Here, Arrow securities are "generalized" in the sense that they can deliver different quantities of *many* commodities at different states of the world, rather than only one unit of a commodity at only one state of the world. Although Arrow (1964) considered only *numéraire* securities, his theory was subsequently generalized to generalized Arrow securities Geanakoplos (1990).

[16]This terminology does not contradict the economy being at a competitive equilibrium, but rather indicates that at all times, the spot and asset markets are at a competitive equilibrium, hence implying the overall economy is at a general equilibrium.

to determine the impact of various policy choices (via simulation; see, for instance, Sargent & Ljungqvist (2000)).

*Infinite horizon stochastic economies* are one of the new and interesting directions in this more recent work on stochastic economies. Infinite horizon models come with one significant difficulty that has no counterpart in a finite horizon model, namely the possibility for agents to run a *Ponzi scheme* via asset markets, in which they borrow but then indefinitely postpone repaying their debts by refinancing them continually, from one period to the next. From this perspective infinite horizon models represent very interesting objects of study, not only theoretically; it has also been argued that they are a better modeling paradigm for macroeconomists who employ simulations Magill & Quinzii (1994), because they facilitate the modeling of complex phenomena, such as asset bubbles Huang & Werner (2000), which can be impacted by economic policy decisions.

Magill & Quinzii (1994) introduced an extension of Radner's model to an infinite horizon setting, albeit with financial assets, and presented suitable assumptions under which a sequential competitive equilibrium is guaranteed to exist in this model. Progress on the computational aspects of stochastic economies has been slow, however, and mostly confined to finite horizon settings (see, Sargent & Ljungqvist (2000) and Volume 2 of Taylor & Woodford (1999) for a standard survey, and Fernández-Villaverde (2023) for a more recent entry-level survey of computational methods used by macroeconomists). Indeed, demands for novel computational methods for solving macroeconomic models, and theoretical frameworks in which to understand their computational complexity, have been repeatedly shared by macroeconomists Taylor & Woodford (1999). This gap in the literature points to a novel research opportunity; however, it is challenging for non-macroeconomists to approach these problems with their computational tools.

## C.2 STATIC EXCHANGE ECONOMIES

A *static exchange economy* (or *market*[17]) $(n, m, d, \mathcal{X}, \mathcal{E}, \mathcal{T}, \boldsymbol{r}, \boldsymbol{E}, \boldsymbol{\Theta})$, abbreviated by $(\boldsymbol{E}, \boldsymbol{\Theta})$ when clear from context, comprises a finite set of $n \in \mathbb{N}_+$ *consumers* and $m \in \mathbb{N}_+$ *commodities*. Each consumer $i \in [n]$ arrives at the market with an *endowment* of commodities represented as vector $\boldsymbol{e}_i = (e_{i1}, \ldots, e_{im}) \in \mathcal{E}_i$, where $\mathcal{E}_i \subset \mathbb{R}^m$ is called the *endowment space*.[18] Any consumer $i$ can sell its endowment $\boldsymbol{e}_i \in \mathcal{E}_i$ at *prices* $\boldsymbol{p} \in \Delta_m$, where $p_j \geq 0$ represents the value (resp. cost) of selling (resp. buying) a unit of commodity $j \in [m]$, to purchase a consumption $\boldsymbol{x}_i \in \mathcal{X}_i$ of commodities in its *consumption space* $\mathcal{X}_i \subseteq \mathbb{R}^m$.[19] Every consumer is constrained to buy a consumption with a cost weakly less than the value of its endowment, i.e., consumer $i$'s *budget set*—the set of consumptions $i$ can afford with its endowment $\boldsymbol{e}_i \in \mathcal{E}_i$ at prices $\boldsymbol{p} \in \Delta_m$—is determined by its *budget correspondence* $\mathcal{B}_i(\boldsymbol{e}_i, \boldsymbol{p}) \doteq \{\boldsymbol{x}_i \in \mathcal{X}_i \mid \boldsymbol{x}_i \cdot \boldsymbol{p} \leq \boldsymbol{e}_i \cdot \boldsymbol{p}\}$.

Each consumer's consumption preferences are determined by its type-dependent preference relation $\succeq_{i, \boldsymbol{\theta}_i}$ on $\mathcal{X}_i$, represented by a type-dependent *utility function* $\boldsymbol{x}_i \mapsto r_i(\boldsymbol{x}_i; \boldsymbol{\theta}_i)$, for *type* $\boldsymbol{\theta}_i \in \mathcal{T}_i$ that characterizes consumer $i$'s preferences within the *type space* $\mathcal{T}_i \subset \mathbb{R}^d$ of possible preferences.[20] The goal of each consumer $i$ is thus to buy a consumption $\boldsymbol{x}_i \in \mathcal{B}_i(\boldsymbol{e}_i, \boldsymbol{p})$ that maximizes its utility function $\boldsymbol{x}_i \mapsto r_i(\boldsymbol{x}_i; \boldsymbol{\theta}_i)$ over its budget set $\mathcal{B}_i(\boldsymbol{e}_i, \boldsymbol{p})$.

---

[17]Although a static exchange "market" is an economy, we prefer the term "market" for the static components of an infinite horizon Markov exchange economy, a dynamic exchange economy in which each time-period comprises one static market among many.

[18]Commodities are assumed to include labor services. Further, for any consumer $i$ and endowment $\boldsymbol{e}_i \in \mathcal{E}_i$, $e_{ij} \geq 0$ denotes the quantity of commodity $j$ in consumer $i$'s possession, while $e_{ij} < 0$ denotes consumer $i$'s debt, in terms of commodity $j$.

[19]We note that, for any labor service $j$, consumer $i$'s consumption $x_{ij}$ is negative and restricted by its consumption space to be lower bounded by the negative of $i$'s endowment, i.e., $x_{ij} \in [-e_{ij}, 0]$. This modeling choice allows us to model a consumer's preferences over the labor services she can provide. More generally, the consumption space models the constraints imposed on consumption by the "physical properties" of the world. That is, it rules out impossible combinations of commodities, such as strictly positive quantities of a commodity that is not available in the region where a consumer resides, or a supply of labor that amounts to more than 24 labor hours in a given day.

[20]In the sequel, we will be assuming, for any consumer $i$ with any type $\boldsymbol{\theta}_i \in \mathcal{T}_i$, the type-dependent utility function $\boldsymbol{x}_i \mapsto r_i(\boldsymbol{x}_i; \boldsymbol{\theta}_i)$ is continuous, which implies that it can represent any type-dependent preference relation $\succeq_{i, \boldsymbol{\theta}_i}$ on $\mathbb{R}^m$ that is complete, transitive, and continuous Debreu et al. (1954).

We denote any *endowment profile* (resp. *type profile* and *consumption profile*) as $\boldsymbol{E} \doteq (\boldsymbol{e}_1, \ldots, \boldsymbol{e}_n)^T \in \mathcal{E}$ (resp. $\boldsymbol{\Theta} \doteq (\boldsymbol{\theta}_1, \ldots, \boldsymbol{\theta}_n)^T \in \mathcal{T}$ and $\boldsymbol{X} \doteq (\boldsymbol{x}_1, \ldots, \boldsymbol{x}_n)^T \in \mathcal{X}$). The *aggregate demand* (resp. *aggregate supply*) of a *consumption profile* $\boldsymbol{X} \in \mathcal{X}$ (resp. *an endowment profile* $\boldsymbol{E} \in \mathcal{E}$) is defined as the sum of consumptions (resp. endowments) across all consumers, i.e., $\sum_{i \in [n]} \boldsymbol{x}_i$ (resp. $\sum_{i \in [n]} \boldsymbol{e}_i$).

**Definition 3** (Arrow-Debreu Equilibrium). *An* Arrow-Debreu (or Walrasian or competitive) *equilibrium (ADE) of an exchange economy* $(\boldsymbol{E}, \boldsymbol{\Theta})$ *is a tuple* $(\boldsymbol{X}^*, \boldsymbol{p}^*) \in \mathcal{X} \times \Delta_m$, *which consists respectively of a consumption profile and prices* $\boldsymbol{p} \in \Delta_m$ *s.t.: 1. each consumer* $i$*'s equilibrium consumption maximizes its utility over its budget set:* $\boldsymbol{x}_i^* \in \arg\max_{\boldsymbol{x}_i \in \mathcal{B}_i(\boldsymbol{e}_i, \boldsymbol{p}^*)} r_i(\boldsymbol{x}_i; \boldsymbol{\theta}_i)$; *2. the consumption profile is* feasible, *meaning aggregate demand is less than or equal to aggregate supply:,* $\sum_{i=1}^n \boldsymbol{x}_i^* - \sum_{i=1}^n \boldsymbol{e}_i \leq \boldsymbol{0}_m$; *3.* Walras' law *holds, so that the cost of the aggregate demand is equal to the value of the aggregate supply:* $\boldsymbol{p}^* \cdot (\sum_{i=1}^n \boldsymbol{x}_i^* - \sum_{i=1}^n \boldsymbol{e}_i) = 0$.

# D   OMITTED ASSUMPTIONS, RESULTS, AND PROOFS

## D.1   OMITTED ASSUMPTIONS, RESULTS, AND PROOFS FROM SECTION 2

Before proceed to the assumptions, results and proofs, we present the algorithm we use, namely, Two time-scale stochastic simultaneous GDA (TTSSGDA) (Daskalakis et al., 2021):

---

**Algorithm 1** Two time-scale simultaneous SGDA

---

**Inputs:** $\mathcal{M}, (\boldsymbol{\pi}, \boldsymbol{\rho}, \mathbb{R}^\Omega, \mathbb{R}^\Sigma), \eta_{\boldsymbol{\omega}}, \eta_{\boldsymbol{\sigma}}, \boldsymbol{\omega}^{(0)}, \boldsymbol{\sigma}^{(0)}, T$

**Outputs:** $(\boldsymbol{\omega}^{(t)}, \boldsymbol{\sigma}^{(t)})_{t=0}^T$

1: Build gradient estimator $\widehat{G}$ associated with $\mathcal{M}$
2: **for** $t = 0, \ldots, T-1$ **do**
3: $\quad \boldsymbol{h} \sim \nu^{\boldsymbol{\omega}}, \boldsymbol{h}' \sim \bigtimes_{i \in [n]} \nu^{(\boldsymbol{\sigma}_i(\boldsymbol{\omega}_{-i}), \boldsymbol{\omega}_{-i})}$
4: $\quad \boldsymbol{\omega}^{(t+1)} \leftarrow \boldsymbol{\omega}^{(t)} - \eta_{\boldsymbol{\omega}} \widehat{G_{\boldsymbol{\omega}}}(\boldsymbol{\omega}^{(t)}, \boldsymbol{\sigma}^{(t)}; \boldsymbol{h}, \boldsymbol{h}')$
5: $\quad \boldsymbol{\sigma}^{(t+1)} \leftarrow \boldsymbol{\sigma}^{(t)} + \eta_{\boldsymbol{\sigma}} \widehat{G_{\boldsymbol{\sigma}}}(\boldsymbol{\omega}^{(t)}, \boldsymbol{\sigma}^{(t)}; \boldsymbol{h}, \boldsymbol{h}')$
6: **return** $(\boldsymbol{\omega}^{(t)}, \boldsymbol{\sigma}^{(t)})_{t=0}^T$

---

.

**Assumption 2** (Existence). *For all* $i \in [n]$, *assume 1.* $\mathcal{A}_i$ *is convex; 2.* $\mathcal{X}_i(\boldsymbol{s}, \cdot)$ *is upper- and lower-hemicontinuous, for all* $\boldsymbol{s} \in \mathcal{S}$; *3.* $\mathcal{X}_i(\boldsymbol{s}, \boldsymbol{a}_{-i})$ *is non-empty, convex, and compact, for all* $\boldsymbol{s} \in \mathcal{S}$ *and* $\boldsymbol{a}_{-i} \in \mathcal{A}_{-i}$; *and 4. for any policy* $\boldsymbol{\pi} \in \mathcal{P}$, $\boldsymbol{a}_i \mapsto q_i^{\boldsymbol{\pi}}(\boldsymbol{s}, \boldsymbol{a}_i, \boldsymbol{a}_{-i})$ *is continuous and concave over* $\mathcal{X}_i(\boldsymbol{s}, \boldsymbol{a}_{-i})$, *for all* $\boldsymbol{s} \in \mathcal{S}$ *and* $\boldsymbol{a}_{-i} \in \mathcal{A}_{-i}$.

**Assumption 3** (Policy Class). *Given* $\mathcal{P}^{\mathrm{sub}} \subseteq \mathcal{P}^{\mathrm{markov}}$, *assume 1.* $\mathcal{P}^{\mathrm{sub}}$ *is non-empty, compact, and convex; and 2. (Closure under policy improvement) for each* $\boldsymbol{\pi} \in \mathcal{P}^{\mathrm{sub}}$, *there exists* $\boldsymbol{\pi}^+ \in \mathcal{P}^{\mathrm{sub}}$ *s.t.* $q_i^{\boldsymbol{\pi}}(\boldsymbol{s}, \boldsymbol{\pi}_i^+(\boldsymbol{s}), \boldsymbol{\pi}_{-i}(\boldsymbol{s})) = \max_{\boldsymbol{\pi}_i' \in \mathcal{F}(\boldsymbol{\pi}_{-i})} q_i^{\boldsymbol{\pi}}(\boldsymbol{s}, \boldsymbol{\pi}_i'(\boldsymbol{s}), \boldsymbol{\pi}_{-i}(\boldsymbol{s}))$, *for all* $i \in [n]$ *and* $\boldsymbol{s} \in \mathcal{S}$.

**Theorem 2.1.** *If* $\mathcal{M}$ *is an MPG for which Assumption 2 holds, and* $\mathcal{P}^{\mathrm{sub}} \subseteq \mathcal{P}^{\mathrm{markov}}$ *is a subspace of Markov policy profiles that satisfies Assumption 3, then there exists a GMPE* $\boldsymbol{\pi}^* \in \mathcal{P}^{\mathrm{sub}}$ *of* $\mathcal{M}$.

*Proof.* First, by Part 3 of Assumption 2, we know that for any $i \in [n]$, $\mathcal{F}_i^{\mathrm{sub}}(\boldsymbol{\pi}_{-i})$ is non-empty, convex, and compact, for all $\boldsymbol{\pi}_{-i} \in \mathcal{P}_{-i}$. Moreover, 2 of Assumption 2, $\mathcal{F}^{\mathrm{sub}}$ is upper-hemicontinuous. Therefore, by the Fan's fixed-point theorem Fan (1952), the set $\mathcal{F}^{\mathrm{sub}} \doteq \{\boldsymbol{\pi} \in \mathcal{P}^{\mathrm{sub}} \mid \boldsymbol{\pi} \in \mathcal{F}^{\mathrm{sub}}(\boldsymbol{\pi})\}$ is non-empty.

For any player $i \in [n]$ and state $\boldsymbol{s} \in \mathcal{S}$, we define the *individual state best-response correspondence* $\Phi_i^{\boldsymbol{s}} : \mathcal{P}^{\mathrm{sub}} \rightrightarrows \mathcal{A}_i$ by

$$\Phi_i^{\boldsymbol{s}}(\boldsymbol{\pi}) \doteq \underset{\boldsymbol{a}_i \in \mathcal{X}_i(\boldsymbol{s}, \boldsymbol{\pi}_{-i}(\boldsymbol{s}))}{\arg\max} \, r_i(\boldsymbol{s}, \boldsymbol{a}_i, \boldsymbol{\pi}_{-i}(\boldsymbol{s})) + \underset{S' \sim p(\cdot | \boldsymbol{s}, \boldsymbol{a}_i, \boldsymbol{\pi}_{-i}(\boldsymbol{s}))}{\mathbb{E}} [\gamma v_i^{\boldsymbol{\pi}}(S')] \tag{5}$$

$$= \underset{\boldsymbol{a}_i \in \mathcal{X}_i(\boldsymbol{s}, \boldsymbol{\pi}_{-i}(\boldsymbol{s}))}{\arg\max} \, q_i^{\boldsymbol{\pi}}(\boldsymbol{s}, \boldsymbol{a}_i, \boldsymbol{\pi}_{-i}(\boldsymbol{s})) \tag{6}$$

Then, for any player $i \in [n]$, we define the *restricted individual best-response correspondence* $\Phi_i : \mathcal{P}^{\mathrm{sub}} \rightrightarrows \mathcal{P}_i^{\mathrm{sub}}$ as the Cartesian product of individual state best-response correspondences across the states restricted to $\mathcal{P}^{\mathrm{sub}}$:

$$\Phi_i(\boldsymbol{\pi}) = \left( \underset{\boldsymbol{s} \in \mathcal{S}}{\times} \Phi_i^{\boldsymbol{s}}(\boldsymbol{\pi}) \right) \bigcap \mathcal{P}_i^{\mathrm{sub}} \tag{7}$$

$$= \{ \boldsymbol{\pi}_i \in \mathcal{P}_i^{\mathrm{sub}} \mid \boldsymbol{\pi}_i(\boldsymbol{s}) \in \Phi_i^{\boldsymbol{s}}(\boldsymbol{\pi}), \forall \, \boldsymbol{s} \in \mathcal{S} \} \tag{8}$$

Finally, we can define the *multi-player best-response correspondence* $\Phi : \mathcal{P}^{\mathrm{sub}} \rightrightarrows \mathcal{P}^{\mathrm{sub}}$ as the Cartesian product of the individual correspondences, i.e., $\Phi(\boldsymbol{\pi}) \doteq \times_{i \in [n]} \Phi_i(\boldsymbol{\pi})$.

To show the existence of GMPE, we first want to show that there exists a fixed point $\boldsymbol{\pi}^* \in \mathcal{P}^{\mathrm{sub}}$ such that $\boldsymbol{\pi}^* \in \Phi(\boldsymbol{\pi}^*)$. To this end, we need to show that 1. for any $\boldsymbol{\pi} \in \mathcal{P}^{\mathrm{sub}}$, $\Phi(\boldsymbol{\pi})$ is non-empty, compact, and convex; 2. $\Phi$ is upper hemicontinuous.

Take any $\boldsymbol{\pi} \in \mathcal{P}^{\mathrm{sub}}$. Fix $i \in [n], \boldsymbol{s} \in \mathcal{S}$, we know that $\boldsymbol{a}_i \mapsto q_i^{\boldsymbol{\pi}}(\boldsymbol{s}, \boldsymbol{a}_i, \boldsymbol{\pi}_{-i}(\boldsymbol{s}))$ is concave over $\mathcal{X}_i(\boldsymbol{s}, \boldsymbol{\pi}_{-i}(\boldsymbol{s}))$, and $\mathcal{X}_i(\boldsymbol{s}, \boldsymbol{\pi}_{-i}(\boldsymbol{s}))$ is non-empty, convex, and compact by Assumption 2, then by Proposition 4.1 of Fiacco & Kyparisis (1986), $\Phi_i^{\boldsymbol{s}}(\boldsymbol{\pi})$ is non-empty, compact, and convex.

Now, notice $\times_{\boldsymbol{s} \in \mathcal{S}} \Phi_i^{\boldsymbol{s}}(\boldsymbol{\pi})$ is compact and convex as it is a Cartesian product of compact, convex sets. Thus, as $\mathcal{P}^{\mathrm{sub}}$ is also compact and convex by Assumption 3, we know that $\Phi_i(\boldsymbol{\pi}) = \left( \times_{\boldsymbol{s} \in \mathcal{S}} \Phi_i^{\boldsymbol{s}}(\boldsymbol{\pi}) \right) \bigcap \mathcal{P}_i^{\mathrm{sub}}$ is compact and convex. By the assumption of *closure under policy improvement* under Assumption 3, we know that since $\boldsymbol{\pi} \in \mathcal{P}^{\mathrm{sub}}$, there exists $\boldsymbol{\pi}^+ \in \mathcal{P}^{\mathrm{sub}}$ such that

$$\boldsymbol{\pi}_i^+ \in \underset{\boldsymbol{\pi}_i' \in \mathcal{F}_i^{\mathrm{markov}}(\boldsymbol{\pi}_{-i})}{\arg\max} q_i^{\boldsymbol{\pi}}(\boldsymbol{s}, \boldsymbol{\pi}_i'(\boldsymbol{s}), \boldsymbol{\pi}_{-i}(\boldsymbol{s}))$$

for all $\boldsymbol{s} \in \mathcal{S}$, and that means $\boldsymbol{\pi}_i^+(\boldsymbol{s}) \in \Phi_i^{\boldsymbol{s}}(\boldsymbol{\pi})$ for all $\boldsymbol{s} \in \mathcal{S}$. Thus, $\Phi_i(\boldsymbol{\pi})$ is also non-empty. Since Cartesian product preserves non-emptiness, compactness, and convexity, we can conclude that $\Phi(\boldsymbol{\pi}) = \times_{i \in [n]} \Phi_i(\boldsymbol{\pi})$ is non-empty, compact, and convex.

Similarly, fix $i \in [n], \boldsymbol{s} \in \mathcal{S}$, for any $\boldsymbol{\pi} \in \mathcal{P}^{\mathrm{sub}}$, since $\mathcal{X}_i(\boldsymbol{s}, \cdot)$ is continuous (i.e. both upper and lower hemicontinuous), by the Maximum theorem, $\Phi_i^{\boldsymbol{s}}$ is upper hemicontinuous. $\boldsymbol{\pi} \mapsto \times_{\boldsymbol{s} \in \mathcal{S}} \Phi_i^{\boldsymbol{s}}(\boldsymbol{\pi})$ is upper hemicontinuous as it is a Cartesian product of upper hemicontinuous correspondence, and consequently, $\boldsymbol{\pi} \mapsto \left( \times_{\boldsymbol{s} \in \mathcal{S}} \Phi_i^{\boldsymbol{s}}(\boldsymbol{\pi}) \right) \bigcap \mathcal{P}^{\mathrm{sub}}$ is also upper hemicontinuous. Therefore, $\Phi$ is also upper hemicontinuous.

Since $\Phi(\boldsymbol{\pi})$ is non-empty, compact, and convex for any $\boldsymbol{\pi} \in \mathcal{P}^{\mathrm{sub}}$ and $\Phi$ is upper hemicontinuous, by Fan's fixed-point theorem Fan (1952), $\Phi$ admits a fixed point.

Finally, say $\boldsymbol{\pi}^* \in \mathcal{P}^{\mathrm{sub}}$ is a fixed point of $\Phi$, and we want to show that $\boldsymbol{\pi}^*$ is a generalized Markov perfect equilibrium (GMPE) of $\mathcal{M}$. Since $\boldsymbol{\pi}^* \in \Phi(\boldsymbol{\pi}^*) = \times_{i \in [n]} \Phi_i(\boldsymbol{\pi}^*)$, we know that for all $i \in [n], \boldsymbol{\pi}_i^*(\boldsymbol{s}) \in \Phi_i^{\boldsymbol{s}}(\boldsymbol{\pi}^*) = \arg\max_{\boldsymbol{a}_i \in \mathcal{X}_i(\boldsymbol{s}, \boldsymbol{\pi}_{-i}^*(\boldsymbol{s}))} q_i^{\boldsymbol{\pi}^*}(\boldsymbol{s}, \boldsymbol{a}_i, \boldsymbol{\pi}_{-i}^*(\boldsymbol{s}))$. We now show that for any $i \in [n]$, for any $\boldsymbol{\pi}_i \in \mathcal{F}_i(\boldsymbol{\pi}_{-i}^*)$, $v_i^{\boldsymbol{\pi}^*}(\boldsymbol{s}) \geq v_i^{(\boldsymbol{\pi}_i, \boldsymbol{\pi}_{-i}^*)}(\boldsymbol{s})$ for all $\boldsymbol{s} \in \mathcal{S}$. Take any policy $\boldsymbol{\pi}_i \in \mathcal{F}_i(\boldsymbol{\pi}_{-i}^*)$. Note that $\boldsymbol{\pi}_i$ may not be Markov, so we denote $\{\boldsymbol{\pi}_i(\boldsymbol{h}_{:t})\}_{t \in \mathbb{N}} = \{\boldsymbol{a}_i^{(t)}\}_{t \in \mathbb{N}}$. Then, for all $\boldsymbol{s}^{(0)} \in \mathcal{S}$,

$$\begin{aligned}
&v_i^{\boldsymbol{\pi}^*}(\boldsymbol{s}^{(0)}) \\
&= q_i^{\boldsymbol{\pi}^*}(\boldsymbol{s}^{(0)}, \boldsymbol{\pi}_i^*(\boldsymbol{s}^{(0)}), \boldsymbol{\pi}_{-i}^*(\boldsymbol{s}^{(0)})) \\
&= \max_{\boldsymbol{a}_i \in \mathcal{X}_i(\boldsymbol{s}^{(0)}, \boldsymbol{\pi}_{-i}^*(\boldsymbol{s}^{(0)}))} q_i^{\boldsymbol{\pi}^*}(\boldsymbol{s}^{(0)}, \boldsymbol{a}_i, \boldsymbol{\pi}_{-i}^*(\boldsymbol{s}^{(0)})) \\
&= \max_{\boldsymbol{a}_i \in \mathcal{X}(\boldsymbol{s}^{(0)}, \boldsymbol{\pi}_{-i}^*(\boldsymbol{s}^{(0)}))} r_i(\boldsymbol{s}^{(0)}, \boldsymbol{a}_i, \boldsymbol{\pi}_{-i}^*(\boldsymbol{s}^{(0)})) + \underset{\boldsymbol{s}^{(1)} \sim p(\cdot | \boldsymbol{s}^{(0)}, \boldsymbol{a}_i, \boldsymbol{\pi}_{-i}^*(\boldsymbol{s}^{(0)}))}{\mathbb{E}} [\gamma v_i^{\boldsymbol{\pi}^*}(\boldsymbol{s}^{(1)})] \\
&\geq r_i(\boldsymbol{s}^{(0)}, \boldsymbol{a}_i^{(0)}, \boldsymbol{\pi}_{-i}^*(\boldsymbol{s}^{(0)})) + \underset{\boldsymbol{s}^{(1)} \sim p(\cdot | \boldsymbol{s}^{(0)}, \boldsymbol{a}_i^{(0)}, \boldsymbol{\pi}_{-i}^*(\boldsymbol{s}^{(0)}))}{\mathbb{E}} [\gamma v_i^{\boldsymbol{\pi}^*}(\boldsymbol{s}^{(1)})] \tag{9}
\end{aligned}$$

For any $\boldsymbol{s}^{(0)} \in \mathcal{S}$, define $v_i'(\boldsymbol{s}^{(0)}) \doteq r_i(\boldsymbol{s}^{(0)}, \boldsymbol{a}_i^{(0)}, \boldsymbol{\pi}_{-i}^*(\boldsymbol{s}^{(0)})) +$ $\mathbb{E}_{\boldsymbol{s}^{(1)} \sim p(\cdot|\boldsymbol{s}^{(0)}, \boldsymbol{a}_i^{(0)}, \boldsymbol{\pi}_{-i}^*(\boldsymbol{s}^{(0)}))}[\gamma v_i^{\boldsymbol{\pi}^*}(\boldsymbol{s}^{(1)})]$ . Since $v_i^{\boldsymbol{\pi}^*}(\boldsymbol{s}) \geq v_i'(\boldsymbol{s})$ for all $i \in [n]$, $\boldsymbol{s} \in \mathcal{S}$, we have for any $\boldsymbol{s}^{(0)} \in \mathcal{S}$

$$
\begin{aligned}
& v_i^{\boldsymbol{\pi}^*}(\boldsymbol{s}^{(0)}) \\
& \geq r_i(\boldsymbol{s}^{(0)}, \boldsymbol{a}_i^{(0)}, \boldsymbol{\pi}_{-i}^*(\boldsymbol{s}^{(0)})) + \mathbb{E}_{\boldsymbol{s}^{(1)} \sim p(\cdot|\boldsymbol{s}^{(0)}, \boldsymbol{a}_i^{(0)}, \boldsymbol{\pi}_{-i}^*(\boldsymbol{s}^{(0)}))}[\gamma v_i^{\boldsymbol{\pi}^*}(\boldsymbol{s}^{(1)})] \\
& \geq r_i(\boldsymbol{s}^{(0)}, \boldsymbol{a}_i^{(0)}, \boldsymbol{\pi}_{-i}^*(\boldsymbol{s}^{(0)})) + \mathbb{E}_{\boldsymbol{s}^{(1)} \sim p(\cdot|\boldsymbol{s}^{(0)}, \boldsymbol{a}_i^{(0)}, \boldsymbol{\pi}_{-i}^*(\boldsymbol{s}^{(0)}))}[\gamma v_i'(\boldsymbol{s}^{(1)})] \\
& \geq r_i(\boldsymbol{s}^{(0)}, \boldsymbol{a}_i^{(0)}, \boldsymbol{\pi}_{-i}^*(\boldsymbol{s}^{(0)})) \\
& \quad + \mathbb{E}_{\boldsymbol{s}^{(1)} \sim p(\cdot|\boldsymbol{s}^{(0)}, \boldsymbol{a}_i^{(0)}, \boldsymbol{\pi}_{-i}^*(\boldsymbol{s}^{(0)}))}\left[\gamma\left(r_i(\boldsymbol{s}^{(1)}, \boldsymbol{a}_i^{(1)}, \boldsymbol{\pi}_{-i}^*(\boldsymbol{s}^{(1)}))\right.\right. \\
& \quad + \mathbb{E}_{\boldsymbol{s}^{(2)} \sim p(\cdot|\boldsymbol{s}^{(1)}, \boldsymbol{a}_i^{(1)}, \boldsymbol{\pi}_{-i}^*(\boldsymbol{s}^{(1)}))}\left.\left.[\gamma v_i^{\boldsymbol{\pi}^*}(\boldsymbol{s}^{(2)})]\right)\right] \\
& \geq r_i(\boldsymbol{s}^{(0)}, \boldsymbol{a}_i^{(0)}, \boldsymbol{\pi}_{-i}^*(\boldsymbol{s}^{(0)})) \\
& \quad + \mathbb{E}_{\boldsymbol{s}^{(1)} \sim p(\cdot|\boldsymbol{s}^{(0)}, \boldsymbol{a}_i^{(0)}, \boldsymbol{\pi}_{-i}^*(\boldsymbol{s}^{(0)}))}\left[\gamma\left(r_i(\boldsymbol{s}^{(1)}, \boldsymbol{a}_i^{(1)}, \boldsymbol{\pi}_{-i}^*(\boldsymbol{s}^{(1)}))\right.\right. \\
& \quad + \mathbb{E}_{\boldsymbol{s}^{(2)} \sim p(\cdot|\boldsymbol{s}^{(1)}, \boldsymbol{a}_i^{(1)}, \boldsymbol{\pi}_{-i}^*(\boldsymbol{s}^{(1)}))}\left.\left.[\gamma v_i'(\boldsymbol{s}^{(2)})]\right)\right] \\
& \vdots \\
& \geq v_i^{(\boldsymbol{\pi}_i, \boldsymbol{\pi}_{-i}^*)}(\boldsymbol{s})
\end{aligned}
\tag{10}
$$

where in Equation (10), we recursively expand $v_i'$ and eliminate $\boldsymbol{v}^{\boldsymbol{\pi}^*}$ using Equation (9). We therefore conclude that for all states $\boldsymbol{s} \in \mathcal{S}$, and for all $i \in [n]$,

$$
v_i^{\boldsymbol{\pi}^*}(\boldsymbol{s}) \geq \max_{\boldsymbol{\pi}_i \in \mathcal{F}_i(\boldsymbol{\pi}_{-i}^*)} v_i^{(\boldsymbol{\pi}_i, \boldsymbol{\pi}_{-i}^*)}(\boldsymbol{s}).
$$

$\square$

**Lemma 1.** *Given an MPG $\mathcal{M}$, a Markov policy profile $\boldsymbol{\pi}^* \in \mathcal{F}^{\mathrm{markov}}(\boldsymbol{\pi}^*)$ is a GMPE iff $\phi(\boldsymbol{s}, \boldsymbol{\pi}^*) = 0$, for all states $\boldsymbol{s} \in \mathcal{S}$. Similarly, a policy profile $\boldsymbol{\pi}^* \in \mathcal{F}(\boldsymbol{\pi}^*)$ is an GNE iff $\varphi(\boldsymbol{\pi}^*) = 0$.*

*Proof of Lemma 1.* We first prove the result for state exploitability.

($\boldsymbol{\pi}^*$ is a GMPE $\implies \phi(\boldsymbol{s}, \boldsymbol{\pi}^*) = 0$ for all $\boldsymbol{s} \in \mathcal{S}$): Suppose that $\boldsymbol{\pi}^*$ is a GMPE, i.e., for all players $i \in [n]$, $v_i^{\boldsymbol{\pi}^*}(\boldsymbol{s}) \geq \max_{\boldsymbol{\pi}_i \in \mathcal{F}_i(\boldsymbol{\pi}_{-i}^*)} v_i^{(\boldsymbol{\pi}_i, \boldsymbol{\pi}_{-i}^*)}(\boldsymbol{s})$ for all state $\boldsymbol{s} \in \mathcal{S}$. Then, for all state $\boldsymbol{s} \in \mathcal{S}$, we have

$$
\forall i \in [n], \quad \max_{\boldsymbol{\pi}_i \in \mathcal{F}_i(\boldsymbol{\pi}_{-i}^*)} v_i^{(\boldsymbol{\pi}_i, \boldsymbol{\pi}_{-i}^*)}(\boldsymbol{s}) - v_i^{\boldsymbol{\pi}^*}(\boldsymbol{s}) = 0
\tag{11}
$$

Summing up across all players $i \in [n]$, we get

$$
\phi(\boldsymbol{s}, \boldsymbol{\pi}^*) = \sum_{i \in [n]} \max_{\boldsymbol{\pi}_i \in \mathcal{F}_i(\boldsymbol{\pi}_{-i}^*)} v_i^{(\boldsymbol{\pi}_i, \boldsymbol{\pi}_{-i}^*)}(\boldsymbol{s}) - v_i^{\boldsymbol{\pi}^*}(\boldsymbol{s}) = 0
\tag{12}
$$

($\phi(\boldsymbol{s}, \boldsymbol{\pi}^*) = 0$ for all $\boldsymbol{s} \in \mathcal{S} \implies \boldsymbol{\pi}^*$ is a GMPE): Suppose we have $\boldsymbol{\pi}^* \in \mathcal{F}^{\mathrm{markov}}(\boldsymbol{\pi}^*)$ and $\phi(\boldsymbol{s}, \boldsymbol{\pi}^*) = 0$ for all $\boldsymbol{s} \in \mathcal{S}$. That is, for any $\boldsymbol{s} \in \mathcal{S}$

$$
\phi(\boldsymbol{s}, \boldsymbol{\pi}^*) = \sum_{i \in [n]} \max_{\boldsymbol{\pi}_i \in \mathcal{F}_i(\boldsymbol{\pi}_{-i}^*)} v_i^{(\boldsymbol{\pi}_i, \boldsymbol{\pi}_{-i}^*)}(\boldsymbol{s}) - v_i^{\boldsymbol{\pi}^*}(\boldsymbol{s}) = 0.
\tag{13}
$$

Since for any $i \in [n]$, $\boldsymbol{\pi}_i^* \in \mathcal{F}_i^{\mathrm{markov}}(\boldsymbol{\pi}_{-i}^*)$, $\max_{\boldsymbol{\pi}_i \in \mathcal{F}_i^{\mathrm{markov}}(\boldsymbol{\pi}_{-i}^*)} v_i^{(\boldsymbol{\pi}_i, \boldsymbol{\pi}_{-i}^*)}(\boldsymbol{s}) - v_i^{\boldsymbol{\pi}^*} \geq v_i^{\boldsymbol{\pi}^*}(\boldsymbol{s}) - v_i^{\boldsymbol{\pi}^*}(\boldsymbol{s}) = 0$. As a result, we must have for all player $i \in [n]$,

$$v_i^{\boldsymbol{\pi}^*}(\boldsymbol{s}) = \max_{\boldsymbol{\pi}_i \in \mathcal{F}(\boldsymbol{\pi}_{-i}^*)} v_i^{(\boldsymbol{\pi}_i, \boldsymbol{\pi}_{-i}^*)}(\boldsymbol{s}), \quad \forall \boldsymbol{s} \in \mathcal{S} \tag{14}$$

Thus, we can conclude that $\boldsymbol{\pi}^*$ is a GMPE.

Then, we can prove results for exploitability in an analogous way.

($\boldsymbol{\pi}^*$ is a GNE $\implies \varphi(\boldsymbol{\pi}^*) = 0$ ): Suppose that $\boldsymbol{\pi}^*$ is a GNE, i.e., for all players $i \in [n]$, $u_i(\boldsymbol{\pi}^*) \geq \max_{\boldsymbol{\pi}_i \in \mathcal{F}_i(\boldsymbol{\pi}_{-i}^*)} u_i(\boldsymbol{\pi}_i, \boldsymbol{\pi}_{-i}^*)$. Then, we have

$$\forall i \in [n], \quad \max_{\boldsymbol{\pi}_i \in \mathcal{F}_i(\boldsymbol{\pi}_{-i}^*)} u_i(\boldsymbol{\pi}_i, \boldsymbol{\pi}_{-i}^*) - u_i(\boldsymbol{\pi}^*) = 0 \tag{15}$$

Summing up across all players $i \in [n]$, we get

$$\varphi(\boldsymbol{\pi}^*) = \sum_{i \in [n]} \max_{\boldsymbol{\pi}_i \in \mathcal{F}_i(\boldsymbol{\pi}_{-i}^*)} u_i(\boldsymbol{\pi}_i, \boldsymbol{\pi}_{-i}^*) - u_i(\boldsymbol{\pi}^*) = 0 \tag{16}$$

($\varphi(\boldsymbol{s}, \boldsymbol{\pi}^*) = 0 \implies \boldsymbol{\pi}^*$ is a GNE): Suppose we have $\boldsymbol{\pi}^* \in \mathcal{F}(\boldsymbol{\pi}^*)$ and $\varphi(\boldsymbol{\pi}^*) = 0$. That is,

$$\varphi(\boldsymbol{\pi}^*) = \sum_{i \in [n]} \max_{\boldsymbol{\pi}_i \in \mathcal{F}_i(\boldsymbol{\pi}_{-i}^*)} u_i(\boldsymbol{\pi}_i, \boldsymbol{\pi}_{-i}^*) - u_i(\boldsymbol{\pi}^*) = 0. \tag{17}$$

Since for any $i \in [n]$, $\boldsymbol{\pi}_i^* \in \mathcal{F}_i(\boldsymbol{\pi}_{-i}^*)$, $\max_{\boldsymbol{\pi}_i \in \mathcal{F}_i(\boldsymbol{\pi}_{-i}^*)} u_i(\boldsymbol{\pi}_i, \boldsymbol{\pi}_{-i}^*) - u_i(\boldsymbol{\pi}^*) \geq u_i(\boldsymbol{\pi}^*) - u_i(\boldsymbol{\pi}^*) = 0$. As a result, we must have for all player $i \in [n]$,

$$u_i(\boldsymbol{\pi}^*) = \max_{\boldsymbol{\pi}_i \in \mathcal{F}(\boldsymbol{\pi}_{-i}^*)} u_i(\boldsymbol{\pi}_i, \boldsymbol{\pi}_{-i}^*) \tag{18}$$

Thus, we can conclude that $\boldsymbol{\pi}^*$ is a GNE. $\qquad \square$

**Observation 1.** *Given an MPG $\mathcal{M}$, $\min_{\boldsymbol{\pi} \in \mathcal{F}(\boldsymbol{\pi})} \varphi(\boldsymbol{\pi}) = \min_{\boldsymbol{\pi} \in \mathcal{F}(\boldsymbol{\pi})} \max_{\boldsymbol{\pi}' \in \mathcal{F}^{\mathrm{markov}}(\boldsymbol{\pi})} \Psi(\boldsymbol{\pi}, \boldsymbol{\pi}')$.*

*Proof.* The per-player maximum operator can be pulled out of the sum in the definition of state-exploitability, because the $i$th player's best-response policy is independent of the other players' best-response policies, given a fixed policy profile $\boldsymbol{\pi}$:

$$\forall \, \boldsymbol{s} \in \mathcal{S}, \quad \phi(\boldsymbol{s}, \boldsymbol{\pi}) = \sum_{i \in [n]} \max_{\boldsymbol{\pi}_i' \in \mathcal{F}_i^{\mathrm{markov}}(\boldsymbol{\pi}_{-i})} v_i^{(\boldsymbol{\pi}_i', \boldsymbol{\pi}_{-i})}(\boldsymbol{s}) - v_i^{\boldsymbol{\pi}}(\boldsymbol{s}) \tag{19}$$

$$= \max_{\boldsymbol{\pi}' \in \mathcal{F}^{\mathrm{markov}}(\boldsymbol{\pi})} \sum_{i \in [n]} v_i^{(\boldsymbol{\pi}_i', \boldsymbol{\pi}_{-i})}(\boldsymbol{s}) - v_i^{\boldsymbol{\pi}}(\boldsymbol{s}) \tag{20}$$

$$= \max_{\boldsymbol{\pi}' \in \mathcal{F}^{\mathrm{markov}}(\boldsymbol{\pi})} \psi(\boldsymbol{s}, \boldsymbol{\pi}, \boldsymbol{\pi}') \tag{21}$$

The argument is analogous for exploitability:

$$\varphi(\boldsymbol{\pi}) = \sum_{i \in [n]} \max_{\boldsymbol{\pi}_i' \in \mathcal{F}_i^{\mathrm{markov}}(\boldsymbol{\pi}_{-i})} u_i(\boldsymbol{\pi}_i', \boldsymbol{\pi}_{-i}) - u_i(\boldsymbol{\pi}) \tag{22}$$

$$= \max_{\boldsymbol{\pi}' \in \mathcal{F}^{\mathrm{markov}}(\boldsymbol{\pi})} \sum_{i \in [n]} u_i(\boldsymbol{\pi}_i', \boldsymbol{\pi}_{-i}) - u_i(\boldsymbol{\pi}) \tag{23}$$

$$= \max_{\boldsymbol{\pi}' \in \mathcal{F}(\boldsymbol{\pi})} \Psi(\boldsymbol{\pi}, \boldsymbol{\pi}') \tag{24}$$

$\qquad \square$

**Lemma 2.** *Given an MPG $\mathcal{M}$,*
$\min_{\boldsymbol{\pi} \in \mathcal{F}(\boldsymbol{\pi})} \max_{\boldsymbol{\pi}' \in \mathcal{F}^{\mathrm{markov}}(\boldsymbol{\pi})} \Psi(\boldsymbol{\pi}, \boldsymbol{\pi}') = \min_{\boldsymbol{\pi} \in \mathcal{F}(\boldsymbol{\pi})} \max_{\boldsymbol{\rho} \in \mathcal{R}} \Psi(\boldsymbol{\pi}, \boldsymbol{\rho}(\cdot, \boldsymbol{\pi}(\cdot))).$

*Proof.* Fix $\boldsymbol{\pi}^* \in \mathcal{F}^{\mathrm{markov}}(\boldsymbol{\pi}^*)$. We want to show that

$$\max_{\boldsymbol{\pi}' \in \mathcal{F}^{\mathrm{markov}}(\boldsymbol{\pi}^*)} \varphi(\boldsymbol{\pi}^*, \boldsymbol{\pi}') = \max_{\boldsymbol{\rho} \in \mathcal{R}} \varphi(\boldsymbol{\pi}^*, \boldsymbol{\rho}(\cdot, \boldsymbol{\pi}(\cdot))) \ .$$

Define $\mathcal{P}^{\mathcal{R}, \boldsymbol{\pi}^*} \doteq \{ \boldsymbol{\pi} : \boldsymbol{s} \mapsto \boldsymbol{\rho}(\boldsymbol{s}, \boldsymbol{\pi}^*(\boldsymbol{s})) \mid \boldsymbol{\rho} \in \mathcal{R} \} \subseteq \mathcal{P}^{\mathrm{markov}}$.

First, for all $\boldsymbol{\pi}' \in \mathcal{P}^{\mathcal{R}, \boldsymbol{\pi}^*}$, $\boldsymbol{\pi}'(\boldsymbol{s}) = \boldsymbol{\rho}(\boldsymbol{s}, \boldsymbol{\pi}^*(\boldsymbol{s})) \in \mathcal{X}(\boldsymbol{s}, \boldsymbol{\pi}^*(\boldsymbol{s}))$, for all $\boldsymbol{s} \in \mathcal{S}$, by the definition of $\mathcal{R}$. Thus, $\boldsymbol{\pi}' \in \mathcal{F}^{\mathrm{markov}}(\boldsymbol{\pi}^*) = \{ \boldsymbol{\pi} \in \mathcal{P}^{\mathrm{markov}} \mid \forall \boldsymbol{s} \in \mathcal{S}, \boldsymbol{\pi}(\boldsymbol{s}) \in \mathcal{X}(\boldsymbol{s}, \boldsymbol{\pi}^*(\boldsymbol{s})) \}$. Therefore, $\mathcal{P}^{\mathcal{R}, \boldsymbol{\pi}^*} \subseteq \mathcal{F}^{\mathrm{markov}}(\boldsymbol{\pi}^*)$, which implies that $\max_{\boldsymbol{\pi}' \in \mathcal{F}^{\mathrm{markov}}(\boldsymbol{\pi}^*)} \varphi(\boldsymbol{\pi}^*, \boldsymbol{\pi}') \geq \max_{\boldsymbol{\pi}' \in \mathcal{P}^{\mathcal{R}, \boldsymbol{\pi}^*}} \varphi(\boldsymbol{\pi}^*, \boldsymbol{\pi}') = \max_{\boldsymbol{\rho} \in \mathcal{R}} \varphi(\boldsymbol{\pi}^*, \boldsymbol{\rho}(\cdot, \boldsymbol{\pi}(\cdot)))$.

Moreover, for all $\boldsymbol{\pi}' \in \mathcal{F}^{\mathrm{markov}}(\boldsymbol{\pi}^*)$, $\boldsymbol{\pi}'(\boldsymbol{s}) \in \mathcal{X}(\boldsymbol{s}, \boldsymbol{\pi}^*(\boldsymbol{s}))$, for all $\boldsymbol{s} \in \mathcal{S}$, by the definition of $\mathcal{F}^{\mathrm{markov}}$. Define $\boldsymbol{\rho}'$ such that for all $\boldsymbol{s} \in \mathcal{S}$, $\boldsymbol{\rho}'(\boldsymbol{s}, \boldsymbol{a}) = \boldsymbol{\pi}'(\boldsymbol{s})$ if $\boldsymbol{a} = \boldsymbol{\pi}^*(\boldsymbol{s})$, and $\boldsymbol{\rho}'(\boldsymbol{s}, \boldsymbol{a}) = \boldsymbol{a}'$ for some $\boldsymbol{a}' \in \mathcal{X}(\boldsymbol{s}, \boldsymbol{a})$ otherwise. Note that $\boldsymbol{\rho}' \in \mathcal{R}$, since $\forall (\boldsymbol{s}, \boldsymbol{a}) \in \mathcal{S} \times \mathcal{A}$, $\boldsymbol{\rho}(\boldsymbol{s}, \boldsymbol{a}) \in \mathcal{X}(\boldsymbol{s}, \boldsymbol{a})$. Thus, as $\boldsymbol{\pi}'(\boldsymbol{s}) = \boldsymbol{\rho}'(\boldsymbol{s}, \boldsymbol{\pi}^*(\boldsymbol{s}))$, for all $\boldsymbol{s} \in \mathcal{S}$, it follows that $\boldsymbol{\pi}' \in \mathcal{P}^{\mathcal{R}, \boldsymbol{\pi}^*}$. Therefore, $\mathcal{F}^{\mathrm{markov}}(\boldsymbol{\pi}^*) \subseteq \mathcal{P}^{\mathcal{R}, \boldsymbol{\pi}^*}$, which implies that $\max_{\boldsymbol{\pi}' \in \mathcal{F}^{\mathrm{markov}}(\boldsymbol{\pi}^*)} \varphi(\boldsymbol{\pi}^*, \boldsymbol{\pi}') \leq \max_{\boldsymbol{\pi}' \in \mathcal{P}^{\mathcal{R}, \boldsymbol{\pi}^*}} \varphi(\boldsymbol{\pi}^*, \boldsymbol{\pi}') = \max_{\boldsymbol{\rho} \in \mathcal{R}} \varphi(\boldsymbol{\pi}^*, \boldsymbol{\rho}(\cdot, \boldsymbol{\pi}(\cdot)))$.

Finally, we conclude that $\max_{\boldsymbol{\pi}' \in \mathcal{F}^{\mathrm{markov}}(\boldsymbol{\pi}^*)} \varphi(\boldsymbol{\pi}^*, \boldsymbol{\pi}') = \max_{\boldsymbol{\rho} \in \mathcal{R}} \varphi(\boldsymbol{\pi}^*, \boldsymbol{\rho}(\cdot, \boldsymbol{\pi}(\cdot)))$. $\qquad \square$

**Lemma 3.** *Given a MPG $\mathcal{M}$, if*

1. *For any player $i \in [n]$, $\boldsymbol{a}_{-i} \in \mathcal{A}_{-i}$, $\boldsymbol{a}_i \mapsto \mathcal{X}_i(\boldsymbol{s}, \boldsymbol{a}_{-i})$ is non-empty, compact, and convex for all $\boldsymbol{s} \in \mathcal{S}$.*

2. *There exists a partition $\mathcal{P} = \{P_1, \cdots, P_K\}$ of $[n]$ and a directed acyclic graph (DAG) $G = (\mathcal{P}, E)$ that specifies the precedence relations. For any $P_k \in \mathcal{P}$, let $anc(P_k) = \{P'_k \in \mathcal{P} \mid$ there is a direct path $P'_k \rightarrow \cdots \rightarrow P_k\}$ be the set of ancestors of $P_k$. For any $P_k \in \mathcal{P}$, and any player $i \in P_k$, $\mathcal{X}_i(\boldsymbol{s}, \boldsymbol{a}_i, \boldsymbol{a}_{-i}) = \{\boldsymbol{a}_i \in \mathcal{A}_i \mid g_{kl}(\boldsymbol{s}, \boldsymbol{a}_{P_k}, \boldsymbol{a}_{anc(P_k)}) \leq 0$ for all $l\}$ for some $g_{kl}$ that is concave in $\boldsymbol{a}_i$. Note that $\boldsymbol{a}_{P_k}$ and $\boldsymbol{a}_{anc(P_k)}$ denotes the action profile for all the players in group $P_k$ and action profile for all the players belongs to any ancestor groups of $P_k$.*

*Then, there exists parameterization scheme satsifies Assumption 1.*

*Proof.* For any MPG $\mathcal{M}$ that satisfies condition 2, We first construct a feasible joint policy $\pi(\cdot; \omega)$ by projecting raw neural-network outputs onto the feasible sets in a topological order over the DAG $G$.

Fix any parameter vector $\omega$. For each player $i \in [n]$, let

$$\boldsymbol{a}_i^{\mathrm{raw}}(s) = \mathrm{NN}_{\omega, i}(s)$$

denote the *raw* action output of a neural network at state $s$. Stack these into a raw joint action $\boldsymbol{a}^{\mathrm{raw}}(s) = (\boldsymbol{a}_i^{\mathrm{raw}}(s))_{i \in [n]}$.

Consider any group $P_k$ with $\mathrm{anc}(P_k) = \emptyset$. For such groups, the feasibility constraints do not depend on the actions of any other groups. By condition 2, for any $s$, the joint feasible set for players in $P_k$ can be written as

$$\mathcal{X}_{P_k}(s) = \left\{ \boldsymbol{a}_{P_k} \in \mathcal{A}_{P_k} \mid g_{kl}(s, \boldsymbol{a}_{P_k}) \leq 0 \text{ for all } l \right\}.$$

By condition 1 (applied componentwise and using the product structure across players in $P_k$), $\mathcal{X}_{P_k}(s)$ is non-empty, compact, and convex. Therefore we can define

$$\boldsymbol{a}_{P_k}(s) \triangleq \Pi_{\mathcal{X}_{P_k}(s)}(\boldsymbol{a}_{P_k}^{\mathrm{raw}}(s)),$$

where $\Pi_{\mathcal{X}_{P_k}(s)}(\cdot)$ denotes Euclidean projection onto the convex set $\mathcal{X}_{P_k}(s)$.

Now consider any group $P_k$ with $\mathrm{anc}(P_k) \neq \emptyset$. By the topological ordering on $G$, all ancestor groups $P_j \in \mathrm{anc}(P_k)$ have already been processed, so we have fixed feasible actions $\boldsymbol{a}_{anc(P_k)}(s)$.

For fixed $s$ and fixed ancestor actions $\boldsymbol{a}_{anc(P_k)}(s)$, define the joint feasible set for $P_k$ as

$$\mathcal{X}_{P_k}(s, \boldsymbol{a}_{anc(P_k)}(s)) = \left\{ \boldsymbol{a}_{P_k} \in \mathcal{A}_{P_k} \mid g_{kl}(s, \boldsymbol{a}_{P_k}, \boldsymbol{a}_{anc(P_k)}(s)) \leq 0 \text{ for all } l \right\}.$$

Again, by condition 1 and the product structure, this set is non-empty, compact, and convex. We can thus define

$$\boldsymbol{a}_{P_k}(s) \triangleq \Pi_{\mathcal{X}_{P_k}(s, \boldsymbol{a}_{anc(P_k)}(s))}(\boldsymbol{a}_{P_k}{}^{\text{raw}}(s)).$$

Proceeding in this way for all groups $P_k$ in a topological order over $G$ yields, for each state $s$, a well-defined joint action profile $\boldsymbol{\pi}(s; \boldsymbol{\omega}) = \boldsymbol{a}(s) = (\boldsymbol{a}_{P_k}(s))_{k=1}^{K}$ such that, by construction, all constraints of the form $g_{kl}(s, \boldsymbol{a}_{P_k}(s), \boldsymbol{a}_{anc(P_k)}(s)) \leq 0$ are satisfied. This means precisely that $\boldsymbol{\pi}(s; \boldsymbol{\omega}) \in \mathcal{X}(s, \boldsymbol{\pi}(s; \boldsymbol{\omega}))$ for every $s$.

Then, for the dependent policy $\boldsymbol{\rho}(\cdot, \boldsymbol{\pi}(\cdot; \boldsymbol{\omega}); \boldsymbol{\sigma})$, it's much simpler. Fix any parameter $\boldsymbol{\sigma}$. For each player $i \in [n]$, let

$$\boldsymbol{a}_i^{\text{raw}}(s) = \text{NN}_{\boldsymbol{\sigma}, i}(s)$$

denote the *raw* action output of a neural network at state $s$. Note that for any player $i \in [n]$, the feasible action set given the minimizer policy $\mathcal{X}_i(s, \boldsymbol{\pi}_{-i}(s; \boldsymbol{\omega}))$ is a fixed, convex set, so we can simply project every $\boldsymbol{a}_i^{\text{raw}}(s)$ into $\mathcal{X}_i(s, \boldsymbol{\pi}_{-i}(s; \boldsymbol{\omega}))$ through Euclidean projection:

$$\boldsymbol{\rho}(s, \boldsymbol{\pi}(s; \boldsymbol{\omega}); \boldsymbol{\sigma}) = \left( \Pi_{\mathcal{X}_i(s, \boldsymbol{\pi}_{-i}(s; \boldsymbol{\omega}))}(\boldsymbol{a}_i^{\text{raw}}(s)) \right)_{i \in [n]}$$

$\square$

**Lemma 4.** *Given an MPG $\mathcal{M}$ and a parameterization scheme $(\boldsymbol{\pi}, \boldsymbol{\rho}, \mathbb{R}^\Omega, \mathbb{R}^\Sigma)$ with $\phi(s, \cdot)$ differentiable at $\boldsymbol{\omega} \in \mathbb{R}^\Omega$, for all $s \in \mathcal{S}$. If $\|\nabla_{\boldsymbol{\omega}} \varphi(\boldsymbol{\omega})\| = 0$, then $\|\nabla_{\boldsymbol{\omega}} \phi(s, \boldsymbol{\omega})\| = 0$ $\mu$-almost surely, for all states $s \in \mathcal{S}$, i.e., $\mu(\{s \in \mathcal{S} \mid \|\nabla_{\boldsymbol{\omega}} \phi(s, \boldsymbol{\omega})\| = 0\}) = 1$. Moreover, for any $\varepsilon > 0$ and $\delta \in [0, 1]$, if $\text{supp}(\mu) = \mathcal{S}$ and $\|\nabla_{\boldsymbol{\omega}} \varphi(\boldsymbol{\omega})\| \leq \varepsilon$, then with probability at least $1 - \delta$, $\|\nabla_{\boldsymbol{\omega}} \phi(s, \boldsymbol{\omega})\| \leq \varepsilon/\delta$, for all states $s \in \mathcal{S}$.*

*Proof.* First, using Jensen's inequality, by the convexity of the 2-norm $\|\cdot\|$, we have:

$$\mathbb{E}_{s \sim \mu} \left[ \|\nabla_{\boldsymbol{\omega}} \phi(s, \boldsymbol{\omega})\| \right] \leq \left\| \mathbb{E}_{s \sim \mu} \left[ \nabla_{\boldsymbol{\omega}} \phi(s, \boldsymbol{\omega}) \right] \right\|$$

$$= \left\| \nabla_{\boldsymbol{\omega}} \mathbb{E}_{s \sim \mu} \left[ \phi(s, \boldsymbol{\omega}) \right] \right\|$$

$$= \|\nabla_{\boldsymbol{\omega}} \varphi(\boldsymbol{\omega})\| .$$

The first claim follows directly from the fact that for all $s \in \mathcal{S}$, $\|\nabla_{\boldsymbol{\omega}} \varphi(s, \boldsymbol{\omega})\| \geq 0$, and hence for the expectation $\mathbb{E}_{s \sim \mu} \left[ \|\nabla_{\boldsymbol{\omega}} \varphi(s, \boldsymbol{\omega})\| \right]$ to be equal to 0, its value should be equal to zero on a set of measure 1.

Now, for the second part, by Markov's inequality, we have: $\mathbb{P}\left( \|\nabla_{\boldsymbol{\omega}} \phi(s, \boldsymbol{\omega})\| \geq \varepsilon/\delta \right) \leq \frac{\mathbb{E}_{s \sim \mu} \left[ \|\nabla_{\boldsymbol{\omega}} \phi(s, \boldsymbol{\pi})\| \right]}{\varepsilon/\delta} \leq \frac{\varepsilon}{\varepsilon/\delta} = \delta.$ $\square$

**Lemma 5.** *Let $\mathcal{M}$ be an MPG with initial state distribution $\mu$. Given policy parameter $\boldsymbol{\omega} \in \mathbb{R}^\Omega$, for any state distribution $\upsilon \in \Delta(\mathcal{S})$, if both $\phi(\upsilon, \cdot)$ and $\varphi(\cdot)$ are differentiable at $\boldsymbol{\omega}$, then $\|\nabla \phi(\upsilon, \boldsymbol{\omega})\| \leq C_{br}(\boldsymbol{\pi}(\cdot; \boldsymbol{\omega}), \mu, \upsilon) \|\nabla \varphi(\boldsymbol{\omega})\|$. In particular, for any $s \in \mathcal{S}$, if $\upsilon_s$ is the Dirac distribution on $\mathcal{S}$ centered at $s$, then $\|\nabla \phi(s, \boldsymbol{\omega})\| \leq C_{br}(\boldsymbol{\pi}(\cdot; \boldsymbol{\omega}), \mu, \upsilon_s) \|\nabla \varphi(\boldsymbol{\omega})\|$.*

*Proof.* In this proof, for any $i \in [n]$, we define $\boldsymbol{\sigma}_i(\boldsymbol{\omega}) = \boldsymbol{\rho}_i(\cdot, \boldsymbol{\pi}(\cdot; \boldsymbol{\omega}); \boldsymbol{\sigma})$ as player $i$'s policy in the policy profile $\boldsymbol{\sigma}(\boldsymbol{\omega}) = \boldsymbol{\rho}(\cdot, \boldsymbol{\pi}(\cdot; \boldsymbol{\omega}); \boldsymbol{\sigma})$. Similarly, we define $\boldsymbol{\omega}_i = \boldsymbol{\pi}_i(\cdot; \boldsymbol{\omega})$ as player $i$'s policy in the policy profile $\boldsymbol{\omega} = \boldsymbol{\pi}(\cdot; \boldsymbol{\omega})$.

Given a policy parametrization scheme $(\boldsymbol{\pi}, \boldsymbol{\rho}, \mathbb{R}^\Omega, \mathbb{R}^\Sigma)$, consider any two parameters $\boldsymbol{\omega} \in \mathbb{R}^\Omega, \boldsymbol{\sigma} \in \mathbb{R}^\Sigma$, and any two initial state distributions $\mu, \upsilon \in \Delta(\mathcal{S})$, we know that

$$\left\| \nabla_{\boldsymbol{\omega}} \psi(\upsilon, \boldsymbol{\omega}, \boldsymbol{\sigma}) \right\| \tag{25}$$

$$= \left\| \nabla_{\boldsymbol{\omega}} \sum_{i \in [n]} u_i(\boldsymbol{\sigma}_i(\boldsymbol{\omega}), \boldsymbol{\omega}_{-i}) - u_i(\boldsymbol{\omega}) \right\| \tag{26}$$

$$= \left\| \sum_{i \in [n]} \nabla_{\boldsymbol{\omega}} (u_i(\boldsymbol{\sigma}_i(\boldsymbol{\omega}), \boldsymbol{\omega}_{-i}) - u_i(\boldsymbol{\omega})) \right\| \tag{27}$$

$$= \left\| \sum_{i \in [n]} \nabla_{\boldsymbol{\omega}} \left[ \mathbb{E}_{\substack{\boldsymbol{s}' \sim \delta_\upsilon^{(\boldsymbol{\sigma}_i(\boldsymbol{\omega}), \boldsymbol{\omega}_{-i})} \\ \boldsymbol{s} \sim \delta_\upsilon^{\boldsymbol{\omega}}}} \left[ r_i(\boldsymbol{s}', \boldsymbol{\rho}_i(\boldsymbol{s}', \boldsymbol{\pi}(\boldsymbol{s}; \boldsymbol{\omega}); \boldsymbol{\sigma}), \boldsymbol{\pi}_{-i}(\boldsymbol{s}'; \boldsymbol{\omega}) - r_i(\boldsymbol{s}, \boldsymbol{\pi}(\boldsymbol{s}; \boldsymbol{\omega})) \right] \right] \right\| \tag{28}$$

$$= \left\| \sum_{i \in [n]} \mathbb{E}_{\substack{\boldsymbol{s}' \sim \delta_\upsilon^{(\boldsymbol{\sigma}_i(\boldsymbol{\omega}), \boldsymbol{\omega}_{-i})} \\ \boldsymbol{s} \sim \delta_\upsilon^{\boldsymbol{\omega}}}} \left[ \nabla_{\boldsymbol{a}_{-i}} q_i^{\boldsymbol{\sigma}_i(\boldsymbol{\omega}), \boldsymbol{\omega}_{-i}}(\boldsymbol{s}', \boldsymbol{\rho}_i(\boldsymbol{s}', \boldsymbol{\pi}(\boldsymbol{s}'; \boldsymbol{\omega}); \boldsymbol{\sigma}), \boldsymbol{\pi}_{-i}(\boldsymbol{s}'; \boldsymbol{\omega})) \nabla_{\boldsymbol{\omega}} \left( \boldsymbol{\rho}_i(\boldsymbol{s}', \boldsymbol{\pi}_{-i}(\boldsymbol{s}'; \boldsymbol{\omega}); \boldsymbol{\sigma}), \boldsymbol{\pi}(\boldsymbol{s}'; \boldsymbol{\omega}) \right) \right. \right.$$

$$\left. \left. - \nabla_{\boldsymbol{a}} q_i^{\boldsymbol{\omega}}(\boldsymbol{s}, \boldsymbol{\pi}(\boldsymbol{s}; \boldsymbol{\omega})) \nabla_{\boldsymbol{\omega}} \boldsymbol{\pi}(\boldsymbol{s}; \boldsymbol{\omega}) \right] \right\| \tag{29}$$

$$\leq \max_{i \in [n]} \max_{\boldsymbol{s}', \boldsymbol{s} \in \mathcal{S}} \frac{\delta_\upsilon^{(\boldsymbol{\sigma}_i(\boldsymbol{\omega}), \boldsymbol{\omega}_{-i})}(\boldsymbol{s}') \delta_\upsilon^{\boldsymbol{\omega}}(\boldsymbol{s})}{\delta_\mu^{(\boldsymbol{\sigma}_i(\boldsymbol{\omega}), \boldsymbol{\omega}_{-i})}(\boldsymbol{s}') \delta_\mu^{\boldsymbol{\omega}}(\boldsymbol{s})} \left\| \mathbb{E}_{\substack{\boldsymbol{s}' \sim \delta_\mu^{(\boldsymbol{\sigma}_i(\boldsymbol{\omega}), \boldsymbol{\omega}_{-i})} \\ \boldsymbol{s} \sim \delta_\mu^{\boldsymbol{\omega}}}} \left[ \nabla_{\boldsymbol{a}_{-i}} q_i^{\boldsymbol{\sigma}_i(\boldsymbol{\omega}), \boldsymbol{\omega}_{-i}}(\boldsymbol{s}', \boldsymbol{\rho}_i(\boldsymbol{s}', \boldsymbol{\pi}(\boldsymbol{s}'; \boldsymbol{\omega}); \boldsymbol{\sigma}), \boldsymbol{\pi}_{-i}(\boldsymbol{s}'; \boldsymbol{\omega})) \right. \right.$$

$$\left. \left. \nabla_{\boldsymbol{\omega}} \left( \boldsymbol{\rho}_i(\boldsymbol{s}', \boldsymbol{\pi}_{-i}(\boldsymbol{s}'; \boldsymbol{\omega}); \boldsymbol{\omega}), \boldsymbol{\pi}(\boldsymbol{s}'; \boldsymbol{\omega}) \right) - \nabla_{\boldsymbol{a}} q_i^{\boldsymbol{\omega}}(\boldsymbol{s}, \boldsymbol{\pi}(\boldsymbol{s}; \boldsymbol{\omega})) \nabla_{\boldsymbol{\omega}} \boldsymbol{\pi}(\boldsymbol{s}; \boldsymbol{\omega}) \right] \right\| \tag{30}$$

$$\leq \max_{i \in [n]} \max_{\boldsymbol{s}', \boldsymbol{s} \in \mathcal{S}} \frac{\delta_\upsilon^{(\boldsymbol{\sigma}_i(\boldsymbol{\omega}), \boldsymbol{\omega}_{-i})}(\boldsymbol{s}') \delta_\upsilon^{\boldsymbol{\omega}}(\boldsymbol{s})}{\delta_\mu^{(\boldsymbol{\sigma}_i(\boldsymbol{\omega}), \boldsymbol{\omega}_{-i})}(\boldsymbol{s}') \delta_\mu^{\boldsymbol{\omega}}(\boldsymbol{s})} \left\| \nabla_{\boldsymbol{\omega}} \left[ v_i^{\boldsymbol{\sigma}_i(\boldsymbol{\omega}), \boldsymbol{\omega}_{-i}}(\mu) - v_i^{\boldsymbol{\omega}}(\mu) \right] \right\| \tag{31}$$

$$\leq \left( \frac{1}{1-\gamma} \right)^2 \max_{i \in [n]} \max_{\boldsymbol{s}', \boldsymbol{s} \in \mathcal{S}} \frac{\delta_\upsilon^{(\boldsymbol{\sigma}_i(\boldsymbol{\omega}), \boldsymbol{\omega}_{-i})}(\boldsymbol{s}') \delta_\upsilon^{\boldsymbol{\omega}}(\boldsymbol{s})}{\mu(\boldsymbol{s}') \mu(\boldsymbol{s})} \left\| \nabla_{\boldsymbol{\omega}} \psi(\mu, \boldsymbol{\omega}, \boldsymbol{\sigma}) \right\| \tag{32}$$

$$= \left( \frac{1}{1-\gamma} \right)^2 \max_{i \in [n]} \left\| \frac{\delta_\upsilon^{(\boldsymbol{\sigma}_i(\boldsymbol{\omega}), \boldsymbol{\omega}_{-i})}}{\mu} \right\|_\infty \left\| \frac{\delta_\mu^{\boldsymbol{\omega}}}{\mu} \right\|_\infty \left\| \nabla_{\boldsymbol{\omega}} \psi(\mu, \boldsymbol{\omega}, \boldsymbol{\sigma}) \right\| \tag{33}$$

where Equation (29) and Equation (31) are obtained by deterministic policy gradient theorem Silver et al. (2014), and Equation (32) is due to the fact that $\delta_\mu^{\boldsymbol{\omega}}(\boldsymbol{s}) \geq (1-\gamma)\mu(\boldsymbol{s})$ for any $\boldsymbol{\pi} \in \mathcal{P}, \boldsymbol{s} \in \mathcal{S}$.

Given condition (1) of Assumption 5, fix any $\boldsymbol{\omega} \in \mathbb{R}^\Omega$, there exists $\boldsymbol{\sigma}^* \in \mathbb{R}^\Sigma$ s.t. for all $i \in [n]$, $\boldsymbol{s} \in \mathcal{S}$:

$$q_i^{\boldsymbol{\omega}}(\boldsymbol{s}, \boldsymbol{\rho}_i(\boldsymbol{s}, \boldsymbol{\pi}(\boldsymbol{s}; \boldsymbol{\omega}); \boldsymbol{\sigma}^*), \boldsymbol{\pi}_{-i}(\boldsymbol{s}; \boldsymbol{\omega})) = \max_{\boldsymbol{\pi}_i' \in \mathcal{F}_i(\boldsymbol{\pi}(\cdot; \boldsymbol{\omega}))} q_i^{\boldsymbol{\omega}}(\boldsymbol{s}, \boldsymbol{\pi}_i'(\boldsymbol{s}), \boldsymbol{\pi}_{-i}(\boldsymbol{s}; \boldsymbol{\omega})) .$$

Thus, $\phi(\boldsymbol{s}, \boldsymbol{\omega}) = \psi(\boldsymbol{s}, \boldsymbol{\omega}, \boldsymbol{\sigma}^*)$ for all $\boldsymbol{s} \in \mathcal{S}$. Hence, plugging in the optimal best-response policy $\boldsymbol{\sigma} = \boldsymbol{\sigma}^*$, we obtain that

$$\| \nabla_{\boldsymbol{\omega}} \phi(\upsilon, \boldsymbol{\omega}) \| \leq \left( \frac{1}{1-\gamma} \right)^2 \max_{i \in [n]} \left\| \frac{\delta_\upsilon^{(\boldsymbol{\sigma}_i^*(\boldsymbol{\omega}), \boldsymbol{\omega}_{-i})}}{\mu} \right\|_\infty \left\| \frac{\delta_\mu^{\boldsymbol{\omega}}}{\mu} \right\|_\infty \| \nabla_{\boldsymbol{\omega}} \phi(\mu, \boldsymbol{\omega}) \| \tag{34}$$

$$\leq \left( \frac{1}{1-\gamma} \right)^2 \max_{i \in [n]} \max_{\boldsymbol{\pi}_i' \in \Phi_i(\boldsymbol{\pi}_{-i}(\cdot; \boldsymbol{\omega}))} \left\| \frac{\delta_\upsilon^{(\boldsymbol{\pi}_i', \boldsymbol{\pi}_{-i}(\cdot; \boldsymbol{\omega}))}}{\mu} \right\|_\infty \left\| \frac{\delta_\mu^{\boldsymbol{\omega}}}{\mu} \right\|_\infty \| \nabla_{\boldsymbol{\omega}} \phi(\mu, \boldsymbol{\omega}) \| \tag{35}$$

where eq. (35) is due to the fact that $\boldsymbol{\sigma}_i^*(\boldsymbol{\omega}) \in \Phi_i(\boldsymbol{\pi}_{-i}(\cdot; \boldsymbol{\omega}))$.

$\square$

**Assumption 4** (Lipschitz Smooth Payoffs). *Given a Markov pseudo-game $\mathcal{M}$ and a parameterization scheme $(\boldsymbol{\pi}, \boldsymbol{\rho}, \mathbb{R}^\Omega, \mathbb{R}^\Sigma)$, assume 1. $\mathbb{R}^\Omega$ and $\mathbb{R}^\Sigma$ are non-empty, compact, and convex, 2. $\boldsymbol{\omega} \mapsto \boldsymbol{\pi}(\boldsymbol{s}; \boldsymbol{\omega})$*

*is twice continuously differentiable, for all $s \in \mathcal{S}$, and $\sigma \mapsto \rho(s, a; \sigma)$ is twice continuously differentiable, for all $(s, a) \in \mathcal{S} \times \mathcal{A}$; 3. $a \mapsto r(s, a)$ is twice continuously differentiable, for all $s \in \mathcal{S}$; 4. $a \mapsto p(s' \mid s, a)$ is twice continuously differentiable, for all $s, s' \in \mathcal{S}$.*

**Assumption 5** (Gradient Dominance Conditions). *Given a Markov pseudo-game $\mathcal{M}$ together with a parameterization scheme $(\pi, \rho, \mathbb{R}^\Omega, \mathbb{R}^\Sigma)$, assume 1. (Closure under policy improvement) For each $\omega \in \mathbb{R}^\Omega$, there exists $\sigma \in \mathbb{R}^\Sigma$ s.t. $q_i^\omega(s, \rho_i(s, \pi(s; \omega); \sigma), \pi_{-i}(s; \omega)) = \max_{\pi_i' \in \mathcal{F}_i(\pi(\cdot; \omega))} q_i^\omega(s, \pi_i'(s), \pi_{-i}(s; \omega))$ for all $i \in [n]$, $s \in \mathcal{S}$. 2. (Concavity of action-value) $\sigma \mapsto q_i^{\omega'}(s, \rho_i(s, \pi_{-i}(s; \omega); \sigma), \pi_{-i}(s; \omega))$ is concave, for all $s \in \mathcal{S}$ and $\omega, \omega' \in \mathbb{R}^\Omega$.*

**Theorem 2.2** (Detailed Statement). *Given an MPG $\mathcal{M}$ and a parameterization scheme $(\pi, \rho, \mathbb{R}^\Omega, \mathbb{R}^\Sigma)$ satisfying Assumption 1, assume Assumptions 2, 4, and 5 hold. For any $\delta > 0$, set $\varepsilon = \delta \|C_{br}(\cdot, \mu, \cdot)\|_\infty^{-1}$. If Algorithm 1 is run with inputs that satisfy for all $t \in [T]$,*

$$\eta_\omega \asymp \frac{\epsilon^4 \left( \left\| \partial \delta_\mu^{\pi^*} / \partial \mu \right\|_\infty / 1 - \gamma \right)^2}{\ell_{\nabla\Psi}^3 \left( \ell_{\nabla\Psi}^2 + \varsigma_\sigma^2 \right) \left( \ell_\Psi / \ell_{\nabla\Psi} + 1 \right)^2}, \quad \eta_\sigma \asymp \frac{\varepsilon^8 \left( \left\| \partial \delta_\mu^{\pi^*} / \partial \mu \right\|_\infty / 1 - \gamma \right)^4}{\ell_{\nabla\Psi}^5 \ell_\Psi \left( \ell_\Psi / \ell_{\nabla\Psi} + 1 \right)^4 \left( \ell_\Psi^2 + \varsigma_\sigma^2 \right)^{3/2}} \wedge \frac{\varepsilon^2}{\ell_{\nabla\Psi} \left( \ell_\Psi^2 + \varsigma_\sigma^2 \right)}, \text{ and } T \geq$$

$$\left( (1-\gamma) / \left\| \partial \delta_\mu^{\pi^*} / \partial \mu \right\|_\infty + \ell_\Psi / 2 \ell_{\nabla\Psi} \right)^{-1} \frac{\ell_\Psi \mathrm{diam}(\mathbb{R}^\Omega \times \mathbb{R}^\Sigma)}{\varepsilon^2 \eta_\omega}, \text{ then there exists } k \leq T \text{ s.t. } \omega_{\mathrm{best}}^{(T)} = \omega^{(k)} \text{ is an}$$

*$(\varepsilon, \varepsilon/2\ell_\Psi)$-stationary point of exploitability, i.e., there exists $\omega^* \in \mathbb{R}^\Omega$ s.t. $\|\omega_{\mathrm{best}}^{(T)} - \omega^*\| \leq \varepsilon/2\ell_\Psi$ and $\min_{h \in \mathcal{D}\varphi(\omega^*)} \|h\| \leq \varepsilon$. And, for any distribution $\upsilon \in \Delta(\mathcal{S})$, if $\phi(\upsilon, \cdot)$ is differentiable at $\omega^*$, then $\|\nabla_\omega \varphi(\upsilon, \omega^*)\| \leq \delta$, i.e., $\omega_{\mathrm{best}}^{(T)}$ is an $(\varepsilon, \delta)$-stationary point of expected state exploitability $\phi(\upsilon, \cdot)$.*

*Proof.* As is common in the optimization literature (see, for instance, Davis et al. (2018)), we consider the Moreau envelope of the exploitability, which we simply call the *Moreau exploitability*, i.e.,

$$\tilde{\varphi}(\omega) \doteq \min_{\omega' \in \mathbb{R}^\Omega} \left\{ \varphi(\omega') + \ell_{\nabla\psi} \|\omega - \omega'\|^2 \right\} \ .$$

Similarly, we also consider the *state Moreau exploitability*, i.e., the Moreau envelope of the state exploitability:

$$\tilde{\phi}(s, \omega) \doteq \min_{\omega' \in \mathbb{R}^\Omega} \left\{ \phi(s, \omega') + \ell_{\nabla\psi} \|\omega - \omega'\|^2 \right\} \ .$$

We recall that in these definitions, by our notational convention, $\ell_{\nabla\psi} \geq 0$, refers to the Lipschitz-smoothness constants of the state exploitability which in this case we take to be the largest across all states, i.e., for all $s \in \mathcal{S}$, $(\omega, \sigma) \mapsto \psi(s, \omega, \sigma)$ is $\ell_{\nabla\psi}$-Lipschitz-smooth, respectively, and which we note is guaranteed to exist under Assumption 4. Further, we note that since $\Psi(\omega, \sigma) = \mathbb{E}_{s \sim \mu} [\psi(s, \omega, \sigma)]$ is a weighted average of $\psi$, $(\omega, \sigma) \mapsto \Psi(\omega, \sigma)$ is also $\ell_{\nabla\psi}$-Lipschitz-smooth.

We invoke Theorem 2 of Daskalakis et al. (2020). Although their result is stated for gradient-dominated-gradient-dominated functions, their proof applies in the more general case of non-convex-gradient-dominated functions.

First, Assumption 4 guarantees that the cumulative regret $\Psi$ is Lipschitz-smooth w.r.t. $(\omega, \sigma)$. Moreover, under Assumption 4, which guarantees that $\sigma \mapsto q_i^{\omega'}(s, \rho_i(s, \pi_{-i}(s; \omega); \sigma), \pi_{-i}(s; \omega))$ is continuously differentiable for all $s \in \mathcal{S}$ and $\omega, \omega' \in \mathbb{R}^\Omega$, and Assumption 5, we have that $\Psi$ is $\left( \left\| \partial \delta_\mu^{\pi^*} / \partial \mu \right\|_\infty / 1 - \gamma \right)$-gradient-dominated in $\sigma$, for all $\omega \in \mathbb{R}^\Omega$, by Theorems 2 and 4 of Bhandari & Russo (2019). Finally, under Assumption 4, since the policy, the reward function, and the transition probability function are all Lipschitz-continuous, $\widehat{u}$, $\widehat{\Psi}$, and hence $\widehat{G}$ are also Lipschitz-continuous, since $\mathcal{S}$ and $\mathcal{A}$ are compact. Their variance must therefore be bounded, i.e., there exists $\varsigma_\omega, \varsigma_\sigma \in \mathbb{R}$ s.t. $\mathbb{E}_{h, h'}[\widehat{G_\omega}(\omega, \sigma; h, h') - \nabla_\omega \Psi(\omega, \sigma; h, h')] \leq \varsigma_\omega$ and $\mathbb{E}_{h, h'}[\widehat{G_\sigma}(\omega, \sigma; h, h') - \nabla_\sigma \Psi(\omega, \sigma; h, h')] \leq \varsigma_\sigma$.

Hence, under our assumptions, the assumptions of Theorem 2 of Daskalakis et al. are satisfied. Therefore, $1/T+1 \sum_{t=0}^T \|\nabla \tilde{\varphi}(\omega^{(t)})\| \leq \varepsilon$. Taking a minimum across all $t \in [T]$, we conclude $\left\| \nabla \widetilde{\varphi}(\omega_{\mathrm{best}}^{(T)}) \right\| \leq \varepsilon$.

Then, by the Lemma 3.7 of Lin et al. (2020), there exists some $\boldsymbol{\omega}^* \in \mathbb{R}^\Omega$ such that $\|\boldsymbol{\omega}_{\text{best}}^{(T)} - \boldsymbol{\omega}^*\| \leq \frac{\varepsilon}{2\ell_\Psi}$ and $\boldsymbol{\omega}^* \in \mathbb{R}_\varepsilon^\Omega \doteq \{\boldsymbol{\omega} \in \mathbb{R}^\Omega \mid \exists \alpha \in \mathcal{D}\varphi(\boldsymbol{\omega}), \|\alpha\| \leq \varepsilon\}$. That is, $\boldsymbol{\omega}_{\text{best}}^{(T)}$ is a $(\varepsilon, \frac{\varepsilon}{2\ell_\Psi})$-stationary point of $\varphi$.

Furthermore, if we assume that $\phi(\delta, \cdot)$ is differentiable at $\boldsymbol{\omega}^*$ for any state distribution $\delta \in \Delta(\mathcal{S})$, $\varphi$ is also differentiable at $\boldsymbol{\omega}^*$. Hence, by the proof of Lemma 5, we know that for any state distribution $\upsilon \in \Delta(\mathcal{S})$,

$$\|\nabla_{\boldsymbol{\omega}} \phi(\upsilon, \boldsymbol{\omega})\| \leq \max_{\boldsymbol{\sigma}^* \in \arg\max_{\boldsymbol{\sigma} \in \mathbb{R}^\Sigma} \psi(\upsilon, \boldsymbol{\omega}, \boldsymbol{\sigma})} \|\nabla_{\boldsymbol{\omega}} \psi(\upsilon, \boldsymbol{\omega}, \boldsymbol{\sigma}^*)\| \tag{36}$$

$$\leq \max_{i \in [n]} \max_{\boldsymbol{\sigma}^* \in \arg\max_{\boldsymbol{\sigma} \in \mathbb{R}^\Sigma} \psi(\upsilon, \boldsymbol{\omega}, \boldsymbol{\sigma})} \tag{37}$$

$$\left(\frac{1}{1-\gamma}\right)^2 \left\|\frac{\delta_\upsilon^{\boldsymbol{\sigma}_i^*(\boldsymbol{\omega}), \boldsymbol{\omega}_{-i}}}{\mu}\right\|_\infty \left\|\frac{\delta_\upsilon^{\boldsymbol{\omega}}}{\mu}\right\|_\infty \|\nabla_{\boldsymbol{\omega}} \Psi(\boldsymbol{\omega}, \boldsymbol{\sigma}^*)\| \tag{38}$$

$$= C_{br}(\boldsymbol{\omega}, \mu, \upsilon) \|\nabla_{\boldsymbol{\omega}} \Psi(\boldsymbol{\omega}, \boldsymbol{\sigma}^*)\| \tag{39}$$

$$\frac{1}{C_{br}(\boldsymbol{\omega}, \mu, \upsilon)} \|\nabla_{\boldsymbol{\omega}} \phi(\upsilon, \boldsymbol{\omega})\| \leq \|\nabla_{\boldsymbol{\omega}} \Psi(\boldsymbol{\omega}, \boldsymbol{\sigma}^*)\| \tag{40}$$

Therefore,

$$\boldsymbol{\omega}^* \in \mathbb{R}_\varepsilon^\Omega \doteq \{\boldsymbol{\omega} \in \mathbb{R}^\Omega \mid \exists \alpha \in \mathcal{D}\varphi(\boldsymbol{\omega}), \|\alpha\| \leq \varepsilon\} \tag{41}$$

$$\supseteq \{\boldsymbol{\omega} \in \mathbb{R}^\Omega \mid \exists \boldsymbol{\sigma}^* \in \arg\max_{\boldsymbol{\omega} \in \mathbb{R}^\Omega} \Psi(\boldsymbol{\omega}, \boldsymbol{\sigma}) s.t. \|\nabla_{\boldsymbol{\omega}} \Psi(\boldsymbol{\omega}, \boldsymbol{\sigma}^*)\| \leq \varepsilon\} \tag{42}$$

$$\supseteq \{\boldsymbol{\omega} \in \mathbb{R}^\Omega \mid {}^1/_{C_{br}(\boldsymbol{\omega}, \mu, \upsilon)} \|\nabla_{\boldsymbol{\omega}} \phi(\upsilon, \boldsymbol{\omega})\| \leq \varepsilon\} \tag{43}$$

$$= \{\boldsymbol{\omega} \in \mathbb{R}^\Omega \mid \|\nabla_{\boldsymbol{\omega}} \phi(\upsilon, \boldsymbol{\omega})\| \leq \delta\} \tag{44}$$

Therefore, we can conclude that there exists $\boldsymbol{\omega}^*$ such that $\|\boldsymbol{\omega}_{\text{best}}^{(T)} - \boldsymbol{\omega}^*\| \leq \frac{\varepsilon}{2\ell_\Psi}$ and $\|\nabla_{\boldsymbol{\omega}} \phi(\upsilon, \boldsymbol{\omega})\| \leq \delta$ for any $\upsilon$. Thus, $\boldsymbol{\omega}_{\text{best}}^{(T)}$ is a $(\varepsilon, \delta)$-stationary point of $\phi(\upsilon, \cdot)$ for any $\upsilon \in \Delta(\mathcal{S})$.

$\square$

## D.2 Omitted Assumptions, Results, and Proofs from Section 3

**Assumption 6.** *Given an infinite horizon Markov exchange economy $\mathcal{I}$, assume for all $i \in [n]$,*

1. *$\mathcal{X}$, $\mathcal{Y}$, $\mathcal{E}$, are non-empty, closed, convex, with $\mathcal{E}$ additionally bounded;*

2. *$(\boldsymbol{\theta}_i, \boldsymbol{x}_i) \mapsto r_i(\boldsymbol{x}_i; \boldsymbol{\theta}_i)$ is continuous and concave, and $(\boldsymbol{s}, \boldsymbol{y}_i) \mapsto p(\boldsymbol{s}' \mid \boldsymbol{s}, \boldsymbol{y}_i, \boldsymbol{y}_{-i})$ is continuous and stochastically concave, for all $\boldsymbol{s}' \in \mathcal{S}$ and $\boldsymbol{y}_{-i} \in \mathcal{Y}_{-i}$;*

3. *for all $\boldsymbol{e}_i \in \mathcal{E}_i$, the correspondence*

$$(\boldsymbol{p}, \boldsymbol{q}) \rightrightarrows \mathcal{B}_i(\boldsymbol{e}_i, \boldsymbol{p}, \boldsymbol{q}) \cap \left\{ (\boldsymbol{x}_i, \boldsymbol{y}_i) \,\middle|\, \sum_{i \in [n]} \boldsymbol{x}_i \leq \sum_{i \in [n]} \boldsymbol{e}_i, \sum_{i \in [n]} \boldsymbol{y}_i \leq \boldsymbol{0}_m, (\boldsymbol{X}, \boldsymbol{Y}) \in \mathcal{X} \times \mathcal{Y} \right\}$$

   *is continuous[21] and non-empty, convex, and compact, for all $\boldsymbol{p} \in \Delta_m$ and $\boldsymbol{q} \in \mathbb{R}^l$;[22]*

4. *(no saturation) there exists an $\boldsymbol{x}_i^+ \in \mathcal{X}_i$ s.t. $r_i(\boldsymbol{x}_i^+; \boldsymbol{\theta}_i) > r_i(\boldsymbol{x}_i; \boldsymbol{\theta}_i)$, for all $\boldsymbol{x}_i \in \mathcal{X}_i$ and $\boldsymbol{\theta}_i \in \mathcal{T}_i$.*

---

[21]One way to ensure this condition holds is to assume that for all $\boldsymbol{s} = (o, \boldsymbol{E}, \boldsymbol{\Theta}) \in \mathcal{S}$, returns from assets are positive, i.e., $\boldsymbol{R}_o \geq \boldsymbol{0}_{ml}$, and for all consumers $i \in [n]$, there exists $(\boldsymbol{x}_i, \boldsymbol{y}_i) \in \mathcal{X}_i \times \mathcal{Y}_i$, s.t. $\boldsymbol{x}_i < \boldsymbol{e}_i$ and $\boldsymbol{y}_i < 0$.

[22]One way to ensure this condition holds is to assume that for all $\boldsymbol{s} = (o, \boldsymbol{E}, \boldsymbol{\Theta}) \in \mathcal{S}$, returns from assets are positive, i.e., $\boldsymbol{R}_o \geq \boldsymbol{0}_{ml}$, and $\mathcal{X}, \mathcal{Y}$ are bounded from below.

5. *(uniform solvency / borrowing limit) For each consumer $i \in [n]$, there exists a constant $0 < M_i < \infty$ such that for all admissiable (normalized) asset prices $\boldsymbol{q} \in \mathcal{Q} \doteq [0, \max_{\boldsymbol{E} \in \mathcal{E}} \sum_{i \in [n]} \sum_{j \in [m]} e_{ij}]^l$ and all portfolio $\boldsymbol{y}_i \in \mathcal{Y}_i$,*

$$\boldsymbol{y}_i \cdot \boldsymbol{q} \geq -M_i.$$

*That is, the market value of asset positions is uniformly bounded below.*

**Theorem 3.1.** *Consider an infinite horizon MEE $\mathcal{I}$. Under Assumption 6, the set of RRE of $\mathcal{I}$ is equal to the set of GMPE of the associated exchange economy MPG $\mathcal{M}$.*

*Proof.* Let $\boldsymbol{\pi}^* = (\boldsymbol{X}^*, \boldsymbol{Y}^*, \boldsymbol{p}^*, \boldsymbol{q}^*) : \mathcal{S} \to \mathcal{X} \times \mathcal{Y} \times \mathcal{P} \times \mathcal{Q}$ be an GMPE of the Radner Markov pseudo-game $\mathcal{M}$ associated with $\mathcal{I}$. We want to show that it is also an RRE of $\mathcal{I}$.

First, we want to show that $\boldsymbol{\pi}^*$ is Markov perfect for all consumers. We can make some easy observations: the state value for the player $i \in [n]$ in the Radner Markov pseudo-game at state $\boldsymbol{s} \in \mathcal{S}$ induced by the policy $\boldsymbol{\pi}^*$

$$v_i^{\boldsymbol{\pi}^*}(\boldsymbol{s}) = \mathop{\mathbb{E}}_{H \sim \nu^{\boldsymbol{\pi}^*}} \left[ \sum_{t=0}^{\infty} \gamma^t r'(S^{(t)}, A^{(t)}) \mid S^{(0)} = \boldsymbol{s} \right] \tag{45}$$

$$= \mathop{\mathbb{E}}_{H \sim \nu^{\boldsymbol{\pi}^*}} \left[ \sum_{t=0}^{\infty} \gamma^t r_i(\boldsymbol{x}_i^*(S^{(t)}); \Theta_i^{(t)}) \mid S^{(0)} = \boldsymbol{s} \right] \tag{46}$$

is equal to the consumption state value induced by $(\boldsymbol{X}^*, \boldsymbol{Y}^*, \boldsymbol{p}^*, \boldsymbol{q}^*)$

$$v_i^{(\boldsymbol{X}^*, \boldsymbol{Y}^*, \boldsymbol{p}^*, \boldsymbol{q}^*)}(\boldsymbol{s}) \doteq \mathop{\mathbb{E}}_{H \sim \nu^{(\boldsymbol{X}^*, \boldsymbol{Y}^*, \boldsymbol{p}^*, \boldsymbol{q}^*)}} \left[ \sum_{t=0}^{\infty} \gamma^t r_i \left( \boldsymbol{x}_i^*(H_{:t}); \Theta^{(t)} \right) \mid S^{(0)} = \boldsymbol{s} \right] . \tag{47}$$

as $\boldsymbol{x}_i^*$ is Markov. Since $\boldsymbol{\pi}^*$ is a GMPE, we know that for any $i \in [n]$:

$$(\boldsymbol{x}_i^*, \boldsymbol{y}_i^*) \in \mathop{\arg\max}_{\substack{(\boldsymbol{x}_i, \boldsymbol{y}_i): \mathcal{S} \to \mathcal{X}_i \times \mathcal{Y}_i : \forall \boldsymbol{s} \in \mathcal{S}, \\ (\boldsymbol{x}_i, \boldsymbol{y}_i)(\boldsymbol{s}) \in \mathcal{B}_i(\boldsymbol{e}_i, \boldsymbol{p}^*(\boldsymbol{s}), \boldsymbol{q}^*(\boldsymbol{s}))}} \left\{ v_i^{(\boldsymbol{x}_i, \boldsymbol{x}_{-i}^*, \boldsymbol{y}_i, \boldsymbol{y}_{-i}^*, \boldsymbol{p}^*, \boldsymbol{q}^*)}(\boldsymbol{s}) \right\}$$

for all $\boldsymbol{s} \in \mathcal{S}$, so $(\boldsymbol{X}^*, \boldsymbol{Y}^*, \boldsymbol{p}^*, \boldsymbol{q}^*)$ is Markov perfect.

Next, we want to show that $(\boldsymbol{X}^*, \boldsymbol{Y}^*, \boldsymbol{p}^*, \boldsymbol{q}^*)$ satisfies the Walras's law. First, we show that for any $i \in [n]$, $\boldsymbol{s} \in \mathcal{S}$, $\boldsymbol{x}_i^*(\boldsymbol{s}) \cdot \boldsymbol{p}^*(\boldsymbol{s}) + \boldsymbol{y}_i^*(\boldsymbol{s}) \cdot \boldsymbol{q}^*(\boldsymbol{s}) - \boldsymbol{e}_i \cdot \boldsymbol{p}^*(\boldsymbol{s}) = 0$. By way of contradiction, assume that there exists some $i \in [n]$, $\boldsymbol{s} \in \mathcal{S}$ such that $\boldsymbol{x}_i^*(\boldsymbol{s}) \cdot \boldsymbol{p}^*(\boldsymbol{s}) + \boldsymbol{y}_i^*(\boldsymbol{s}) \cdot \boldsymbol{q}^*(\boldsymbol{s}) - \boldsymbol{e}_i \cdot \boldsymbol{p}^*(\boldsymbol{s}) \neq 0$. Note that $(\boldsymbol{x}_i^*(\boldsymbol{s}), \boldsymbol{y}_i^*(\boldsymbol{s})) \in \mathcal{B}'(\boldsymbol{s}, \boldsymbol{a}_{-i}) = \mathcal{B}(\boldsymbol{e}_i, \boldsymbol{p}^*(\boldsymbol{s}), \boldsymbol{q}^*(\boldsymbol{s})) = \{(\boldsymbol{x}_i, \boldsymbol{y}_i) \in \mathcal{X}_i \times \mathcal{Y}_i \mid \boldsymbol{x}_i \cdot \boldsymbol{p}^*(\boldsymbol{s}) + \boldsymbol{y}_i \cdot \boldsymbol{q}^*(\boldsymbol{s}) \leq \boldsymbol{e}_i \cdot \boldsymbol{p}^*(\boldsymbol{s})\}$, so we must have $\boldsymbol{x}_i^*(\boldsymbol{s}) \cdot \boldsymbol{p}^*(\boldsymbol{s}) + \boldsymbol{y}_i^*(\boldsymbol{s}) \cdot \boldsymbol{q}^*(\boldsymbol{s}) - \boldsymbol{e}_i \cdot \boldsymbol{p}^*(\boldsymbol{s}) < 0$. By the (no saturation) condition of Assumption 6, there exists $\boldsymbol{x}_i^+ \in \mathcal{X}_i$ s.t. $r_i(\boldsymbol{x}_i^+; \boldsymbol{\theta}_i) > r_i(\boldsymbol{x}_i^*(\boldsymbol{s}); \boldsymbol{\theta}_i)$. Moreover, since $\boldsymbol{x}_i \mapsto r_i(\boldsymbol{x}_i; \boldsymbol{\theta}_i)$ is concave, for any $0 < t < 1$, $r_i(t\boldsymbol{x}_i^+ + (1-t)\boldsymbol{x}_i^*(\boldsymbol{s}); \boldsymbol{\theta}_i) > r_i(\boldsymbol{x}_i^*(\boldsymbol{s}); \boldsymbol{\theta}_i)$. Since $\boldsymbol{x}_i^*(\boldsymbol{s}) \cdot \boldsymbol{p}^*(\boldsymbol{s}) + \boldsymbol{y}_i^*(\boldsymbol{s}) \cdot \boldsymbol{q}^*(\boldsymbol{s}) - \boldsymbol{e}_i \cdot \boldsymbol{p}^*(\boldsymbol{s}) < 0$, we can pick $t$ small enough such that $\boldsymbol{x}_i' = t\boldsymbol{x}_i^+ + (1-t)\boldsymbol{x}_i^*(\boldsymbol{s})$ satisfies $\boldsymbol{x}_i'(\boldsymbol{s}) \cdot \boldsymbol{p}^*(\boldsymbol{s}) + \boldsymbol{y}_i^*(\boldsymbol{s}) \cdot \boldsymbol{q}^*(\boldsymbol{s}) - \boldsymbol{e}_i \cdot \boldsymbol{p}^*(\boldsymbol{s}) \leq 0$ but $\boldsymbol{x}_i' \in \mathcal{X}_i$ s.t. $r_i(\boldsymbol{x}_i^+; \boldsymbol{\theta}_i) > r_i(\boldsymbol{x}_i^*(\boldsymbol{s}); \boldsymbol{\theta}_i)$. Thus,

$$q_i^{\boldsymbol{\pi}^*}(\boldsymbol{s}, \boldsymbol{x}_i', \boldsymbol{x}_{-i}^*(\boldsymbol{s}), \boldsymbol{Y}^*(\boldsymbol{s}), \boldsymbol{p}^*(\boldsymbol{s}), \boldsymbol{q}^*(\boldsymbol{s})) \tag{48}$$

$$= r_i'(\boldsymbol{s}, \boldsymbol{x}_i', \boldsymbol{x}_{-i}^*(\boldsymbol{s}), \boldsymbol{Y}^*(\boldsymbol{s}), \boldsymbol{p}^*(\boldsymbol{s}), \boldsymbol{q}^*(\boldsymbol{s})) + \mathop{\mathbb{E}}_{S' \sim p(S' | \boldsymbol{s}, \boldsymbol{Y}^*(\boldsymbol{s}))} [\gamma v_i^{\boldsymbol{\pi}^*}(S')] \tag{49}$$

$$= r_i(\boldsymbol{x}_i'; \boldsymbol{\theta}_i) + \mathop{\mathbb{E}}_{S' \sim p(S' | \boldsymbol{s}, \boldsymbol{Y}^*(\boldsymbol{s}))} [\gamma v_i^{\boldsymbol{\pi}^*}(S')] \tag{50}$$

$$> r_i(\boldsymbol{x}_i^*(\boldsymbol{s}); \boldsymbol{\theta}_i) + \mathop{\mathbb{E}}_{S' \sim p(S' | \boldsymbol{s}, \boldsymbol{Y}^*(\boldsymbol{s}))} [\gamma v_i^{\boldsymbol{\pi}^*}(S')] \tag{51}$$

$$= q_i^{\boldsymbol{\pi}^*}(\boldsymbol{s}, \boldsymbol{X}^*(\boldsymbol{s}), \boldsymbol{Y}^*(\boldsymbol{s}), \boldsymbol{p}^*(\boldsymbol{s}), \boldsymbol{q}^*(\boldsymbol{s})) \tag{52}$$

This contradicts that fact that $\boldsymbol{\pi}^*$ is a GMPE since an optimal policy is supposed to be greedy optimal (i.e., maximize the action-value function of each player over its action space at all states) respect to optimal action value function. Thus, we know that for all $i \in [n]$, $\boldsymbol{s} \in \mathcal{S}$, $\boldsymbol{x}_i^*(\boldsymbol{s}) \cdot \boldsymbol{p}^*(\boldsymbol{s}) + \boldsymbol{y}_i^*(\boldsymbol{s}) \cdot$

$q^*(s) - e_i \cdot p^*(s) = 0$. Summing across the buyers, we get $p^*(s) \cdot \left( \sum_{i \in [n]} x_i^*(s) - \sum_{i \in [n]} e_i \right) + q^*(s) \cdot \left( \sum_{i \in [n]} y_i^*(s) \right) = 0$ for any $s \in \mathcal{S}$, which is the Walras' law.

Finally, we want to show that $(X^*, Y^*, p^*, q^*)$ is feasible. We first show that $\sum_{i \in [n]} x_i^*(s) - \sum_{i \in [n]} e_i \leq \mathbf{0}_m$ for any $s \in \mathcal{S}$. We proved that for any state $s \in \mathcal{S}$, $r'_{n+1}(s, X^*(s), Y^*(s), p^*(s), q^*(s)) = p^*(s) \cdot \left( \sum_{i \in [n]} x_i^*(s) - \sum_{i \in [n]} e_i \right) + q^*(s) \cdot \left( \sum_{i \in [n]} y_i^*(s) \right) = 0$, which implies $v_{n+1}^{\pi^*}(s) = 0$. For any $j \in [m]$, consider a $p : \mathcal{S} \to \mathcal{P}$ defined by $p(s) = j_j$ for all $s \in \mathcal{S}$ and a $q : s \to \mathcal{Q}$ defined by $q(s) = \mathbf{0}_l$ for all $s \in \mathcal{S}$. Then, we know that

$$0 = v_{n+1}^{\pi^*} \tag{53}$$

$$= q_{n+1}^{\pi^*}(s, X^*(s), Y^*(s), p^*(s), q^*(s)) \tag{54}$$

$$\geq q_{n+1}^{\pi^*}(s, X^*(s), Y^*(s), p(s), q(s)) \tag{55}$$

$$= r'_{n+1}(s, X^*(s), Y^*(s), p(s), q(s)) + \mathop{\mathbb{E}}_{S' \sim p(S'|s, Y^*(s))} [\gamma v_i^{\pi^*}(S')] \tag{56}$$

$$= j_j \cdot \left( \sum_{i \in [n]} x_i^*(s) - \sum_{i \in [n]} e_i \right) \qquad \forall j \in [m] \tag{57}$$

$$= \sum_{i \in [n]} x_{ij}^*(s) - \sum_{i \in [n]} e_{ij} \qquad \forall j \in [m] \tag{58}$$

Thus, we know that $\sum_{i \in [n]} x_i^*(s) - \sum_{i \in [n]} e_i \leq \mathbf{0}_m$ for any $s \in \mathcal{S}$. Finally, we show that $\sum_{i \in [n]} y_i^*(s) \leq \mathbf{0}_l$ for all $s \in \mathcal{S}$. By way of contradiction, suppose that for some asset $k \in [l]$, and some state $s \in \mathcal{S}$, $\sum_{i \in [n]} y_{ik}^*(s) > 0$. Then, the auctioneer can increase its cumulative payoff by increasing $q_k^*(s)$, which contradicts the definition of a GMPE.

Therefore, we can conclude that $\pi^* = (X^*, Y^*, p^*, q^*) : \mathcal{S} \to \mathcal{X} \times \mathcal{Y} \times \mathcal{P} \times \mathcal{Q}$ is a RRE of $\mathcal{I}$.

Moreover, in an infinite-horizon Markov economies economy, agents may a priori finance consumption via a Ponzi scheme by rolling over an ever-growing debt position. Assumption 6.(5) rules out this pathology by imposing a *uniform solvency bound*: for each consumer $i \in [n]$, the market value of any admissible portfolio is bounded below uniformly over admissible asset prices $q \in \mathcal{Q}$, i.e., $q \cdot y_i \geq -M_i$ for all $y_i \in \mathcal{Y}_i$. As a consequence, along any feasible (in particular equilibrium) path we have $\gamma^T q(s^{(T)}) \cdot y_i(s^{(T)}) \geq -\gamma^T M_i \to 0$, which is the standard transversality/no-Ponzi implication in the discounted infinite-horizon setting. This ensures the equilibrium concept excludes Ponzi-financed allocations and allows the existence argument to proceed without degenerate unbounded-debt strategies. $\square$

**Corollary 2.** *Under Assumption 6, the set of RRE of an infinite horizon MEE is non-empty.*

*Proof.* For any infinite horizon Markov exchange economy $\mathcal{I}$ for which Assumption 6 holds, consider the associated exchange economy Markov pseudo-game $\mathcal{M}$. By the definition of exchange economy Markov pseudo game, we can see that the transition functions set in the game are all stochastically concave and as such give rise action-value functions which are concave in the actions each of player Atakan (2003a), and it is easy to verify that the game also satisfies all conditions that guarantee the existence of a GMPE (see Section 4 of Atakan (2003a) for detailed proofs). Hence, by Theorem 2.1 which guarantees the existence of GMPE in Markov pseudo-game, we can conclude that there exists an RRE $(X^*, Y^*, p^*, q^*)$ in any Radner economy $\mathcal{I}$. $\square$

**Theorem 3.2.** *Given an infinite horizon MEE $\mathcal{I}$ for which Assumption 6 holds, and the associated exchange economy MPG $\mathcal{M}$. If $(\pi, \rho, \mathbb{R}^\Omega, \mathbb{R}^\Sigma)$ is a parametrization scheme for $\mathcal{M}$ such that Assumptions 4 and 5 hold, then the convergence results in Theorem 2.2 hold, meaning Algorithm 1 converges to a point in the neighborhood of a point that approximately satisfies the necessary conditions of a GMPE in $\mathcal{M}$, which is likewise a point that approximately satisfies the necessary*

*conditions of an RRE of $\mathcal{I}$. Moreover, beyond its finite-time guarantees, in the limit, Algorithm 1 converges to a point that satisfies these conditions exactly.*

*Proof.* In the proof of Theorem 3.1, we can observe that, for any infinite horizon Markov exchange economy $\mathcal{I}$ and its associated exchange economy Markov pseudo-game $\mathcal{M}$, the exploitability of an outcome $(\boldsymbol{X}, \boldsymbol{Y}, \boldsymbol{p}, \boldsymbol{q})$ in $\mathcal{I}$ is equivalent to the exploitability of a policy $\boldsymbol{\pi} = (\boldsymbol{X}, \boldsymbol{Y}, \boldsymbol{p}, \boldsymbol{q})$ in $\mathcal{M}$. Similarly, the state exploitability of an outcome $(\boldsymbol{X}, \boldsymbol{Y}, \boldsymbol{p}, \boldsymbol{q})$ in $\mathcal{I}$ is equivalent to the state exploitability of a policy $\boldsymbol{\pi} = (\boldsymbol{X}, \boldsymbol{Y}, \boldsymbol{p}, \boldsymbol{q})$ in $\mathcal{M}$ given any state $\boldsymbol{s} \in \boldsymbol{s}$.

Therefore, this results follows readily from Theorem 2.2. $\qquad\square$

## E  EXPERIMENTS

### E.1  NEURAL PROJECTION METHOD

The projection method Judd (1992), also known as the weighted residual methods, is a numerical technique often used to approximate solutions to complex economic models, particularly those involving dynamic programming and dynamic stochastic general equilibrium (DSGE) models. These models are common in macroeconomics and often don't have analytical solutions due to their non-linear, dynamic, and high-dimensional nature. The projection method helps approximate these solutions by projecting the problem into a more manageable, lower-dimensional space.

The main idea of the projection method is to express equilibrium of the dynamic economic model as a solution to a functional equation $D(f) = \mathbf{0}$, where $f : \mathcal{S} \to \mathbb{R}^m$ is a function that represent some unknown policy, $D : (\mathcal{S} \to \mathbb{R}^m) \to (\mathcal{S} \to \mathbb{R}^n)$, and $\mathbf{0}$ is the constant zero function. Some classic examples of the operator $D$ includes Euler equations and Bellman equations. A canonical project method consists of four steps: 1) Define a set of basis functions $\{\psi_i : \mathcal{S} \to \mathbb{R}^m\}_{i \in [n]}$ and approximate each each function $f \in \mathcal{F}$ through a linear combination of basis functions: $\hat{f}(\cdot; \boldsymbol{\theta}) = \sum_{i=1}^n \theta_i \psi_i(\cdot)$; 2) Define a residual equation as a functional equation evaluated at the approximation: $R(\cdot; \boldsymbol{\theta}) \doteq D(\hat{f}(\cdot; \boldsymbol{\theta}))$; 3) Choose some weight functions $\{w_i : \mathcal{S} \to \mathbb{R}\}_{i \in [p]}$ over the states and find $\boldsymbol{\theta}$ that solves $F(\boldsymbol{\theta}) \doteq \int_{\mathcal{S}} w_i(\boldsymbol{s}) R(\boldsymbol{s}; \boldsymbol{\theta}) d\boldsymbol{s} = 0$ for all $i \in [p]$. This gets the residual "close" to zero in the weighted integral sense; 4) Simulate the optimal decision rule based on the chosen parameter $\boldsymbol{\theta}$ and basis functions.

Recently, the neural projection method was developed to extend the traditional projection method Maliar et al. (2021); Azinovic et al. (2022); Sauzet (2021). In the neural projection method, neural networks are used as the functional approximators for policy functions instead of traditional basis function approximations. In this section, we show how we can approximate generalized Markov perfect equilibrium of Markov pseudo-game, and consequently Recursive Radner Equilibrium of infinite-horizon Markov exchange economies, through the neural projection method.

**Assumption 7.** *Given a Markov pseudo-game $\mathcal{M}$, assume that 1. for any $i \in [n]$, $\boldsymbol{s} \in \mathcal{S}$, $\boldsymbol{a}_{-i} \in \mathcal{A}_{-i}$, $\mathcal{X}_i(\boldsymbol{s}, \boldsymbol{a}_{-i}) \doteq \{\boldsymbol{a}_i \in \mathcal{A}_i \mid h_{ic}(\boldsymbol{s}, \boldsymbol{a}_i, \boldsymbol{a}_{-i}) \geq 0 \text{ for all } c \in [d]\}$ for a collection of constraint functions $\{h_{ic} : \mathcal{S} \times \mathcal{A} \mid c \in [d]\}$, where $\boldsymbol{a}_i \mapsto h_{ic}(\boldsymbol{s}, \boldsymbol{a}_i, \boldsymbol{a}_{-i})$ is concave for every $c \in [d]$.*

**Theorem E.1.** *Let $\mathcal{M}$ be a Markov pseudo-game that satisfies Assumption 7. For any policy profile $\boldsymbol{\pi} \in \mathcal{F}^{\mathrm{markov}}$, $\boldsymbol{\pi}$ is a GMPE if and only if there exists Lagrange multiplier policy $\boldsymbol{\lambda} : \mathcal{S} \to \mathbb{R}_+^{n \times d}$ such that $(\boldsymbol{\pi}, \boldsymbol{\lambda})$ solves the following functional equation: for all $i \in [n]$, $\boldsymbol{s} \in \mathcal{S}$,*

$$0 \in \partial_{\boldsymbol{a}_i} q_i^{\boldsymbol{\pi}}(\boldsymbol{s}, \boldsymbol{\pi}_i(\boldsymbol{s}), \boldsymbol{\pi}_{-i}(\boldsymbol{s})) + \sum_{c \in [d]} \lambda_{ic}(\boldsymbol{s}) \partial_{\boldsymbol{a}_i} h_{ic}(\boldsymbol{s}, \boldsymbol{\pi}_i(\boldsymbol{s}), \boldsymbol{\pi}_{-i}(\boldsymbol{s})) \tag{59}$$

$$\forall c \in [d], \quad 0 = \lambda_{ic}(\boldsymbol{s}) h_{ic}(\boldsymbol{s}, \boldsymbol{\pi}_i(\boldsymbol{s}), \boldsymbol{\pi}_{-i}(\boldsymbol{s})) \tag{60}$$

$$\forall c \in [d], \quad 0 \leq h_{ic}(\boldsymbol{s}, \boldsymbol{\pi}_i(\boldsymbol{s}), \boldsymbol{\pi}_{-i}(\boldsymbol{s})) \tag{61}$$

*and for all $i \in [n]$, $\boldsymbol{s} \in \mathcal{S}$,*

$$v_i^{\boldsymbol{\pi}}(\boldsymbol{s}) = q_i^{\boldsymbol{\pi}}(\boldsymbol{s}, \boldsymbol{\pi}_i(\boldsymbol{s}), \boldsymbol{\pi}_{-i}(\boldsymbol{s})) \tag{62}$$

*Proof.* First, we know that a policy profile $\boldsymbol{\pi} \in \mathcal{F}^{\text{markov}}$ is a GMPE if and only if it satisfies the following generalized Bellman Optimality equations, i.e., for all $i \in [n]$, $\boldsymbol{s} \in \mathcal{S}$,

$$v_i^{\boldsymbol{\pi}}(\boldsymbol{s}) = \max_{\boldsymbol{a}_i \in \mathcal{X}_i(\boldsymbol{s}, \boldsymbol{\pi}_{-i}(\boldsymbol{s}))} r_i(\boldsymbol{s}, \boldsymbol{a}_i, \boldsymbol{\pi}_{-i}(\boldsymbol{s})) + \gamma \mathbb{E}_{\boldsymbol{s}' \sim p(\cdot|\boldsymbol{s}, \boldsymbol{a}_i, \boldsymbol{\pi}_{-i}(\boldsymbol{s}))}[v_i^{\boldsymbol{\pi}}(\boldsymbol{s}')] \tag{63}$$

$$= \max_{\boldsymbol{a}_i \in \mathcal{X}_i(\boldsymbol{s}, \boldsymbol{\pi}_{-i}(\boldsymbol{s}))} q_i^{\boldsymbol{\pi}}(\boldsymbol{s}, \boldsymbol{a}_i, \boldsymbol{\pi}_{-i}(\boldsymbol{s})) \tag{64}$$

Then since $\boldsymbol{a}_i \mapsto q_i^{\boldsymbol{\pi}}(\boldsymbol{s}, \boldsymbol{a}_i, \boldsymbol{\pi}_{-i}(\boldsymbol{s}))$ is concave over $\mathcal{X}_i(\boldsymbol{s}, \boldsymbol{\pi}_{-i}(\boldsymbol{s}))$ by Assumption 2, the KKT conditions provides sufficient and necessary optimality conditions for the constrained maximization problem

$$\max_{\boldsymbol{a}_i \in \mathcal{X}_i(\boldsymbol{s}, \boldsymbol{\pi}_{-i}(\boldsymbol{s}))} q_i^{\boldsymbol{\pi}}(\boldsymbol{s}, \boldsymbol{a}_i, \boldsymbol{\pi}_{-i}(\boldsymbol{s})) \tag{65}$$

That is, $\boldsymbol{a}_i^* \in \mathcal{X}_i(\boldsymbol{s}, \boldsymbol{\pi}_{-i}(\boldsymbol{s}))$ is a solution to eq. (65) if and only if there exists $\{\lambda_{ic}^* : \mathcal{S} \to \mathbb{R}_+\}_{c \in [d]}$ s.t.

$$0 \in \partial_{\boldsymbol{a}_i} q_i^{\boldsymbol{\pi}}(\boldsymbol{s}, \boldsymbol{a}_i^*, \boldsymbol{\pi}_{-i}(\boldsymbol{s})) + \sum_{c \in [d]} \lambda_{ic}^*(\boldsymbol{s}) \partial_{\boldsymbol{a}_i} h_{ic}(\boldsymbol{s}, \boldsymbol{a}_i^*, \boldsymbol{\pi}_{-i}(\boldsymbol{s})) \tag{66}$$

$$\forall c \in [d], \quad 0 = \lambda_{ic}^*(\boldsymbol{s}) h_{ic}(\boldsymbol{s}, \boldsymbol{a}_i^*, \boldsymbol{\pi}_{-i}(\boldsymbol{s})) \tag{67}$$

$$\forall c \in [d], \quad 0 \le h_{ic}(\boldsymbol{s}, \boldsymbol{a}_i^*, \boldsymbol{\pi}_{-i}(\boldsymbol{s})) \tag{68}$$

Therefore, we can conclude that $\boldsymbol{\pi} \in \mathcal{F}^{\text{markov}}$ is a GMPE if and only if there exists $\{\lambda_{ic} : \mathcal{S} \to \mathbb{R}_+\}_{i \in [n], c \in [d]}$ s.t. for all $i \in [n]$, $\boldsymbol{s} \in \mathcal{S}$,

$$0 \in \partial_{\boldsymbol{a}_i} q_i^{\boldsymbol{\pi}}(\boldsymbol{s}, \boldsymbol{\pi}_i(\boldsymbol{s}), \boldsymbol{\pi}_{-i}(\boldsymbol{s})) + \sum_{c \in [d]} \lambda_{i,c}(\boldsymbol{s}) \partial_{\boldsymbol{a}_i} h_{ic}(\boldsymbol{s}, \boldsymbol{\pi}_i(\boldsymbol{s}), \boldsymbol{\pi}_i(\boldsymbol{s})) \tag{69}$$

$$\forall c \in [d], \quad 0 = \lambda_{ic}(\boldsymbol{s}) h_{ic}(\boldsymbol{s}, \boldsymbol{\pi}_i(\boldsymbol{s}), \boldsymbol{\pi}_i(\boldsymbol{s})) \tag{70}$$

$$\forall c \in [d], \quad 0 \le h_{ic}(\boldsymbol{s}, \boldsymbol{\pi}_i(\boldsymbol{s}), \boldsymbol{\pi}_{-i}(\boldsymbol{s})) \tag{71}$$

and for all $i \in [n]$, $\boldsymbol{s} \in \mathcal{S}$,

$$v_i^{\boldsymbol{\pi}}(\boldsymbol{s}) = q_i^{\boldsymbol{\pi}}(\boldsymbol{s}, \boldsymbol{\pi}_i(\boldsymbol{s}), \boldsymbol{\pi}_{-i}(\boldsymbol{s})) \tag{72}$$

$\square$

Therefore, for a policy profile $\boldsymbol{\pi} \in \mathcal{F}^{\text{markov}}$ and a Lagrange multiplier policy $\boldsymbol{\lambda} : \mathcal{S} \to \mathbb{R}_+^{n \times d}$, consider the *total first-order violation*

$$\Xi_{\text{first-order}}(\boldsymbol{\pi}, \boldsymbol{\lambda}) = \sum_{i \in [n]} \left\| \int_{\boldsymbol{s} \in \mathcal{S}} \partial_{\boldsymbol{a}_i} q_i^{\boldsymbol{\pi}}(\boldsymbol{s}, \boldsymbol{\pi}_i(\boldsymbol{s}), \boldsymbol{\pi}_{-i}(\boldsymbol{s})) + \sum_{c \in [d]} \lambda_{i,c}(\boldsymbol{s}) \partial_{\boldsymbol{a}_i} h_{ic}(\boldsymbol{s}, \boldsymbol{\pi}_i(\boldsymbol{s}), \boldsymbol{\pi}_{-i}(\boldsymbol{s})) ds \right\|_2^2 \tag{73}$$

and the *average Bellman error*

$$\Xi_{\text{Bellman}}(\boldsymbol{\pi}, \boldsymbol{\lambda}) = \sum_{i \in [n]} \left\| \int_{\boldsymbol{s} \in \mathcal{S}} v_i^{\boldsymbol{\pi}}(\boldsymbol{s}) - q_i^{\boldsymbol{\pi}}(\boldsymbol{s}, \boldsymbol{\pi}_i(\boldsymbol{s}), \boldsymbol{\pi}_{-i}(\boldsymbol{s})) ds \right\|_2^2 . \tag{74}$$

We can directly approximate the GMPE through minimizing the sum of these two errors.

Typically, approximating the GMPE using the neural projection method requires optimizing both the policy profile and the Lagrange multiplier policy. However, in exchange economy Markov pseudo-games, we derive a closed-form solution for the optimal Lagrange multiplier, allowing us to focus solely on optimizing the policy profile.

### E.2 MORE RESULTS

### E.3 IMPLEMENTATION DETAILS

**Deterministic Case Training Details** For deterministic transition probability case, for each reward function class we randomly sampled one economy with 10 consumers, 10 commodities, 1 asset, and 5

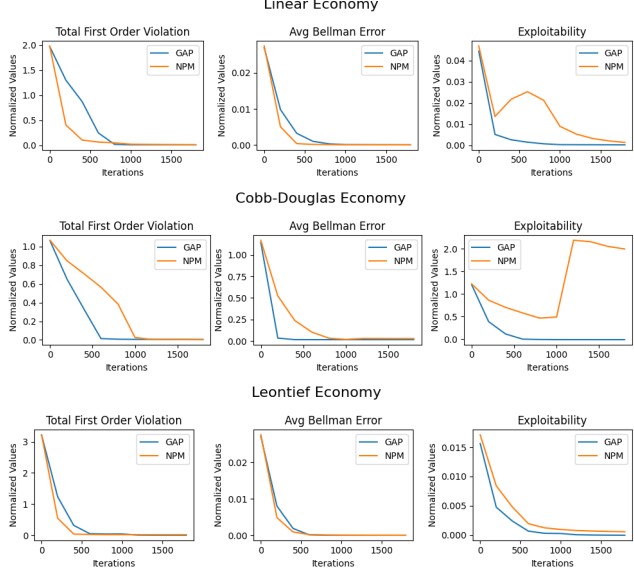

Figure 2: Normalized Metrics for Economies with Deterministic Transition Probability Function

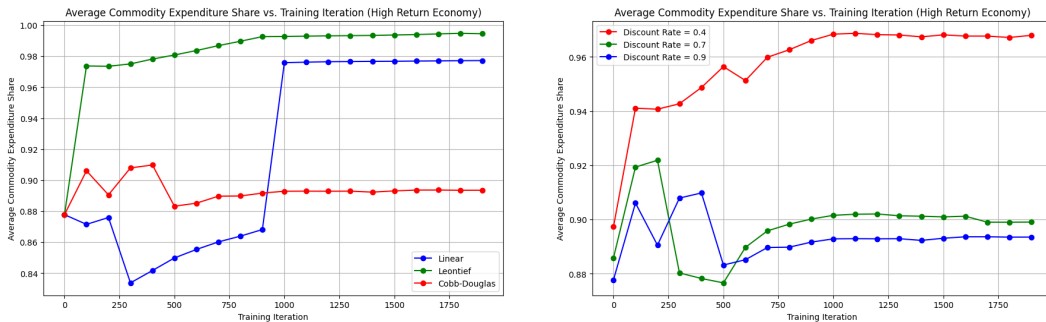

Figure 3: Average Commodity Expenditure Share for Different Utility Structures

Figure 4: Average Commodity Expenditure Share for Different Discount Rate

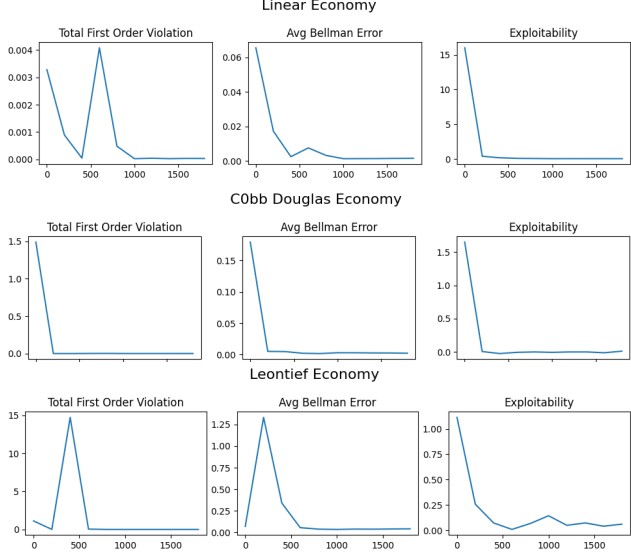

Figure 5: Larger Economies with Stochastic Transition Functions

world state. The asset return matrix $\boldsymbol{R}$ is sampled in a way such that $r_{okj} \sim U([0.5, 1.1])$ for all $o$, $k$, and $j$. Moreover, we set the length of the stochastic process to be 30. For the initial state, we sample each consumer's endowment $\boldsymbol{e}_i \sim U([0.01, 0.1])^m$ and normalized so that the total endowment of each commodity add up to 1. We also sample each consumer's type $\boldsymbol{\theta}_i \sim U([1.0, 5.0])^m$, and set the world state to be 0. The transition probability function $p$ is defined as $p(\boldsymbol{s}' \mid \boldsymbol{s}, \boldsymbol{Y}) = 1$ for all $\boldsymbol{s}(o, \boldsymbol{E}, \boldsymbol{\Theta})$ where $\boldsymbol{s}' = (o', \boldsymbol{E}', \boldsymbol{\Theta}')$ is defined as $o' = 0$, $\boldsymbol{E}' = 0.01 \cdot \boldsymbol{1}_{n \times m}$, and $\boldsymbol{\Theta}' = \boldsymbol{\Theta}$.

Then, for both GAPNets method and neural projection method, we run 1000 episodes for each learning rate candidate in a grid search manner and measure the performance in terms of minimizing total first-order violation and average Bellman error. Finally, we pick the best hyperparameter for the final experiments.

In the final experiments, we run GAPNets for 2000 episodes using learning rates $\boldsymbol{\eta}_{\boldsymbol{\omega}} = 1 \times 10^{-5}$, $\boldsymbol{\eta}_{\boldsymbol{\sigma}} = 1 \times 10^{-5}$ for the linear economy, $\boldsymbol{\eta}_{\boldsymbol{\omega}} = 1 \times 10^{-5}$, $\boldsymbol{\eta}_{\boldsymbol{\sigma}} = 1 \times 10^{-5}$ for the Cobb-Douglas economy, and $\boldsymbol{\eta}_{\boldsymbol{\omega}} = 1 \times 10^{-5}$, $\boldsymbol{\eta}_{\boldsymbol{\sigma}} = 1 \times 10^{-5}$ for the Leontief economy. Similarly, we ran neural projection method for 2000 episodes using learning rates $\boldsymbol{\eta}_{\boldsymbol{\omega}} = 1 \times 10^{-4}$ for the linear economy, $\boldsymbol{\eta}_{\boldsymbol{\omega}} = 2.5 \times 10^{-5}$ for the Cobb-Douglas economy, and $\boldsymbol{\eta}_{\boldsymbol{\omega}} = 1 \times 10^{-4}$ for the Leontief economy. In this process, we compute the exploitability of computed policy profile through gradient ascent of the adversarial network. In specific, we ran 1000 episodes of gradient ascent with learning rate $\boldsymbol{\eta}_{\boldsymbol{\sigma}} = 5 \times 10^{-5}$ for the linear economy, $\boldsymbol{\eta}_{\boldsymbol{\sigma}} = 1 \times 10^{-4}$ for the Cobb-Douglas economy, and $\boldsymbol{\eta}_{\boldsymbol{\sigma}} = 1 \times 10^{-4}$ for the Leontief economy.

Next, for each economy, we randomly sample 50 policy profiles and record their total first-order violations, average Bellman errors, and exploitabilities. Finally, we normalize the results by the average of the sampled values.

**Stochastic Case Training Details** For stochastic transition probability case, for each reward function class we randomly sampled one economy with 10 consumers, 10 commodities, 1 asset, and 5 world state. The asset return matrix $\boldsymbol{R}$ is sampled in a way such that $r_{okj} \sim U([0.5, 1.1])$ for all $o$, $k$, and $j$. Moreover, we set the length of the stochastic process to be 30. For the initial state, we sample each consumer's endowment $\boldsymbol{e}_i \sim U([0.01, 0.1])^m$ and normalized so that the total endowment of each commodity add up to 1. We also sample each consumer's type $\boldsymbol{\theta}_i \sim U([1.0, 5.0])^m$, and set the world state to be 0. The transition probability function will stochastically transition from state $\boldsymbol{s}(o, \boldsymbol{E}, \boldsymbol{\Theta})$ to state $\boldsymbol{s}' = (o', \boldsymbol{E}', \boldsymbol{\Theta}')$ where $o' \sim U(\{0, 1, 2, 3, 4\})$, $\boldsymbol{E}' \sim 0.002 + U([0.01, 0.1])^{n \times m}$, and $\boldsymbol{\Theta}' = \boldsymbol{\Theta}$.

Then, for both GAPNets method and neural projection method, we run 1000 episodes for each learning rate candidate in a grid search manner and measure the performance in terms of minimizing total first-order violation and average Bellman error. Finally, we pick the best hyperparameter for the final experiments.

In the final experiments, we run GAPNets for 2000 episodes using learning rates $\boldsymbol{\eta}_{\boldsymbol{\omega}} = 1 \times 10^{-5}$, $\boldsymbol{\eta}_{\boldsymbol{\sigma}} = 1 \times 10^{-5}$ for the linear economy, $\boldsymbol{\eta}_{\boldsymbol{\omega}} = 2.5 \times 10^{-5}$, $\boldsymbol{\eta}_{\boldsymbol{\sigma}} = 2.5 \times 10^{-5}$ for the Cobb-Douglas economy, and $\boldsymbol{\eta}_{\boldsymbol{\omega}} = 5 \times 10^{-5}$, $\boldsymbol{\eta}_{\boldsymbol{\sigma}} = 5 \times 10^{-5}$ for the Leontief economy. Similarly, we ran neural projection method for 2000 episodes using learning rates $\boldsymbol{\eta}_{\boldsymbol{\omega}} = 5 \times 10^{-5}$ for the linear economy, $\boldsymbol{\eta}_{\boldsymbol{\omega}} = 2.5 \times 10^{-5}$ for the Cobb-Douglas economy, and $\boldsymbol{\eta}_{\boldsymbol{\omega}} = 5 \times 10^{-4}$ for the Leontief economy. In this process, we compute the exploitability of computed policy profile through gradient ascent of the adversarial network. In specific, we ran 1000 episodes of gradient ascent with learning rate $\boldsymbol{\eta}_{\boldsymbol{\sigma}} = 7.5 \times 10^{-4}$ for the linear economy, $\boldsymbol{\eta}_{\boldsymbol{\sigma}} = 1 \times 10^{-4}$ for the Cobb-Douglas economy, and $\boldsymbol{\eta}_{\boldsymbol{\sigma}} = 1 \times 10^{-4}$ for the Leontief economy. When estimating the neural loss function—cumulative regret for the GAPNets method and total first-order violations and average Bellman error for the neural projection method—we use 100 samples for GAPNets and 10 samples for the neural projection method. The primary reason for this difference is the high memory consumption of the neural projection method, which makes larger sample sizes infeasible.

Next, for each economy, we randomly sample 50 policy profiles and record their total first-order violations, average Bellman errors, and exploitabilities. Finally, we normalize the results by the average of the sampled values.

### E.4 OTHER DETAILS

**Programming Languages, Packages, and Licensing**   We ran our experiments in Python 3.7 Van Rossum & Drake Jr (1995), using NumPy Harris et al. (2020), , CVXPY Diamond & Boyd (2016), Jax Bradbury et al. (2018), OPTAX Bradbury et al. (2018), Haiku Hennigan et al. (2020), and JaxOPT Blondel et al. (2021). All figures were graphed using Matplotlib Hunter (2007).

Python software and documentation are licensed under the PSF License Agreement. Numpy is distributed under a liberal BSD license. Pandas is distributed under a new BSD license. Matplotlib only uses BSD compatible code, and its license is based on the PSF license. CVXPY is licensed under an APACHE license.

**Computational Resources**   The experiments were conducted using Google Colab, which provides cloud-based computational resources. Specifically, we utilized an NVIDIA T4 GPU with the following specifications: GPU: NVIDIA T4 (16GB GDDR6), CPU: Intel Xeon (2 vCPUs), RAM: 12GB, Storage: Colab-provided ephemeral storage.

**Code Repository**   the full details of our experiments, including hyperparameter search, final experiment configurations, and visualization code, can be found in our code repository (https://github.com/Sadie-Zhao/Infinite-Horizon-Markov-Economies-ICLR-2026).