# OpenReview forum: "Infinite Horizon Markov Economies"
_ICLR.cc/2026/Conference — ICLR 2026 Poster_

### Official Review · Reviewer_hHrX · 2025-10-31

**Soundness:** 3
**Presentation:** 3
**Contribution:** 4
**Rating:** 6
**Confidence:** 2

**Summary:**

This paper investigates the Markov pseudo game and develops a first-order solution method with polynomial-time guarantees for the existence of equilibria. One key contribution is that the authors demonstrate recursive Radner equilibria in infinite-horizon Markov exchange economies can be reformulated as concave Markov pseudo-games (MPG). Once formulated as an MPG, new solution concepts can be applied to solve for equilibrium. Empirical results are provided via a GAN policy network representing players in an exchange economy, adapting the learning parameters via computation of recursive Radner equilibrium (RRE) via Alg. 1 (TTSSGDA).

**Strengths:**

This paper proves the existence of the generalized markov perfect equilibria(GMPE) for concave Markov pseudo game, which extending Arrow-Debreu's existence theorem.

Based on a two-time scale stochastic gradient descent ascent algorithm, they provided a algorithm for finding the approximate GMPEs. They further provided the convergence rate under this algorithm.

The authors demonstrate recursive Radner equilibria in infinite-horizon Markov exchange economies can be reformulated as MPG and in consequence, proved the existence of recursive Radner equilibria

I also really like the scope of experiments and find them highly relevant - and I appreciate the authors motivations to address non-incremental problems.

**Weaknesses:**

Some of the critical assumptions are relegated to the appendix (e.g. line 79).

Despite being somewhat knowlegeable in Markov game theory - this paper is highly complex, and also subject to a lot of notational overload - for example in line 88 the definition for M, and line 310 the notation for I. I would really suggest being more concise with the notation and even some notational abuse where necessary to reduce the overload and convey meaning faster. I could only understand the scope of the contributions and what the ramifications of these contributions are, but I failed to get an intuitive grasp of the solution (proof) method overall nor been successful to follow through on the proofs entirely (although the sections I did review seem correct).

---
Therefore the result of my assessment hinges on hoping that in the rebuttal the authors could provide a simpler explanations to how they arrived at their conclusions.

**Questions:**

Does the statement in 164-166 regarding $\epsilon$-stationarity imply that no FPTAS solution exists for $\epsilon$-stationarity point in the MPG? Is this tight, or just a claim?

The fact that the reduction to RRE exhibits polynomial-time convergence guarantees are compelling - nevertheless could the authors be transparent about the limitations specifically w.r.t. this reduction?

---

> ### Author Response · Authors · 2025-11-26
>
> We thank the reviewer for their thoughtful comments and the thorough evaluation of our work. We begin with clarification regarding the described weaknesses (W) and then answer the questions (Q).
>
> > **(W1.)** Some of the critical assumptions are relegated to the appendix (e.g. line 79).
>
> **(R1.)** We appreciate the reviewer’s concern. The "technical assumptions" referenced on line 79 are standard terminologies (e.g., Lipschitz continuity, Lipschitz smoothness, stochastic convexity) in the optimization literature and formally stated in Appendix A.1. These assumptions are not unique to our setting and do not restrict economic generality, but we agree that mentioning them earlier would improve readability.
>
> In the revised version, we will move key assumptions and brief intuitive explanations to Section 2, while keeping full technical definitions in the appendix.
>
> > **(W2)**. Despite being somewhat knowlegeable in Markov game theory - this paper is highly complex, and also subject to a lot of notational overload - for example in line 88 the definition for M, and line 310 the notation for I. I would really suggest being more concise with the notation and even some notational abuse where necessary to reduce the overload and convey meaning faster.  I could only understand the scope of the contributions and what the ramifications of these contributions are, but I failed to get an intuitive grasp of the solution (proof) method overall nor been successful to follow through on the proofs entirely (although the sections I did review seem correct). Therefore the result of my assessment hinges on hoping that in the rebuttal the authors could provide a simpler explanations to how they arrived at their conclusions.
>
> **(R2.)** We thank the reviewer for this helpful suggestion. We agree that the notation is dense due to the generality of Markov pseudo-games and infinite-horizon economies, but we will make it more concise where possible (e.g., reducing redundant symbols, combining equivalently defined sets, and adding a notation table).
>
> To provide an intuitive grasp of the proof structure, we now include a high-level summary (without heavy notation) that explains how the theoretical results are constructed:
>
> - **Markov Pseudo-Games**:
>     - **Existence Results (Theorem 2.1.)**: In the proof, we first establish that the *multi-player best-response correspondence* $\Phi: \Pi\to \Pi$ is upper hemicontinuous, non-empty, compact, and convex-valued, and by Fan’s fixed-point theorem, it admits a fixed point. We then argue that any fixed point of $\Phi$ is a GMPE using a recursive argument.
>     - **Computation Results**:
>         - **(Lemma 1.)** We first establish that “computing GNE = minimizing exploitability” and “computing GMPE = minimizing state exploitability for all states”.
>         - **(Observation 1.)** We then formulate the problem of minimizing exploitability and minimizing state exploitability as coupled min-max optimization problems.
>         - **(Lemma 2.)** We introduce dependent policies and proper parameterization schemes so the problem of minimizing exploitability becomes an unconstrained min-max optimization problem!
>         - **(Lemma 3. & Lemma 4.)** We showed that for a large class of Markov pseudo-games, “Minimizing exploitability ≈ Minimizing state exploitability at every state”. In specific, Lemma 3  states that a stationary point of exploitability is almost surely also a stationary point of state exploitability at all states. Lemma 4 upper bounds gradient of state exploitability in terms of gradient of exploitability, when the best-response mismatch coefficient is bounded.
>           Remark: Our best-response mismatch coefficient results (Lemma 4) generalizes the minimax mismatch coefficient in two-player settings [1] and the distribution mismatch coefficient in single-agent settings [2].
>         - **Final convergence results (Theorem 2.2.)** The proof mainly relies on the fact that under mild assumption, the state-exploitability is non-convex in the minimizer’s policy profile but gradient-dominated in the maximizer’s dependent policy profile.
> - **Infinite Horizon Markov Exchange Economies**
>     - **Main Results (Theorem 3.1.)** For any infinite horizon Markov exchange economy (MEE), we can construct a corresponding Markov pseudo-game (Definition 2) such that the equilibria of MEE corresponds to the equilibria of the Markov pseudo-game.
>         - **(Existence)** The existence of GMPE in Markov pseudo-games (Theorem 2.1.) therefore implies the existence of REE in infinite horizon MEEs (Theorem 3.1.)
>         - **(Computation)** The polynomial-time approximation of GMPE in Markov pseudo-game (Theorem 2.2.) implies the polynomial-time approximation of REE in infinite horizon MEEs (Theorem 3.3.)

---

> ### Author Response · Authors · 2025-11-26
>
> ## Questions:
> > **(Q1.)** Does the statement in 164-166 regarding $\epsilon$-stationarity imply that no FPTAS solution exists for $\epsilon$-stationarity point in the MPG? Is this tight, or just a claim?
>
> **(A1.)** In our setting, we cannot assume any strong structural properties such as convexity or strong concavity of cumulative regret $\psi$. Without such assumptions, the objective $\phi(\pi) = \max_{\pi’\in \mathcal{F}(\pi)} \psi(\pi, \pi’)$ belongs to the general class of nonsmooth, nonconvex, Lipschitz functions. This class is expressive enough to encode NP-hard problems [3]. Consequently, computing even an $\varepsilon$-stationary point is NP-hard in the worst case, and no FPTAS exists for the entire class unless P=NP.
>
>
> > **(Q2.)** The fact that the reduction to RRE exhibits polynomial-time convergence guarantees are compelling - nevertheless could the authors be transparent about the limitations specifically w.r.t. this reduction?
>
> **(A2.)** The reduction from an infinite-horizon Markov exchange economy (MEE) to its equivalent Markov pseudo-game (MPG) is exact and relies only on standard economic assumptions (Assumption 6). The main limitation does not lie in this reduction, but in the **solution concept that is computationally tractable**.
>
> Computing an exact Radner equilibrium would require solving a fully nonconvex, nonsmooth, and constrained optimization problem; it is NP-hard to find even an $\varepsilon$-stationary point in such a problem. We thus aim to compute an $(\varepsilon, \delta)$-stationary point of exploitability, which is the strongest polynomial-time guarantee achievable for the general class of infinite-horizon Markov exchange economies without imposing additional extreme assumptions.
>
> Economically, an $(\varepsilon,\delta)$-stationary point of exploitability corresponds to an outcome in the neighborhood of another outcome that approximately satisfies the necessary conditions of an RRE; moreover, in the limit, the necessary conditions are satisfied exactly.
>
> Importantly, while our theory guarantees convergence only to approximate stationary solutions, **in our experiments, the behavior of the learned policies closely resembles that of recursive Radner equilibria**, providing meaningful consumption paths, price dynamics, and portfolio decisions. Thus, the computational method not only matches the theoretical guarantee but also empirically achieves near-equilibrium behavior that goes beyond worst-case complexity limits.
>
>
>
> [1]Constantinos Daskalakis, Dylan J Foster, and Noah Golowich. 2020. Independent policy gradient methods for competitive reinforcement learning. Advances in neural information processing systems 33 (2020), 5527–5540.
> [2]Alekh Agarwal, Sham M Kakade, Jason D Lee, and Gaurav Mahajan. 2020. Optimality and approximation with policy gradient methods in markov decision processes. In Conference on Learning Theory. PMLR, 64–66.
> [3] Zhang, J., Lin, H., Jegelka, S., Sra, S. &amp; Jadbabaie, A.. (2020). Complexity of Finding Stationary Points of Nonconvex Nonsmooth Functions. <i>Proceedings of the 37th International Conference on Machine Learning</i>, in <i>Proceedings of Machine Learning Research</i> 119:11173-11182
>
>
>
> ---------
> We hope our clarification help address your concerns. We would be happy to clarify any remaining points further during the author-reviewer discussion period.

---

### Official Review · Reviewer_giHS · 2025-11-01

**Soundness:** 3
**Presentation:** 2
**Contribution:** 2
**Rating:** 4
**Confidence:** 3

**Summary:**

This work introduces the notion of *Markov pseudo-games*, an extension of Markov games where each agent's action space depends on the actions of other agents. The authors then introduce two solution concepts: GMPE and GNE. Under Assumption 3, the authors reformulate this problem as a min-max optimization problem. By further assuming the concavity with respect to the max-player, the paper proposes a two-timescale GDA algorithm that provably converge to an approximate stationary point. In Section 3, the authors show that infinite horizon Markov economy can be modeled as a Markov pseudo game and provide an existential proof of RRE, along with an algorithm that efficiently computes it.

**Strengths:**

- The framework the authors propose is novel. It is nice to see that the authors model a real-world economic scenario using their proposed framework.

- The paper extends the existential result of RRE in infinite horizon Markov exchange economies.

- The paper provides a provable efficient algorithm that converge to a GMPE in Markov pseudo-games and RRE in infinite horizon Markov exchange economies.

- Numerical results are provided where the authors model agents as deep learning neural networks, validating the theorems.

**Weaknesses:**

- The notations of this paper is quite dense and difficult to follow. Some notations are used before being properly defined (i.e. $D_{\varphi}(\pi)$ in line 158).

- One of the interesting feature of Markov pseudo-games is that each agent's action space depends on actions of other agents. However, Assumption 3 largely simplifies that the problem into a standard min-max optimization problem without any coupling dynamics. This assumption appears overly strong and undermines the main motivation of introducing Markov pseudo-games.

- From an optimization perspective, the problem reduces to a min-max optimization problem with hidden structures, which has been extensively studied in the optimization and game theory literature [1].

- Theorem 2.2 does not provide an explicit bound on the convergence time, making it difficult to compare the algorithm’s efficiency with other min-max optimization methods.

- Section 3 is heavily focused on economics, which may limit its appeal to the broader ICLR audience. I also suggest that the authors include additional background on Markov exchange markets in Section 3 to make the section more accessible to readers without an economics background.

-----

[1] Flokas, L., Vlatakis-Gkaragkounis, E.-V., and Piliouras, G. Solving min-max optimization with hidden structure via gradient descent ascent. arXiv preprint arXiv:2101.05248, 2021.

**Questions:**

See Weaknesses.

---

> ### Author Response · Authors · 2025-11-26
>
> We thank the reviewer for their comments and questions. Below, we address the identified weaknesses (W) and hope that our clarifications will contribute to a more positive evaluation of our paper.
>
> > **(W1.)**  The notations of this paper is quite dense and difficult to follow. Some notations are used before being properly defined (i.e. $D\_\phi(\pi)$  in line 158).
>
> **(R1.)** We appreciate the reviewer’s observation. Due to the generality of infinite-horizon Markov economies and Markov pseudo-games, our notation is necessarily rich. That said, we agree that clarity can be improved. In the revised version, we will: (i) introduce key notation earlier in the main text, (ii) add a notation summary table, and (iii) reference where symbols (e.g., $D\_\phi(\pi)$) are first defined.
>
> Specifically, $D\_\phi(\pi)$ denotes the subdifferential of the exploitability function and is formally defined in Appendix A.1. We will ensure it is clearly referenced at its first appearance in Section 2. These changes should improve accessibility without compromising rigor.
>
> > **(W2.)** One of the interesting feature of Markov pseudo-games is that each agent's action space depends on actions of other agents. However, Assumption 3 largely simplifies that the problem into a standard min-max optimization problem without any coupling dynamics. This assumption appears overly strong and undermines the main motivation of introducing Markov pseudo-games.
>
> **(R2.)** We thank the reviewer for highlighting this important point. We clarify that Assumption 3 does **no** simplify away the pseudo-game dependence -- it merely asserts that the policy representation must be rich enough to encode feasibility constraints with respect to other agents’ actions. We disagree that this condition is too strong; in fact, as we show in our experiments, **it is achievable** by construction for most Markov pseudo-games. Instead, this assumption allows us to clearly link practice and theory, allowing us to provide a mathematical characterization of RRE as well as a practical method for RRE computation.
>
> In a pseudo-game, feasible actions satisfy $a_i \in \mathcal{X}\_i(s, a_{-i})$, so any feasible policy of agent $i$ must depend on other agents’ policies. Rather than assuming independence, we explicitly enforce this dependence by introducing a *dependent policy*  $\rho: \mathcal{S} \times \mathcal{A} \rightarrow \mathcal{A}$ that *takes another policy profile $\pi$ as an input*, where $\rho\_i(s, \pi(s)) \in \mathcal{X}\_i(s, \pi\_{-i}(s))$. This representation preserves the coupling while reformulating the problem into an unconstrained differentiable min--max optimization.
>
> Importantly, Assumption 3 is not *restrictive*: for any Markov pseudo-game, one can always define a policy parameterization that enforces feasibility through **constraint layers, projection operators, or masked action heads**. In other words, feasibility is **handled by policy design**, **not by assuming away action dependence**.
>
> Therefore, Assumption 3 is not a simplification, but a technique that preserves pseudo-game structure within a differentiable policy class, making computation feasible while retaining action-dependence.
>
> > **(W3.)** From an optimization perspective, the problem reduces to a min-max optimization problem with hidden structures, which has been extensively studied in the optimization and game theory literature [1].
>
> **(R3.)** We appreciate the reviewer highlighting this connection. We will cite Flokas et al. (2021) and clarify the differences.
> While both formulations involve mi-max optimization, our problem does not fall under the scope of Flokas et al. (2021) as it is fundamentally different in two ways:
> 1) **Objective structure**: The exploitability and state exploitability functions used in our formulation are neither convex nor concave, and only satisfy gradient-dominance under additional assumptions, whereas Flokas et al. (2021) require hidden convex--concave structure.
>
> 2) **Policy dependence**: In Flokas et al., “each agent selects its own vectors of parameters which are then processed separately by smooth functions”. In contrast, our maximizer’s policy $\rho(s, \pi(s))$ **explicitly depends on** the minimizer’s policy, enforcing pseudo-game feasibility and violating the  separate parameterization assumption of Flokas et al. (2021).
>
> We will revise the Related Work section to clearly articulate these differences.

---

> ### Author Response · Authors · 2025-11-26
>
> >**(W4.)** Theorem 2.2 does not provide an explicit bound on the convergence time, making it difficult to compare the algorithm’s efficiency with other min-max optimization methods.
>
> **(R4)** We appreciate the reviewer’s remark. Theorem 2.2 establishes polynomial-time convergence, but for readability, we presented only the simplified asymptotic form in the main paper, which outlines polynomial-time dependency on the relevant parameters. We will revise Theorem 2.2 in the main text to include the exact condition, which is
> > If Algorithm 1 is run with inputs that satisfy for all $t \in [T]$,
> $\eta\_\theta \asymp
> \frac{\epsilon^4 \left( \left\|  \partial\ \delta_{\mu}^{\pi*}   / \partial \mu \right\|\_\infty / |1-\gamma| \right)^2}
>       {\ell\_{\nabla\Psi}^3 \sqrt{\left(\ell\_\Psi^2+\zeta\_\sigma^2\right)} \left(\ell\_\Psi/\ell\_{\nabla\Psi}+1\right)^2}$,
> $\eta_\sigma \asymp
> \frac{\epsilon^8 \left( \left\|  \partial\ \delta_{\mu}^{\pi*} / \partial \mu \right\|\_\infty / |1-\gamma| \right)^4}{\ell\_{\nabla\Psi}^5 \ell\_\Psi \left(\ell\_\Psi/\ell\_{\nabla\Psi}+1\right)^4 \left(\ell\_\Psi^2+\zeta\_\sigma^2\right)^{3/2}}$,
> and
> $T \ge \left( \frac{(1-\gamma)}{\left\| \partial \delta^{\pi^*}\_\mu / \partial \mu \right\|\_\infty}
>       + \frac{\ell_\Psi}{2\ell\_{\nabla\Psi}} \right)^{-1}
>       \frac{\ell_\Psi\,\mathrm{diam}(\Theta \times \Sigma)}{\epsilon^2 \eta\_\theta}$ ,
> then there exists $k\leq T$ s.t. $\omega^{(T)}\_{\text{best}}=\omega^{(k)}$ is an $(\epsilon, \frac{\epsilon}{2 \ell\_{\nabla\Psi}})$-stationary point of exploitability.
>
> This makes the polynomial dependence explicit and enables comparison with other min–max methods.
>
>
> > **(W5.)** Section 3 is heavily focused on economics, which may limit its appeal to the broader ICLR audience. I also suggest that the authors include additional background on Markov exchange markets in Section 3 to make the section more accessible to readers without an economics background.
>
> **(R5.)** We thank the reviewer for this helpful suggestion. While Section 3 introduces the economic motivation for the pseudo-game formulation and the connection to Radner equilibrium, we agree that additional intuition would make it more accessible to non-economics readers.
>
> We note that Appendix C already provides supporting background:
>  - Appendix C.1 gives a historical overview of the development of general equilibrium models to stochastic economies;
>  -  Appendix C.2 provides a formal definition of a static exchange economy.
>
> In the revised version, we will add a brief, non-technical overview at the beginning of Section 3 to summarize the key concepts (e.g., incomplete markets, Arrow securities, Radner equilibrium) at an intuitive level.

---

> ### Comment · Reviewer_giHS · 2025-11-28
>
> I thank the authors for their thorough rebuttals. However, I still have some additional questions.
>
> - I remain skeptical about the generality of Assumption 3. It is well known that coupled constraints are computationally hard to solve (even when the objective is convex–concave)[1][2]. However, Assumption 3 (together with the other assumptions) appears to imply the existence of efficient algorithms for problems that could potentially be PPAD-hard. Is it possible for the authors to establish explicit sufficient conditions under which this assumption holds (e.g., relating the size of the parameter space to the size of state–action space)?
>
> - Regarding the differences from Flokas et al., subsequent works have extended their results to a variety of settings, including those in which the objective satisfies gradient-domination properties (see [3] and references therein). It still seems the novelty (in terms of min-max optimization) of this paper is limited.
>
> ------
> [1] Daskalakis, C., Skoulakis, S., and Zampetakis, M.. The complexity of constrained min-max optimization. Symposium on Theory of Computing (STOC), 2021.
>
> [2] Bernasconim, M., Castiglioni, M., Celli, A., and Farina, G.. On the role of constraints in the complexity of min-max optimization. arXiv preprint arXiv:2411.03248, 2024.
>
> [3] Kalogiannis, F., Vlatakis-Gkaragkounis E.-V., Gemp I., and Piliouras, G.. Solving zero-sum convex markov games. ICML, 2025.

---

> > ### Author Response · Authors · 2025-11-30
> >
> > We thank the reviewer for the thoughtful follow-up questions.
> >
> > > **(FQ1.)** I remain skeptical about the generality of Assumption 3. It is well known that coupled constraints are computationally hard to solve (even when the objective is convex–concave)[1][2]. However, Assumption 3 (together with the other assumptions) appears to imply the existence of efficient algorithms for problems that could potentially be PPAD-hard. Is it possible for the authors to establish explicit sufficient conditions under which this assumption holds (e.g., relating the size of the parameter space to the size of state–action space)?
> >
> > **(FA1.)** We totally agree that a better clarification on Assumption 3 would be helpful, so in the revised version, we have added **further discussion on intuition of Assumption 3** and **an additional Lemma that explicitly present sufficient conditions on MPGs under which Assumption 3 holds**. In this response, we first clarify several conceptual points and then present our lemma.
> >
> > 1. **Clarifying a misunderstanding about computational hardness.** The reviewer states that coupled constraints are computationally hard even in convex–concave problems. To avoid confusion: **convex–concave min-max optimization problems with constraint are not PPAD-hard**. As the reviewer’s own references [1] note, “when f(x, y) is a convex-concave function, Problem (1.1) is guaranteed to have a solution, under compactness of the constraint set [vN28, Ros65], while computing a solution is amenable to convex programming. In fact, if f is L-smooth, the problem can be solved via first-order methods, which are iterative, only access f through its gradient,1 and achieve an approximation error of poly(L, 1/T) in T iterations.” Thus, our use of coupled constraints does not implicitly promise tractability for problems known to be PPAD-hard.
> >
> > 2. **Why our results do not contradict PPAD hardness.** Our goal is not to compute an exact equilibrium of a general multi-player Markov game---a task that is indeed PPAD-hard in general. Instead, our algorithm aims to compute an $(\varepsilon,\delta)$-stationary point of the exploitability objective,
> > $\min_{\omega}\max_{\sigma}\; \Psi\big(\pi(\cdot;\omega),\, \rho(\cdot,\pi(\cdot;\omega);\sigma)\big)$,
> > which is a local optimality notion, so our results do not contradict the known hardness of computing exact equilibria.
> > Importantly, the dependent-policy shifts the difficulty of the action-dependent feasibility constraints into the structure of $\rho$, which depends on $\pi$. After parametrization, $\omega$ appears in both arguments of $\Psi$; this additional complexity absorbs the structural coupling and enables us (together with the parameterization scheme introduced below) reformulating the problem into an unconstrained min–max optimization problem.
> >
> > 3. **What does Assumption 3 really do.** Assumption 3 specifies the two structural properties for the ideal parameterization scheme: (1) it enforces the **fixed-point feasibility condition** for the minimizer policy, and (2) it explicitly encodes the **dependence** of the maximizer policy on the minimizer through the dependent-policy architecture (i.e., $\rho$ takes $\pi(s)$ as input). In fact, constructing dependent policy that accomplishes (2) is very trivial: since we just want to make sure every player's action satisfies $\rho\_i(s, \pi(s))\in \mathcal{X}\_i(s, \pi\_{-i}(s))$, we just produce some $a_i$ and project it into $\mathcal{X}\_i(s, \pi\_{-i}(s))$ as it's a fixed, convex set. What really hard is constructing self-consistent minimizer policy that satisfies the fixed-point condition. However, we can still construct policy for a large class of MPGs:
> >
> > Given a MPG $\mathcal{M}$, if
> > 1. For any player $i \in [n]$, $a\_{-i} \in A\_{-i}$, $a_i \mapsto \mathcal{X}\_i(s,a_{-i})$ is non-empty, compact, and convex for all $s \in \mathcal{S}$.
> >  2.  There exists a partition $\mathcal{P} = \\{P\_1,\cdots,P\_K\\}$ of $[n]$ and a directed acyclic graph (DAG) $G = (\mathcal{P},E)$ that specifies the precedence relations. For any $P_k \in \mathcal{P}$, let
> >     $\text{anc}(P\_k) = \\{P\_k' \in \mathcal{P} \mid \text{there is a direct path } P\_k' \to \cdots \to P\_k\\}$
> >     be the set of ancestors of $P\_k$. For any $P\_k \in \mathcal{P}$ and any player $i \in P\_k$,
> >     $\mathcal{X}\_i(s,a\_i,a\_{-i})
> >         = \\{a\_i \in A\_i \mid g\_{kl}(s,a\_{P\_k},a_{\text{anc}(P\_k)}) \le 0 \text{ for all } l\\}$
> >     for some $g\_{kl}$ that is concave in $a\_i$. Note that $a\_{P\_k}$ and $a\_{\text{anc}(P\_k)}$ denote the action profile for all the players in group $P\_k$ and the action profile for all the players belonging to any ancestor groups of $P\_k$.
> >
> > Then, there exists a parameterization scheme that satisfies Assumption 3.
> >
> > The complete proof is presented in the revised version, but the high-level idea is that we can construct feasible policy by projecting raw NN outputs onto the feasible sets in a topological order over $G$.

---

> > ### Author Response · Authors · 2025-11-30
> >
> > >**(FQ2.)** Regarding the differences from Flokas et al., subsequent works have extended their results to a variety of settings, including those in which the objective satisfies gradient-domination properties (see [3] and references therein). It still seems the novelty (in terms of min-max optimization) of this paper is limited.
> >
> > We thank the reviewer for the follow-up question and for pointing us to the related line of work. We want to clarify that the focus of our paper is not on proposing a new min–max optimization method. Rather, our objective is to solve a **multi-player general-sum Markov pseudo-game**, which is a strict generalization of the **two-player zero-sum concave Markov games** studied in [3].
> >
> > While our algorithm has a min–max form, the optimization problem itself is fundamentally different from the one considered in Flokas et al. and subsequent works. In particular:
> > 1. The setting in [3] is a *two-player zero-sum concave Markov game*. In contrast, we study a *multi-player general-sum Markov pseudo-game where each agent’s feasible action set depends on the other agents’ actions*. This action-dependent feasibility constraint has no analogue in the min–max structure of [3].
> > 2. Our algorithm minimizes exploitability of the *entire policy profile* against best-responding deviating policy profiles, while [3] optimizes *individual policies*. These two optimization formulations are fundamentally different.
> > 3. In [3], the saddle point arises from the concave-convex structure of the *two-player objective*. In our formulation, the saddle point arises from the definition of *exploitability* in multi-player Markov pseudo-games—not from the original payoff function—so the optimization landscape and theoretical requirements differ substantially.
> >
> > For these reasons, even though both problems can be written as min–max objectives, the structures, goals, and techniques (in terms of policy parameterization) are not comparable.
> >
> > Moreover, the main contribution of our paper is **not a new min–max optimization method**, but a **general theoretical and computational framework for infinite-horizon Markov economies**, including existence of generalized Markov perfect equilibria (Theorem 2.1), existence of recursive Radner equilibria via reduction to pseudo-games (Theorem 3.1), and a polynomial-time convergence guarantee for computing approximate equilibria (Theorem 2.2). These results are, to the best of our knowledge, **the first to unify infinite-horizon stochastic exchange economies with a tractable computational framework and convergence guarantees**. While min–max optimization appears as a *tool* within our method, it is not the main novelty of the work. The main novelty lies in formulating and solving equilibria in a previously unaddressed class of dynamic economic models.
> >
> > We appreciate the reviewer’s suggestion and will ensure the revised version clearly highlights these conceptual differences.

---

### Official Review · Reviewer_9YuJ · 2025-11-02

**Soundness:** 3
**Presentation:** 3
**Contribution:** 3
**Rating:** 8
**Confidence:** 3

**Summary:**

The paper introduces Markov Pseudo Games, a class of stochastic games that combine the standard Markov games and pseudo-games with the flexibility to capture dynamic uncertainty and action-dependent feasibility constraints. This class of games is particularly suitable for modeling infinite-horizon stochastic economies with incomplete markets, where agents face evolving uncertainty and limited strategic feasibility. The main contribution of the work is the proof of existence of a recursive Radner equilibrium within this generalized setting. Authors achieve this by reformulating the problem as a Generalized Markov Perfect Equilibrium (GMPE) of the corresponding Markov Pseudo Game, using a suitable exploitability function. Furthermore, a polynomial-time first-order solution approach is proposed that uses deep neural networks and minimax optimization; this enables an efficient computation of equilibria in high-dimensional, dynamic environments. Some computations are illustrated.

**Strengths:**

The paper considers a novel and challenging problem. Also, it has a good motivation. The proposed solution approach is also novel. The proposed formulation is particularly interesting, as it unifies infinite-horizon stochastic economies (relevant for macroeconomics area) with incomplete markets using a common theoretical model. The notion of recursive Radner equilibria in this framework is good; it offers an  understanding of the market dynamics. It seems that Roy Radner was not awarded a Noble Prize; while the broad area of general equilibrium seems important enough in econ literature that this sub-area had at least a couple of those awards.

The work has a good theoretical rigor in my view; the theorems and proofs are carefully done. This is a good contribution to macroeconomic modelling. The reformulation of the problem as a Generalized Markov Perfect Equilibrium (GMPE) is a key contribution that allows the authors to effectively address challenges in prior literature. Furthermore, the paper opens promising directions for future research in economic theory and computational economics; this is possibly via deep reinforcement learning and economic equilibrium analysis.

**Weaknesses:**

The paper is primarily theoretical, with limited experimental validation. Thus, we have very limited view of the performance of the proposed approach across networks of varying sizes both in terms of the number of agents and the size of the state space. Hence, the important aspect of scalability and robustness remains underexplored.


There are many assumptions that are part of the paper’s analysis (to be expected in a theoretically oriented paper). For example, Assumption 6. But, it is deferred to the appendix. It is better to make the main paper self-contained by summarizing or at least highlighting such essential assumptions within the main text. Similarly, many theorems and results are presented with minimal discussion of their intuition or motivation. Including brief insights/intuition into the meaning and implications of these results would significantly improve the paper’s readability. In general, some discussion on the need or motivation for assumptions is needed.

Similarly, some of the key definitions such as v^\pi  and u^\pi are currently given in the appendix, even though they are central quantities that are used throughout the paper. Bringing these definitions into the main text would enhance clarity and help readers follow the theoretical development more easily.

The polynomial-time convergence and approximation guarantees are built on regularity assumptions such as Lipschitz smoothness and bounded best-response mismatch coefficients. These conditions may not hold in many practical economic settings. A more detailed discussion of their limitations and how potential violations might impact convergence or solution quality would make the contribution more transparent and guide future research in this direction.

Finally, while the authors claim polynomial-time convergence to approximate equilibria, the paper provides limited information on approximation bounds or exploitability levels achievable in practice. The paper needs a clarification or acknowledgment  regarding any possible trade-off between computational effort and the quality of equilibrium approximation proposed in the paper; this would strengthen the practical relevance and interpretability of the results. Also, giving some background for those who are less familiar with the literature would enhance the readability.

**Questions:**

Apart from the key weaknesses mentioned above, some clarification and more discussion on the following points would be useful:

While the idea of infinite economies is theoretically sound and offers strong generalization, it is not very clear how the proposed theoretical results would scale when applied to large agent populations or more complex utility functions. Are there any known limitations or additional challenges that might arise in such scenarios?

Most of the analysis rely on assumptions such as Lipschitz smoothness and bounded best-response mismatch coefficients. Also, the algo needs gradients of the rewards and transition probabilities. This seems to a strong requitement. Please clarify; if so, some discussion is needed.

 What would happen if these assumptions were violated. For instance, if these regularity conditions do not hold, which may often be the case in real-world infinite economic settings, how would that affect convergence, stability, or the overall validity of the theoretical guarantees?

The computational experiments currently focus on linear and Cobb-Douglas utility functions, which are standard but relatively well-behaved.

The plot of exploitability for the Cobb-Douglas seems to have not converged to zero during the reported horizon of the computation; some discussion is needed.

Overall, to me it looks like the discussion about the computational results is very short and can be enhanced.

Could the approach be extended to more general utility forms where these assumptions do not hold? Would such cases introduce potential stability challenges such as mode collapse or convergence failure, and if so, what mitigation strategies could be considered?

The computational architecture is generative-adversarial network (GANs) and they is known to be prone to issues like mode collapse; alternatives like W-GAN are prompted by such considerations. Some discussion on the possible mode collapse would help.

It would strengthen the paper if the Authors could discuss the sensitivity of the results, particularly regarding the equilibrium convergence and approximation quality to the choice of hyperparameters, such as the learning rate, batch size, or regularization coefficients.

---

> ### Author Response · Authors · 2025-11-26
>
> We thank the reviewer for their thoughtful evaluation and their appreciation of our paper.  We begin with clarifications regarding the described weaknesses (W) and then follow up with answers to the questions (Q).
>
> > **(W1.)** The paper is primarily theoretical, with limited experimental validation. Thus, we have very limited view of the performance of the proposed approach across networks of varying sizes both in terms of the number of agents and the size of the state space. Hence, the important aspect of scalability and robustness remains underexplored.
>
> **(R1.)** We thank the reviewer for this valuable suggestion. Our main focus in this work is to establish a general theoretical and computational framework for infinite-horizon stochastic exchange economies that is both conceptually rigorous (model, existence, and convergence) and compatible with deep reinforcement learning for practical computation.
>
> While we did not optimize for computational scalability in this paper, we agree that demonstrating empirical robustness is important. To address this, we conducted **additional experiments with increased model size**—growing the number of consumers (10→20), commodities (10→20), assets (1→5), and world states (5→10). (Note that the state space is already infinite in our original experiments due to continuous endowments and type spaces.) In the meantime, we ran an additional set of experiments (please refer to the comment for Reviewer 9wkb) focusing on **economic behaviors** exhibited at the learned equilibria, and we hope these experiments serve to connect the mathematical formulation to economic meaning.
>
> As the economy becomes larger, the optimization problem naturally becomes more challenging, and we do observe slightly increased instability and greater sensitivity to learning-rate choices due to the higher variance introduced by larger economies. Nevertheless, our extended experiments consistently show clear convergence toward near-zero exploitability, demonstrating that the approach remains effective even as the scale of the economy grows.
>
> Importantly, our framework is fully compatible with common practices (e.g., entropy regularization, gradient clipping, etc.) for improving scalability and robustness in deep reinforcement learning. Incorporating these techniques is a promising and exciting direction for future work, and we expect they would further enhance performance in large-scale economies.
>
> > **(W2.)** There are many assumptions that are part of the paper’s analysis (to be expected in a theoretically oriented paper). For example, Assumption 6. But, it is deferred to the appendix. It is better to make the main paper self-contained by summarizing or at least highlighting such essential assumptions within the main text. Similarly, many theorems and results are presented with minimal discussion of their intuition or motivation. Including brief insights/intuition into the meaning and implications of these results would significantly improve the paper’s readability. In general, some discussion on the need or motivation for assumptions is needed.
> Similarly, some of the key definitions such as v^\pi and u^\pi are currently given in the appendix, even though they are central quantities that are used throughout the paper. Bringing these definitions into the main text would enhance clarity and help readers follow the theoretical development more easily.
>
> **(R2.)** We appreciate the reviewer’s comment. Some assumptions (such as Assumption 6) and definitions (e.g., $v^\pi$, $u^\pi$) were moved to the appendix due to space constraints. However, we agree that bringing key assumptions and economic intuition into the main text will greatly improve readability.
>
> In the revised version, we will: (i) move essential definitions such as $v^\pi$ (value function of policy profile) and $u^\pi$ (expected payoff of policy profile) back to Section 2; (ii) summarize Assumption 6 in the main text, noting that it mirrors standard assumptions used in competitive equilibrium existence theory; and (iii) add intuitive explanations of major theorems.
>
> > **(W3)** The polynomial-time convergence and approximation guarantees are built on regularity assumptions such as Lipschitz smoothness and bounded best-response mismatch coefficients. These conditions may not hold in many practical economic settings. A more detailed discussion of their limitations and how potential violations might impact convergence or solution quality would make the contribution more transparent and guide future research in this direction.
>
> **(R3.)** We thank the reviewer for highlighting this point. It is true that the convergence guarantees rely on regularity assumptions such as Lipschitz smoothness and bounded best-response mismatch coefficients. These assumptions ensure gradient stability and finite-variance policy-gradient estimates, but they are **not strictly necessary for practical convergence**, as observed in deep RL more broadly.
> (To be cont.)

---

> ### Author Response · Authors · 2025-11-26
>
> **(R3. cont.)**
> - **Smoothness in theory vs. practice.**
>  Neural networks act as smooth function approximators even when the underlying economic primitives (e.g., Leontief utility is not smooth) are non-smooth. Our experiments show successful learning of near-equilibrium behavior in such settings, indicating that smoothness is a sufficient condition for analysis but not a necessary condition for optimization to work in practice.
>
> - **Interpretation of the mismatch coefficient.**
> The best-response mismatch coefficient measures how badly the state visitation distribution of one player's policy (together with the adversaries’ best responses) deviates from the initial sampling distribution. It quantifies how hard it is to explore the important states needed for learning equilibrium. It is a learning-theoretic stability assumption, not an economic one, albeit it can be given an economic meaning (e.g., small individual consumption deviations do not create large shifts in market outcomes). Although this coefficient may be large or difficult to estimate in real economies, practical multi-agent RL incorporates implicit exploration mechanisms that help mitigate mismatch, such as entropy regularization or Boltzmann policies, replay buffers, and opponent sampling. These mechanisms are fully compatible with our framework because our algorithm only requires (i) differentiable policy parameterizations and (ii) trajectory-based gradient estimation. Nothing in the pseudo-game structure or min–max formulation prevents the use of these techniques.
>
> - **Overall perspective.**
> We therefore view regularity conditions such as smoothness and bounded mismatch as **technical enablers** for theoretical guarantees, similar to assumptions used in standard RL convergence analyses. Empirically, **our method succeeds even when these assumptions do not strictly hold**, and investigating robustness under weaker conditions is an exciting direction for future work.
>
> > **(W4.)** Finally, while the authors claim polynomial-time convergence to approximate equilibria, the paper provides limited information on approximation bounds or exploitability levels achievable in practice. The paper needs a clarification or acknowledgment regarding any possible trade-off between computational effort and the quality of equilibrium approximation proposed in the paper; this would strengthen the practical relevance and interpretability of the results. Also, giving some background for those who are less familiar with the literature would enhance the readability.
>
> **(R4)** We thank the reviewer for raising this important point. There is indeed an inherent trade-off between computational effort and the precision of the equilibrium approximation. From a theoretical perspective, Theorem 2.2 shows that the number of gradient iterations required to reach an $(\varepsilon,\delta)$-stationary point scales polynomially as $\widetilde{O}(1/\varepsilon)$. Thus, tighter equilibrium approximations necessarily require more computation.
>
> Practically, this means that lowering exploitability beyond a certain threshold requires proportionally more training time, similar to standard deep RL. Our experiments exhibit this behavior: exploitability decreases rapidly at first and then slows down as the policies approach near-equilibrium solutions.
>
> Despite this natural trade-off, we observe that even moderate computational budgets produce economically meaningful near-equilibria, with feasible allocations, near-clearing prices, and stable consumption and investment paths. This demonstrates that the method offers a favorable balance between computational cost and equilibrium accuracy in practice.
>
> Regarding the background, we provided supporting economic background in Appendix C: Appendix C.1 gives a historical overview of the development of general equilibrium models to stochastic economies and Appendix C.2 provides a formal definition of a static exchange economy. That said, we agree that additional discussion in the main body would enhance readability.

---

> > ### Author Response · Authors · 2025-11-26
> >
> > ## Questions
> > > **(Q1.)** While the idea of infinite economies is theoretically sound and offers strong generalization, it is not very clear how the proposed theoretical results would scale when applied to large agent populations or more complex utility functions. Are there any known limitations or additional challenges that might arise in such scenarios?
> >
> > **(A1.)** Thank you for the thoughtful question. In principle, our theoretical results apply to general infinite-horizon Markov economies regardless of the number of agents and any utility functions satisfying Assumption 6. However, it is true that practical computation becomes more challenging as economies grow in dimensionality.
> >
> > - **Utility functions.** Although our theoretical guarantees mainly apply to concave and smooth utility functions, our experiments with Leontief utilities (which are non-differentiable and violate the smoothness condition) show that our methods can, in practice, extend to any utility specification as long as the induced reward function is measurable and bounded. That said, more general utilities may violate smoothness or concavity, which can introduce additional stability challenges, especially when preferences contain sharp kinks or discontinuities. These challenges are typical in non-smooth RL and do not arise from limitations of our theoretical framework.
> > - **Large agent populations.** The primary challenge in large-population settings is variance, not correctness. The cumulative regret $\Psi(\pi, \pi’)$ aggregates best-response deviations across all agents, so when the number of agents grows, both (a) the *magnitude* of regret terms increases (requiring proper normalization), and (b) the *variance* of stochastic gradient estimates grows due to higher-dimensional joint action spaces. These issues do not violate any theoretical assumptions, but they do make optimization harder and more sensitive to learning rates in practice. Both challenges can be mitigated by standard RL techniques such as normalization of advantage terms, variance-reduction baselines, and mini-batching.
> >
> >     In our training, all experiments were run on T4 GPUs, and we are confident that the method should scale to hundreds of consumers/commodities/assets on more advanced GPU hardware.
> >
> >    Moreover, if the reviewer is interested in very large economies where the agent population approaches $n \to \infty$, this connects naturally to the literature on Heterogeneous-Agent New Keynesian (HANK) models. While HANK focuses on distributional dynamics under aggregate shocks, our model emphasizes strategic interactions between agents in stochastic exchange economies, making the two frameworks complementary.
> >
> > In summary, the theoretical framework scales to large and complex economies, but training requires careful variance control and normalization—challenges typical of high-dimensional multi-agent RL in practice, rather than limitations of the theory.
> >
> >
> > > **(Q2.)** Most of the analysis rely on assumptions such as Lipschitz smoothness and bounded best-response mismatch coefficients. Also, the algo needs gradients of the rewards and transition probabilities. This seems to a strong requitement. Please clarify; if so, some discussion is needed. What would happen if these assumptions were violated. For instance, if these regularity conditions do not hold, which may often be the case in real-world infinite economic settings, how would that affect convergence, stability, or the overall validity of the theoretical guarantees?
> >
> > **(A2.)** Thank you for raising this point. We agree that the assumptions (Lipschitz smoothness, bounded best-response mismatch, and access to policy gradients of the rewards and transition laws) may appear strong at first glance. We clarify their role and why they do **not restrict the applicability of our framework**.
> > 1) **Regularity assumptions (smoothness, mismatch).**
> >  As discussed in our response to Reviewer 3, these assumptions serve as technical enablers for establishing the polynomial-time convergence guarantees of TTSSGDA, similar to standard assumptions in policy-gradient convergence analyses for RL. Importantly:
> >     - Smoothness is *sufficient* but *not necessary* for optimization to succeed. Neural networks act as smooth approximators, and our experiments demonstrate convergence even in non-smooth environments (e.g., Leontief utility).
> >     - The mismatch coefficient is a *learning stability condition*, not an *economic assumption*. It ensures bounded variance in gradient estimates, but practical multi-agent RL incorporates implicit exploration (entropy regularization, replay buffers, sampling over past opponents) that reduces mismatch even when no theoretical bound is available.
> >
> > (To be cont.)

---

> > > ### Author Response · Authors · 2025-11-26
> > >
> > > **(A2. cont.)**
> > >
> > > 2) **Gradient requirements.**
> > > In our setting, the goal is to compute equilibria of a *known stochastic economic model*. This is standard in computational general equilibrium, where the primitives (utility, endowment shocks, price evolution) are explicitly specified. Under this assumption, we can construct a **differentiable game simulator** (e.g., [1]), which acts as a stochastic first-order oracle that returns the gradients of the rewards and transition probabilities. These gradients are queried only through automatic differentiation inside the simulator; no additional modeling assumptions are imposed on the economy itself.
> > >
> > > Thus, while the assumptions enable us to derive clean theoretical guarantees, they do not meaningfully restrict the types of economic environments we can handle. Our experiments illustrate that the approach remains effective even when these assumptions do not strictly hold.
> > >
> > > > **(Q3.)** The computational experiments currently focus on linear and Cobb-Douglas utility functions, which are standard but relatively well-behaved. The plot of exploitability for the Cobb-Douglas seems to have not converged to zero during the reported horizon of the computation; some discussion is needed.
> > >
> > > **(A3.)** Thank you for the question. While linear and Cobb–Douglas utilities are standard and well-behaved, we intentionally included **Leontief utilities**, which are **non-differentiable** and explicitly violate the smoothness assumptions used in our theory. Despite having kinks and exhibiting complementarities, our method still learns near-equilibrium behavior under Leontief preferences. This suggests that our approach is robust to violations of smoothness via approximation of the original equilibria of the problem by appropriate neural reparametrization of the problem, and that the regularity assumptions are mainly technical rather than necessary for practical convergence.
> > >
> > > Regarding the Cobb–Douglas experiment: **our GAPNet method does converge toward zero exploitability**, but the **benchmark neural projection baseline fails to do so**. This baseline attempts to directly minimize first-order optimality violations and Bellman errors via neural projection. This is most likely due to the fact that, even when these surrogate quantities are small, they only certify local KKT-style stationarity and Bellman consistency under the current policy approximation, but do not ensure global best-response optimality. In particular, approximation and distributional errors in the networks, as well as policy-class mismatch, allow the true exploitability to remain large even when the observed first-order and Bellman residuals appear small. Moreover, the objective function of the neural projection method is highly non-differentiable, which makes its convergence more unstable.
> > >
> > > > **(Q4.)** Could the approach be extended to more general utility forms where these assumptions do not hold? Would such cases introduce potential stability challenges such as mode collapse or convergence failure, and if so, what mitigation strategies could be considered?
> > >
> > > **(A4)**. Thank you for the question. As noted in **(R3)**, **(A2)** and **(A3)**, our framework can naturally extend to more general utility forms where assumptions for theoretical results do not hold.
> > > In fact, the experiments show that our method still learns near-equilibrium behavior under Leontief preferences, which is *not smooth*. Therefore, our method can, in principle, extend to any utility specification as long as the reward function is measurable and bounded. That said, more general utility forms may violate smoothness or concavity, which can introduce additional stability challenges, especially when preferences create sharp kinks and discontinuities. These challenges are typical in non-smooth RL and do not arise from limitations of our theoretical framework.
> > >
> > >
> > > [1] Hyung Ju Suh, Max Simchowitz, Kaiqing Zhang, and Russ Tedrake. Do differentiable simulators give better policy gradients? In International Conference on Machine Learning, pages 20668–20696. PMLR, 2022.

---

> > > ### Author Response · Authors · 2025-11-26
> > >
> > > > **(Q5.)** The computational architecture is generative-adversarial network (GANs) and they is known to be prone to issues like mode collapse; alternatives like W-GAN are prompted by such considerations. Some discussion on the possible mode collapse would help.
> > >
> > > **(A5.)** Thank you for the suggestion. While our min–max formulation is structurally similar to GAN training, the dynamics are fundamentally different: our maximizer is not generating synthetic data but computing best-response deviations within a known economic model, and feasibility is enforced through the dependent-policy parameterization. This greatly reduces the risk of classical GAN-style mode collapse,.
> > > That said, as with any non-convex min–max optimization, instability can arise in highly non-smooth settings, and investigating alternative formulations (e.g., Wasserstein-style objectives or regularization-based stabilizers) is an interesting direction for future work. A deeper study of stabilization techniques is beyond the scope of the current paper, but is already the object of some of our ongoing research. Thank you for the suggestion.
> > >
> > >
> > > > **(Q6.)** It would strengthen the paper if the Authors could discuss the sensitivity of the results, particularly regarding the equilibrium convergence and approximation quality to the choice of hyperparameters, such as the learning rate, batch size, or regularization coefficients.
> > >
> > >
> > > **(A6.)** Thank you for the question. We report the final hyperparameters in Appendix E.3. As predicted by Theorem 2.2, the convergence of TTSSGDA is most sensitive to the learning rates, and in particular to the *relative scale* between the minimizer and maximizer step sizes $(\eta_\omega, \eta_\sigma)$. If this ratio is not chosen appropriately, the two-timescale dynamics can become unstable and convergence is no longer guaranteed. For this reason, we conducted a hyperparameter search to identify a stable region for the learning-rate pair.
> > >
> > >
> > > In contrast, convergence is much less sensitive to other hyperparameters such as batch size. Larger batches reduce the variance of the gradient estimates but do not qualitatively affect the limiting equilibrium.
> > >
> > >
> > > Overall, the observed sensitivity aligns with theoretical expectations, and the chosen hyperparameters yield stable and consistent convergence across all reported experiments.
> > >
> > >
> > > ----------
> > > We hope our extended experiments, clarifications, and answers to questions help address your concerns. We would be happy to clarify any remaining points further during the author-reviewer discussion period.

---

### Official Review · Reviewer_9wkb · 2025-11-11

**Soundness:** 2
**Presentation:** 3
**Contribution:** 3
**Rating:** 6
**Confidence:** 2

**Summary:**

This paper introduces Markov pseudo-games (MPGs), a novel framework that unifies Markov games and pseudo-games to model dynamic, uncertain environments where agents’ feasible actions depend on others’ choices. The authors prove the existence of generalized Markov perfect equilibria (GMPE) in concave MPGs (Theorem 2.1), extending Arrow–Debreu’s pseudo-game existence theorem to dynamic settings. They further show that recursive Radner equilibria (RRE) in infinite-horizon stochastic exchange economies can be characterized as equilibria of certain MPGs (Theorem 3.1), thereby establishing the existence of equilibria in a broad class of infinite-horizon incomplete markets. Algorithmically, they propose a first-order exploitability-minimization method—implemented via a generative adversarial policy network (GAPNet)—that computes approximate equilibria in polynomial time under standard smoothness assumptions (Theorem 2.2).

**Strengths:**

1. The notion of Markov pseudo-games is conceptually elegant and unifies concepts in game theory and general equilibrium theory in a dynamic setting. The theoretical results are rigorous and intense, and build on a well-established foundation.
2. The mapping from recursive Radner equilibria to GMPE of MPGs appears novel, extending previous equilibrium existence results that were limited to representative-agent or finite-horizon settings.
3. The application of deep multi-agent RL to macroeconomic equilibrium computation is interesting.

**Weaknesses:**

1. Experiments are limited to synthetic, small-scale economies. The experimental section focuses on numerical metrics (exploitability, Bellman error) but lacks economic interpretation of the learned equilibria (e.g., consumption paths, asset prices, welfare implications). This limits the insight for economics audiences.
2. Notational issue:  $\mathcal{F}(\boldsymbol{\pi})$ is used without giving definition, where $\boldsymbol{\pi}$ is the joint policy profile of $n$-agents. The only definition I found is given other player's policy $\boldsymbol{\pi}_{-i}$ the meaning of $\mathcal{F}(\boldsymbol{\pi}\_{-i})$. Since the actions of the policies are mutually constrained in a pseudo-game setup, how does this further imply the definition of  $\mathcal{F}(\boldsymbol{\pi})$?

**Questions:**

1. Can you provide examples of equilibrium behavior in the tested economies? This would help connect the mathematical constructs to economic meaning.
2. Can you explain more on the high-level intuition of how does the "pseudo game" aspect of the formulation affects or restricts the development of the entire theory?

---

> ### Author Response · Authors · 2025-11-26
>
> We thank the reviewer for their thoughtful comments and the thorough evaluation of our work. We begin with clarification regarding the described weaknesses (W).
>
> >  **(W1.)** Experiments are limited to synthetic, small-scale economies. The experimental section focuses on numerical metrics (exploitability, Bellman error) but lacks economic interpretation of the learned equilibria (e.g., consumption paths, asset prices, welfare implications). This limits the insight for economics audiences.
>
> **R1.** We appreciate the reviewer’s suggestion to provide experiments with economic interpretation. Our primary objective in this paper is to establish a general theoretical and computational framework for infinite-horizon Markov economies, including existence results (Theorems 2.1 and 3.1), the reduction to MPGs, and the polynomial-time convergence guarantee (Theorem 2.2). That said, we fully agree that demonstrating economically meaningful equilibrium behavior is valuable for evaluating the plausibility of the infinite-horizon Markov economy model and of the equilibria learned by our method.
>
> To address this, we have conducted an additional set of experiments (experiments (1) and (2) are in the same setting as experiments in the paper), focusing not only on exploitability and Bellman error, but also on **economic behaviors** exhibited at the learned equilibria. We summarize our findings below:
>
> (1) Consumption and portfolio choices across utility specifications.
> We compare equilibrium policies under linear, Cobb–Douglas, and Leontief utilities. The commodity expenditure share is the fraction of an agent’s total wealth in a period that is allocated to consumption of goods rather than to asset purchases.
> | Utility Type                | Observed Behavior                                                                 | Economic Interpretation                                                                                                                                                                                                 |
> |-----------------------------|------------------------------------------------------------------------------------|---------------------------------------------------------------------------------------------------------------------------------------------------------------------------------------------------------------------------|
> | **Linear (Perfect Substitutes)** | • Commodity spending ≈97–98%                          | Linear utility has **no diminishing marginal utility** → agents are risk-neutral and have **no motive to smooth consumption** over time. With discounting, it is optimal to “consume now” unless asset returns are extremely large, hence minimal investment. |
> | **Cobb–Douglas**            | • Lowest commodity expenditure ≈88–90%    | Cobb–Douglas is **strictly concave**, motivating positive intertemporal smoothing. Agents optimally trade off current vs. future consumption, leading to non-zero investment.                                        |
> | **Leontief (Perfect Complements)** | • Commodity expenditure increases toward ≈99%              | Leontief exhibits **zero substitutability**; utility is limited by the scarcest good, so reducing present consumption is very costly. Agents avoid intertemporal substitution and concentrate spending on bottleneck goods, resulting in low investment. |
>
> (2) Effect of discount rate. Fixing the Cobb-Douglas utility, we compare the effects of different discount rates on consumers’ equilibrium behavior.
> | Discount Rate β                       | Observed Behavior                                      | Economic Interpretation                                                                                                                                                               |
> |---------------------------------------|---------------------------------------------------------|----------------------------------------------------------------------------------------------------------------------------------------------------------------------------------------|
> | **β = 0.4 (impatient)**               | Highest commodity expenditure share (about 0.96–0.97). | Impatient agents place relatively little weight on future utility, so they optimally shift their budget toward current goods consumption rather than saving.                            |
> | **β = 0.7 (moderately patient)**      | Intermediate commodity share (about 0.90).              | With a higher discount factor, agents value the future more and therefore allocate a larger fraction of wealth to the asset, partially smoothing consumption over time and reducing current consumption. |
> | **β = 0.9 (patient)**                 | Lowest commodity share (about 0.88–0.89).              | Very patient agents place substantial weight on future consumption and therefore invest more aggressively in the asset, implying less spending on current goods.                         |

---

> ### Author Response · Authors · 2025-11-26
>
> **(R1. cont.)** (3) Larger-scale economies. We conducted additional experiments with an increased model size—growing the number of consumers (10→20), commodities (10→20), assets (1→5), and world states (5→10). (Note that the state space is already infinite in our original experiments due to continuous endowments and type spaces.) While larger economies naturally make the optimization problem more challenging—as is typical in high-dimensional multi-agent RL—our extended experiments show that convergence remains stable and exploitability continues to decrease meaningfully, even as we scale up the model size.
>
> Figures for all experiments will be added to the revised version, and extending to calibration using real-world macroeconomic or financial data is an exciting future direction, but it is beyond the scope of the current study.
>
> > **(W2.)** Notational issue:  $\mathcal{F}\_i$  is used without giving definition, where $\pi$ is the joint policy profile of $n$-agents. The only definition I found is given other player's policy $\boldsymbol{\pi}\_{-i}$ the meaning of $\mathcal{F}\_i(\boldsymbol{\pi}\_{-i})$. Since the actions of the policies are mutually constrained in a pseudo-game setup, how does this further imply the definition of $\mathcal{F}(\boldsymbol{\pi})$?
>
> **(R2.)** We thank the reviewer for pointing out the missing definition. We will be sure to include the full definition of the *feasible policy correspondence* in all future versions of the paper.
>
> We define the *feasible policy profile correspondence* $ \mathcal{F}(\pi) \doteq \bigtimes\_{i \in [n]} \mathcal{F}\_i(\pi\_{-i}) $, meaning that a policy profile $\pi$ is feasible if and only if for all  $i\in [n]$, $\pi_i \in \mathcal{F}\_i (\pi\_{-i})$. Similarly, for a policy subclass (e.g., Markov or parametric policies), $\mathcal{F}^{sub}(\pi) \doteq \bigtimes\_{i\in [n]} \mathcal{F}\_i^{sub}(\pi\_{-i})$.
>
> This definition captures the fact that in a pseudo-game, feasibility is mutual and action-dependent, and thus forms a fixed point defined by $\pi \in \mathcal{F}(\pi)$. This fixed-point property is used explicitly in the equilibrium definition of GMPE.
>
>
> ## Questions:
> >**(Q1.)** Can you provide examples of equilibrium behavior in the tested economies? This would help connect the mathematical constructs to economic meaning.
>
> **(A1.)** Yes, thank you for the suggestion. As discussed in our response to W1, we will add experiments depicting equilibrium trajectories, illustrating consumption paths and asset holdings under different utility structures. These new experiments highlight how agents optimally choose portfolios based on the risk inherent in different utility structures, and how agents’ patience levels affect their equilibrium behavior. We hope these experiments serve to connect the mathematical formulation to economic meaning.
>
> >**(Q2.)** Can you explain more on the high-level intuition of how does the "pseudo game" aspect of the formulation affects or restricts the development of the entire theory?
>
> **(A2.)** The "pseudo-game’" structure is essential because, in exchange economies, each agent’s feasible action set depends on the actions of others. Unlike standard Markov games with fixed action spaces $\mathcal{A}\_i$​, in pseudo-games the feasible set is defined by an action-dependent correspondence $\mathcal{X}\_i (s, \boldsymbol{a}\_{-i})$.
>
> For example, in our exchange economy (Sec. 3): (1) Budget constraints depend on prices, which are chosen by the auctioneer (another agent); (2) Resource feasibility depends on aggregate consumption $\sum_{i \in [n]} x_i$, so agent i’s feasible bundle depends on others’ choices.
>
> This coupling has three implications:
> (1) Modeling: It is the minimal structure needed to extend Arrow–Debreu exchange economies to dynamic settings.
> (2) Equilibrium definition: Feasibility is endogenous and must hold for all histories, leading to a fixed-point condition $\boldsymbol{\pi} \in \mathcal{F}(\boldsymbol{\pi})$
> (3) Algorithmic consequences: The dependence of feasible sets on the outer policy prevents direct application of standard policy-gradient or best-reply methods. Our parameterization (Lemma 2) addresses this dependence by explicitly encoding “dependent policies,’’ allowing us to reformulate equilibrium computation as an unconstrained min–max problem (Eq. 7).
>
> From a technical perspective, the pseudo-game constraints matter in two ways:
> 1. Equilibrium definition: A policy profile is valid only if it is self-consistent, i.e., $\mathbf{\pi} \in \mathcal{F}(\mathbf{\pi})$, meaning each player’s policy always selects a feasible action given the actions of others. This fixed-point requirement is unique to pseudo-games.
> (To be Cont.)

---

> > ### Author Response · Authors · 2025-11-26
> >
> > **(A2. cont.)**
> >
> > 2. Algorithmic implications: The dependence of feasible actions on other players’ actions introduces an additional layer of coupling in the optimization problem (as noted below Observation 1). Standard policy-gradient methods cannot directly operate over such implicitly constrained policy classes. One contribution of our paper is to decouple this dependence using a neural-network parameterization: we construct a class of dependent policies that always output feasible actions given the outer policy. This yields an unconstrained min–max problem over parameters and enables first-order methods with theoretical guarantees (Theorem 2.2).
> >
> >
> > In summary, the pseudo-game structure is not merely a modeling preference; it is what allows us to faithfully represent infinite-horizon exchange economies, to prove equilibrium existence, and to design algorithms that operate on the resulting fixed-point-defined feasible sets.
> > We hope our clarification and proposed extensions help address your concerns. We would be happy to clarify any remaining points further during the author-reviewer discussion period.‏
> >
> > We hope our clarification and extended experiments help address your concerns. We would be happy to clarify any remaining points further during the author-reviewer discussion period.‏

---

### Author Response · Authors · 2025-11-30
**Summary Comment**

Dear Area Chair,

We thank all reviewers for their thoughtful and constructive feedback. We have carefully revised the paper in response to every comment, addressed all reviewers' concerns, and incorporated the following main changes:

1. **Expanded Experiments.** We added two new sets of experiments:
   - **Economic interpretation experiments** that compare equilibrium behavior across utility specifications and discount factors, providing clear economic meaning for the learned equilibria.
   - **Scalability experiments** on substantially larger economies (20 consumers, 20 commodities, 5 assets, 10 states), showing that the method continues to converge with near-zero exploitability even in higher-dimensional settings.
2. **Clarified Notation and Definitions.** We now include formal definitions of key objects (e.g., $\mathcal{F}(\pi)$, $u(\pi)$, $v^\pi$) in the main body to improve readability.
3. **Strengthened Discussion of Assumptions.** We significantly expanded our explanation of Assumption 3, clarified its two-fold role in encoding fixed-point feasibility and dependency structures between policy profiles, and added a new lemma giving explicit sufficient conditions under which the assumption holds. We also improved the discussion of smoothness and mismatch assumptions.

In summary, in this paper, we present **a unified theoretical and computational framework for infinite-horizon Markov exchange economies through the lens of Markov pseudo games**. By doing so, we **unifies concepts in game theory and general equilibrium theory in a dynamic setting** as noted by Reviewer 9wkb. Our technical contribution includes:
1. proving the existence of GMPE in concave Markov pseudo-games;
2. reducing recursive Radner equilibrium computation to GMPE via Markov pseudo-games; and
3. providing a provably polynomial-time deep RL algorithm for approximating GMPE/RRE, together with empirical evidence validating the approach.

These results are, to the best of our knowledge, **the first to unify infinite-horizon stochastic exchange economies with a tractable computational framework and convergence guarantees.**

Reviewers consistently highlighted the **novelty**, **significance**, and **theoretical rigor** of our work: **Reviewer 9wkb** highlighted that the notion of Markov pseudo-games is “conceptually elegant” and that the theoretical development is “rigorous and intense”. **Reviewer 9YuJ** offered especially strong praise, emphasizing that the paper tackles a “novel and challenging problem” with a “particularly interesting formulation” that "unifies infinite-horizon stochastic economies with incomplete markets using a common theoretical model". They further commented that the paper’s theorems and proofs have “good theoretical rigor,” are “carefully done,” and constitute “a good contribution to macroeconomic modelling” and underscored that the reformulation of the problem as a GMPE is a “key contribution that overcomes challenges in prior literature", and remarked that the framework “opens promising directions for future research in economic theory and computational economics”. **Reviewer giHS** found the framework “novel,” praised the extension of RRE existence results, and noted that the paper provides a “provable efficient algorithm” validated by experiments. **Reviewer hHrX** stated that the work “extends Arrow–Debreu’s existence theorem,” offers a convergent two-timescale algorithm, and includes “highly relevant” experiments, and they viewed our problem as "non-incremental". Taken together, the reviewers’ statements indicate clear support for the novelty, rigor, and significance of the paper.

Though **Reviewer giHS** expressed certain concerns about the generality of Assumption 3, we believe our new Lemma should sufficiently address this concern. We thank Reviewer giHS for highlighting this issue, as it helped us improve the clarity and comprehensiveness of the paper.

Thank you again for your consideration.

The Authors

---

### Meta-Review · Area_Chair_974g · 2026-01-10

**Summary:**

The paper proposes a new mathematical framework called Markov Pseudo-Games (MPGs). MPGs generalize Markov games. In an MPG, a player's feasible set of actions depends on the actions chosen by other players. The authors use this to model Infinite-Horizon Markov Exchange Economies. They aim to extend the classic Arrow-Debreu general equilibrium theory into dynamic, stochastic settings. Key Results: (1) Proof of existence for a game-theoretic equilibrium. (2) Demonstration that "Recursive Radner Equilibria" (RRE) in economics are a subset of these GMPEs, and (3) GAPNet: A Deep RL algorithm with polynomial-time convergence guarantees to approximate these equilibria.

The paper has received generally positive reviews, with one reviewer expressing strong skepticism regarding the theoretical assumptions. The authors have provided a rigorous defense (including a new Lemma) to justify their assumption in the rebuttal. The AC believes the technical contributions are solid and significant and hence recommends acceptance.

**Reviewer Concerns:**

Most of the concerns are addressed. Two significant concerns from giHS remain:

(1) The Validity and Generality of Assumption 3: The reviewer argues that this might "assuming away" the core difficulty of the problem. The authors argue they aren't claiming to solve the PPAD-hard global equilibrium, but rather a local stationary point. They introduced a new Lemma showing that Assumption 3 holds for any game with a DAG dependency structure. This is acceptable.

(2) The literature (Flokas et al., Kalogiannis et al.) suggests that min-max optimization with hidden structure is already well-studied, implying limited algorithmic novelty. The authors argue their contribution is the economic framework and that the min-max optimization is just a tool. They also distinguish their problem (multi-player, general-sum, non-convex) from the cited works (two-player, zero-sum, concave-convex). The AC accepts the authors' rebuttal.

**Reviewer Scores:**

I would believe most of the reviewers would either increase or remain positive after the rebuttal.

---

### Decision · Program_Chairs · 2026-01-26

Accept (Poster)